# Variational Uncertainty Decomposition for In-Context Learning

**I. Shavindra Jayasekera,** [*] **Jacob Si,** [*]
**Filippo Valdettaro, Wenlong Chen, A. Aldo Faisal, Yingzhen Li**
Imperial College London
{i.jayasekera23, y.si23, yingzhen.li}@imperial.ac.uk

## Abstract

As large language models (LLMs) gain popularity in conducting prediction tasks in-context, understanding the sources of uncertainty in in-context learning becomes essential to ensuring reliability. The recent hypothesis of in-context learning performing predictive Bayesian inference opens the avenue for Bayesian uncertainty estimation, particularly for decomposing uncertainty into epistemic uncertainty due to lack of in-context data and aleatoric uncertainty inherent in the in-context prediction task. However, the decomposition idea remains under-explored due to the intractability of the latent parameter posterior from the underlying Bayesian model. In this work, we introduce a variational uncertainty decomposition framework for in-context learning without explicitly sampling from the latent parameter posterior, by optimising auxiliary queries as probes to obtain an upper bound to the aleatoric uncertainty of an LLM's in-context learning procedure, which also induces a lower bound to the epistemic uncertainty. Through experiments on synthetic and real-world tasks, we show quantitatively and qualitatively that the decomposed uncertainties obtained from our method exhibit desirable properties of epistemic and aleatoric uncertainty. Code is available at: https://github.com/jacobyhsi/VUD.

## 1 Introduction

Large Language Models (LLMs) have demonstrated remarkable abilities in natural language generation [17, 88, 102], and are being extended to a wide range of applications such as question answering [116], retrieval-augmented generation [55], information analysis [97, 75], and bandit problems [49]. In particular, an emergent property of an LLM is *in-context learning* (ICL), where the model acquires task behavior at inference time, without the need for prior pre-training or fine-tuning [12]. With the rising importance and presence of LLMs, understanding where and why these models are uncertain is essential in assessing their trustworthiness and robustness. A straightforward method of assessing uncertainty is to directly prompt the LLM to quantify the uncertainty of its outputs. However, this can be unreliable due to the overconfidence of language models [106]. Therefore, being able to faithfully quantify and determine the sources of uncertainties from the LLMs' output can assist practitioners in better understanding and addressing the model's limitations.

Recent work has hypothesised that ICL exhibits properties of Bayesian inference [113]. If we concatenate a dataset of a predictive task $\mathcal{D} = \{(\boldsymbol{x}_i, \mathbf{y}_i)\}_{i=1}^n$ and a test input $\boldsymbol{x}^*$ into a prompt, then we can view ICL as (approximately) inferring an implicit latent parameter $\theta$ for an underlying posterior distribution $p(\theta|\mathcal{D})$ and computing a posterior predictive distribution $p(\mathbf{y}^*|\boldsymbol{x}^*, \mathcal{D})$. This interpretation allows estimation of uncertainty through a Bayesian framework, which measures a model's *total (predictive) uncertainty* by computing the entropy $\mathbb{H}[\mathbf{y}^*|\boldsymbol{x}^*, \mathcal{D}]$ or, in regression settings, the total variance $\mathrm{Var}[\mathbf{y}^*|\boldsymbol{x}^*, \mathcal{D}]$. The total uncertainty can then be decomposed further into two

---

[*]Equal contribution.

39th Conference on Neural Information Processing Systems (NeurIPS 2025).

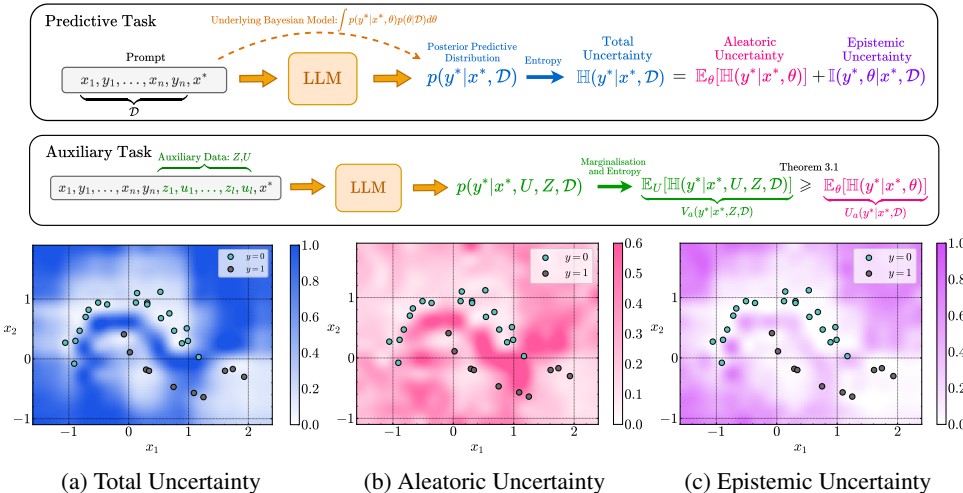

Figure 1: Uncertainty Decomposition with Auxiliary Data (Above).
Decomposition Example for Two-Moons Dataset (Below).

sources [45, 98]: *aleatoric uncertainty*, which captures noise inherent in the data generation process (thus irreducible), and *epistemic uncertainty* that accounts for uncertainty in the model due to the lack of knowledge (reducible with more data). In the bottom of Figure 1, we motivate the importance of a decomposition on the two-moons classification dataset. This decomposition provides valuable insights: aleatoric uncertainty pinpoints regions of ambiguity around the decision boundary, while epistemic uncertainty exposes areas lacking sufficient in-context data, guiding practitioners on where additional data or model refinement is needed. This notion of uncertainty decomposition has been explored in various domains, including computer vision [45, 46] and reinforcement learning [81, 18].

Obtaining high-quality Bayesian uncertainty estimates and decomposition for LLM-based ICL poses two major challenges. First, an LLM's auto-regressive prediction procedure often does not satisfy the exchangeability condition [20, 118], which questions the existence of the implicit Bayesian model with latent parameter $\theta$. Second, even if an implicit Bayesian model exists, one cannot explicitly simulate posterior samples $\theta \sim p(\theta|\mathcal{D})$, which are required by the uncertainty decomposition procedure in many existing Bayesian neural network methods [76, 11, 27, 36, 57]. In this regard, recent work on Martingale posterior [20] proposes generating a long sequence of future data and estimating a posterior distribution over $\theta$ via risk minimisation. But the Martingale posterior approach incurs a high computational cost and, still, the missing guarantee of exchangeability makes its uncertainty estimates questionable in aligning with the uncertainty from a coherent Bayesian model.

In this work, we propose a Variational Uncertainty Decomposition (VUD) framework for LLM-based ICL, focusing on addressing the mentioned two challenges. Our contributions are as follows:

- We propose an *optimisable* variational upper-bound to the aleatoric (predictive) uncertainty without explicit simulating the parameter posterior $p(\theta|\mathcal{D})$, by appending in optimisable auxiliary inputs $\mathbf{Z}$ to the context and computing uncertainty measures with $\mathbf{Z}$ conditioning. This variational estimator also induces a lower-bound on the epistemic uncertainty, which can be used in relevant tasks. An overview of our two-task variational decomposition pipeline can be found in the above of Figure 1.
- We propose novel LLM prompting and optimisation techniques for computing $p(\mathbf{y}^*|\boldsymbol{x}^*, \mathcal{D})$ and searching optimal $\mathbf{Z}$. Our design facilitates (approximate) exchangeability for ICL, making the variational uncertainty estimates better aligned with desirable Bayesian properties such as epistemic uncertainty reduction with increasing amount of data.

Experiments on synthetic regression and classification datasets show that our uncertainty decomposition framework is effective, behaving qualitatively similar to a Bayesian model. Quantitatively, the variational estimation of epistemic uncertainty also benefits downstream tasks such as bandit and in-context abstention tasks applied to real-world natural language datasets.

## 2   Background

**In-Context Learning and Bayesian Inference**. A (pre-trained) LLM with weights $\phi$ parametrises a set of conditional distributions $\{p_\phi^i(\boldsymbol{t}_i|\boldsymbol{t}_{1:i-1})\}_{i \in \mathbb{N}^+}$ over tokens $\{\boldsymbol{t}_i\}_{i \in \mathbb{N}^+}$. Given a predictive

task of covariate-label pairs, $\mathcal{D} = \{(\boldsymbol{x}_i, \mathbf{y}_i)\}_{i=1}^n$, and test covariate $\boldsymbol{x}^*$, the ICL procedure with an LLM sets $(\boldsymbol{t}_{2i-1}, \boldsymbol{t}_{2i}) = (\boldsymbol{x}_i, \mathbf{y}_i)$ and $(\boldsymbol{t}_{2n+1}, \boldsymbol{t}_{2n+2}) = (\boldsymbol{x}^*, \mathbf{y}^*)$ and computes the predictive distribution as $p(\mathbf{y}^*|\boldsymbol{x}^*, \mathcal{D}) = p_\phi^{2n+2}(\boldsymbol{t}_{2n+2}|\boldsymbol{t}_{1:2n+1})$. Now suppose the random variables $\mathbf{y}_{1:n}|\boldsymbol{x}_{1:n} \sim \prod_{i=1}^n p(\mathbf{y}_i|\boldsymbol{x}_i, \boldsymbol{x}_{<i}, \mathbf{y}_{<i})$ (with $p(\mathbf{y}_i|\boldsymbol{x}_i, \boldsymbol{x}_{<i}, \mathbf{y}_{<i}) = p_\phi^{2i}(\boldsymbol{t}_{2i}|\boldsymbol{t}_{1:2i-1})$) are *exchangeable*, namely for all permutations $\sigma$ of $[n]$,

$$p(\mathbf{y}_{\sigma(1)}, \ldots, \mathbf{y}_{\sigma(n)}|\boldsymbol{x}_{\sigma(1)}, \ldots, \boldsymbol{x}_{\sigma(n)}) = p(\mathbf{y}_1, \ldots, \mathbf{y}_n|\boldsymbol{x}_1, \ldots, \boldsymbol{x}_n), \tag{1}$$

then by de Finetti's theorem [16] there exists a Bayesian model w.r.t. a parameter $\theta$ such that

$$p(\mathbf{y}_1, \ldots, \mathbf{y}_n|\boldsymbol{x}_1, \ldots, \boldsymbol{x}_n) = \int \prod_{i=1}^n p(\mathbf{y}_i|\boldsymbol{x}_i, \theta)p(\theta)d\theta. \tag{2}$$

Notably, the parameter $\theta$ here is defined *implicitly*. We discuss the link between ICL and Bayesian models as well as existing methods to promote exchangeability further in Appendix D and F.

**Decomposing Predictive Uncertainty**. Consider a *prescribed* Bayesian model $\mathbf{y}|\boldsymbol{x} \sim p(\mathbf{y}|\boldsymbol{x}, \theta)$ with prior $\theta \sim p(\theta)$. Given a dataset $\mathcal{D} = \{(\boldsymbol{x}_i, \mathbf{y}_i)\}_{i=1}^n$, we can (approximately) compute the posterior predictive distribution $p(\mathbf{y}^*|\boldsymbol{x}^*, \mathcal{D}) = \int p(\mathbf{y}^*|\boldsymbol{x}^*, \theta)p(\theta|\mathcal{D})d\theta$. Then the predictive *total (entropic) uncertainty* is defined as $U(\mathbf{y}^*|\boldsymbol{x}^*, \mathcal{D}) = \mathbb{H}[p(\mathbf{y}^*|\boldsymbol{x}^*, \mathcal{D})]$, which can be decomposed further into *aleatoric uncertainty* $U_a(\mathbf{y}^*|\boldsymbol{x}^*, \mathcal{D})$ and *epistemic uncertainty* $U_e(\mathbf{y}^*|\boldsymbol{x}^*, \mathcal{D})$ [45]:

$$\underbrace{\mathbb{H}[p(\mathbf{y}^*|\boldsymbol{x}^*, \mathcal{D})]}_{=:U(\mathbf{y}^*|\boldsymbol{x}^*,\mathcal{D})} = \underbrace{\mathbb{E}_{p(\theta|\mathcal{D})}[\mathbb{H}[p(\mathbf{y}^*|\boldsymbol{x}^*, \theta)]]}_{=:U_a(\mathbf{y}^*|\boldsymbol{x}^*,\mathcal{D})} + \underbrace{\mathbb{I}[\mathbf{y}^*; \theta|\boldsymbol{x}^*, \mathcal{D}]}_{=:U_e(\mathbf{y}^*|\boldsymbol{x}^*,\mathcal{D})}. \tag{3}$$

The two different notions of uncertainty have distinct statistical interpretation presented as follows.

- **Aleatoric uncertainty** measures the inherent and irreducible randomness in data. Technically, under model correctness and identifiablity assumptions, there exists a parameter $\theta^*$ such that $p(\mathbf{y}|\boldsymbol{x}, \theta^*) = p_{\text{data}}(\mathbf{y}|\boldsymbol{x})$, where $\mathcal{D} \overset{\text{i.i.d.}}{\sim} p_{\text{data}}(\mathbf{y}|\boldsymbol{x})$ is the data distribution. Therefore the inherent stochasticity in data prediction can be measured via entropy $\mathbb{H}[p_{\text{data}}(\mathbf{y}^*|\boldsymbol{x}^*)] = \mathbb{H}[p(\mathbf{y}^*|\boldsymbol{x}^*, \theta^*)]$. However, $\theta^*$ is unlikely to be recovered precisely from finite observations in $\mathcal{D}$. Instead $U_a(\mathbf{y}^*|\boldsymbol{x}^*, \mathcal{D})$ defines a *Bayesian estimator* of aleatoric uncertainty, by considering the uncertainty in $\theta$ (described by the posterior $p(\theta|\mathcal{D})$) and averaging the entropy $\mathbb{H}[p(\mathbf{y}^*|\boldsymbol{x}^*, \theta)]$ over plausible $\theta \sim p(\theta|\mathcal{D})$. This estimator will converge to the true aleatoric uncertainty $\mathbb{H}[p_{\text{data}}(\mathbf{y}^*|\boldsymbol{x}^*)]$, if $p(\theta|\mathcal{D}) \to \delta(\theta = \theta^*)$ as $|\mathcal{D}| \to \infty$. We also refer to e.g., [98] for additional discussions regarding this Bayesian definition.

- **Epistemic uncertainty** reveals the model's uncertainty in prediction due to lack of knowledge from data, which is reducible by adding in new and meaningful data. Specifically, by definition of $\mathbb{I}[\mathbf{y}^*; \theta|\boldsymbol{x}^*, \mathcal{D}] = \mathbb{E}_{p(\mathbf{y}^*|\boldsymbol{x}^*, \mathcal{D})}[D_{\text{KL}}[p(\theta|\mathbf{y}^*, \boldsymbol{x}^*, \mathcal{D})||p(\theta|\mathcal{D})]]$ shows another interpretation of epistemic uncertainty as the *expected information gain* of acquiring a new datum $(\boldsymbol{x}^*, \mathbf{y}^*)$ under the current posterior belief $p(\theta|\mathcal{D})$. This motivates Bayesian active learning [40, 25] and Bayesian optimisation [63, 100, 99, 37] with epistemic uncertainty to assist the exploration-exploitation process. On the other hand, writing $\mathbb{I}[\mathbf{y}^*; \theta|\boldsymbol{x}^*, \mathcal{D}] = \mathbb{E}_{p(\theta|\mathcal{D})}[D_{\text{KL}}[p(\mathbf{y}^*|\boldsymbol{x}^*, \theta)||p(\mathbf{y}^*|\boldsymbol{x}^*, \mathcal{D})]]$, epistemic uncertainty is reflected by the *disagreement* between "experts" from the posterior $\theta \sim p(\theta|\mathcal{D})$. This leads to the use of epistemic uncertainty in detection tasks for e.g., out-of-distribution data and adversarial inputs [56].

When $\mathbf{y}^* \in \mathbb{R}$, we can also use variance as the uncertainty measure, meaning that we can compute the *total variance* of the prediction, and perform a similar decomposition into *aleatoric and epistemic variances* by the tower rule property:

$$\underbrace{\text{Var}[\mathbf{y}^*|\boldsymbol{x}, \mathcal{D}]}_{=:U^\Sigma(\mathbf{y}^*|\boldsymbol{x}^*,\mathcal{D})} = \underbrace{\mathbb{E}_{p(\theta|\mathcal{D})}[\text{Var}[\mathbf{y}^*|\boldsymbol{x}^*, \theta]]}_{=:U_a^\Sigma(\mathbf{y}^*|\boldsymbol{x}^*,\mathcal{D})} + \underbrace{\text{Var}_{p(\theta|\mathcal{D})}[\mathbb{E}[\mathbf{y}^*|\boldsymbol{x}^*, \theta]]}_{=:U_e^\Sigma(\mathbf{y}^*|\boldsymbol{x}^*,\mathcal{D})}. \tag{4}$$

Typically, these decompositions are obtained by Monte Carlo estimation with (approximate) samples from $p(\theta|\mathcal{D})$ [51]. However, this approach poses a challenge when we don't have access to $p(\theta|\mathcal{D})$, which may occur if the Bayesian model is only implicitly defined [113] as in Eq. (2), or if sampling from $p(\theta|\mathcal{D})$ is prohibitively expensive.

## 3 Method

We present an alternative approach for uncertainty decomposition defined in (3) and (4), which sidesteps explicit posterior sampling of the parameter $\theta$ and thus, is suitable for implicitly defined Bayesian models.

Although our practical algorithmic development focuses on LLM in-context learning on context $\mathcal{D} = \{(\boldsymbol{x}_i, \mathbf{y}_i)\}_{i=1}^n$ and test query $\boldsymbol{x}^*$, the decomposition technique applies to any Bayesian model *a la* de Finetti (2), including prescribed Bayesian models such as Bayesian linear regression and Gaussian processes (Appendix B).

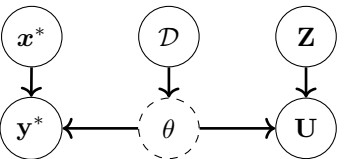

Figure 2: The DAG $\mathcal{G}$ of the conditional independence assumptions.

## 3.1 Variational Estimates of Uncertainty Decomposition

**Total Uncertainty Decomposition**. Suppose we can directly compute (or approximate) the posterior predictive distribution $p(\mathbf{y}^*|\boldsymbol{x}^*, \mathcal{D})$ for arbitrary $\mathcal{D}$ and $\boldsymbol{x}^*$. Now consider a set of *auxiliary inputs* ("queries") $\mathbf{Z} = \{\boldsymbol{z}_j\}_{j=1}^m$, and corresponding outputs ("answers") as $\mathbf{U} = \{\mathbf{u}_j\}_{j=1}^m$. Then we define the following *variational estimation* of the aleatoric uncertainty as:

$$V_a(\mathbf{y}^*|\boldsymbol{x}^*, \mathbf{Z}, \mathcal{D}) := \mathbb{E}_{p(\mathbf{U}|\mathbf{Z}, \mathcal{D})}[\mathbb{H}[p(\mathbf{y}^*|\boldsymbol{x}^*, \mathbf{U}, \mathbf{Z}, \mathcal{D})]]. \tag{5}$$

To ensure consistency with an underlying Bayesian model (2), we assume that $\boldsymbol{x}^*, \mathbf{y}^*, \mathbf{Z}, \mathbf{U}, \mathcal{D}$ obey the conditional independence relations given by the directed acyclic graph (DAG) $\mathcal{G}$ in Figure 2. This assumption allows us to prove the following theorem relating the variational estimation of the aleatoric uncertainty to the exact Bayesian estimate of aleatoric uncertainty.

**Theorem 3.1** (Aleatoric Uncertainty Upper-Bound). *If the conditional independence relations in $\mathcal{G}$ hold, then the variational estimator provides an upper-bound to the aleatoric uncertainty:*

$$V_a(\mathbf{y}^*|\boldsymbol{x}^*, \mathbf{Z}, \mathcal{D}) \geq U_a(\mathbf{y}^*|\boldsymbol{x}^*, \mathcal{D}), \tag{6}$$

*where the gap between $U_a(\mathbf{y}^*|\boldsymbol{x}^*, \mathcal{D})$ and $V_a(\mathbf{y}^*|\boldsymbol{x}^*, \mathcal{D})$ is:*

$$\mathbb{E}_{p(\mathbf{U}|\mathbf{Z}, \mathcal{D})}[\mathbb{I}[\mathbf{y}^*; \theta|\boldsymbol{x}^*, \mathbf{U}, \mathbf{Z}, \mathcal{D}]] = \mathbb{E}_{p(\mathbf{y}^*, \mathbf{U}|\boldsymbol{x}^*, \mathbf{Z}, \mathcal{D})}[D_{\mathrm{KL}}[p(\theta|\mathbf{y}^*, \boldsymbol{x}^*, \mathbf{U}, \mathbf{Z}, \mathcal{D})||p(\theta|\mathbf{U}, \mathbf{Z}, \mathcal{D})]]$$
$$= \mathbb{E}_{p(\theta, \mathbf{U}|\mathbf{Z}, \mathcal{D})}[D_{\mathrm{KL}}[p(\mathbf{y}^*|\boldsymbol{x}^*, \theta)||p(\mathbf{y}^*|\boldsymbol{x}^*, \mathbf{U}, \mathbf{Z}, \mathcal{D})]]. \tag{7}$$

See Appendix A.1 for the proof. Importantly, the upper-bound (6) holds for *arbitrary* $\mathbf{Z}$ which inspires the following optimisation procedure to obtain the best variational estimate:

$$V_a(\mathbf{y}^*|\boldsymbol{x}^*, \mathcal{D}) := \min_{\mathbf{Z}} V_a(\mathbf{y}^*|\boldsymbol{x}^*, \mathbf{Z}, \mathcal{D}), \tag{8}$$

Since the aleatoric uncertainty is trivially upper-bounded by the total uncertainty in (3), we denote

$$\tilde{V}_a(\mathbf{y}^*|\boldsymbol{x}^*, \mathcal{D}) = \min\{V_a(\mathbf{y}^*|\boldsymbol{x}^*, \mathcal{D}), \mathbb{H}[p(\mathbf{y}^*|\boldsymbol{x}^*, \mathcal{D})]\},$$

as the *variational estimate of the aleatoric uncertainty*. We can obtain a *variational estimate for the epistemic uncertainty* by defining $V_e(\mathbf{y}^*|\boldsymbol{x}^*, \mathcal{D}) := \mathbb{H}[p(\mathbf{y}^*|\boldsymbol{x}^*, \mathcal{D})] - \tilde{V}_a(\mathbf{y}^*|\boldsymbol{x}^*, \mathcal{D})$, which implies that $V_e(\mathbf{y}^*|\boldsymbol{x}^*, \mathcal{D}) \leq U_e(\mathbf{y}^*|\boldsymbol{x}^*, \mathcal{D})$, and the gap between $U_e(\mathbf{y}^*|\boldsymbol{x}^*, \mathcal{D})$ and $V_e(\mathbf{y}^*|\boldsymbol{x}^*, \mathcal{D})$ is again $\mathbb{E}_{p(\mathbf{U}|\mathbf{Z}, \mathcal{D})}[\mathbb{I}[\mathbf{y}^*; \theta|\boldsymbol{x}^*, \mathbf{U}, \mathbf{Z}, \mathcal{D}]]$. This motivates our Variational Uncertainty Decomposition approach illustrated in Figure 1. We discuss another information-theoretic view in Appendix A.1.

The effectiveness of this variational decomposition hinges on the choice of $\mathbf{Z}$ to optimise (8), which is equivalent to minimising the gap (7). Critically, similar to the two interpretations of the epistemic uncertainty $U_e(\mathbf{y}^*|\boldsymbol{x}^*, \mathcal{D})$ presented in Section 2, this gap can also be viewed from two angles.

- **Residual information gain in fantasy:** From the first definition of mutual information in (7), we see that this gap quantifies the (expected) *residual information gain* of acquiring a new datum $(\boldsymbol{x}^*, \mathbf{y}^*)$ assuming the model has further *fantasised* observations $(\mathbf{Z}, \mathbf{U})$ in addition to $\mathcal{D}$. Therefore, "clever queries" $\mathbf{Z}$, together with the fantasied answers $\mathbf{U}$, should provide sufficient information regarding the model's epistemic "belief" in $\theta$, such that further observing $\mathbf{y}^*$ and $\boldsymbol{x}^*$ does not provide much more certainty in $\theta$.

- **Remaining disagreement in fantasy:** Alternatively, from the second definition of mutual information in (7), we see that this gap also captures the expected amount of *remaining disagreement* between posterior experts after conditioning on additional *fantasised* data $(\mathbf{Z}, \mathbf{U})$. Therefore, "clever queries" $\mathbf{Z}$ should be constructed by encouraging model agreement in its epistemic "belief" of the answer $\mathbf{y}^*$ to the target query $\boldsymbol{x}^*$, after fantasising the answers $\mathbf{U}$ to the queries $\mathbf{Z}$.

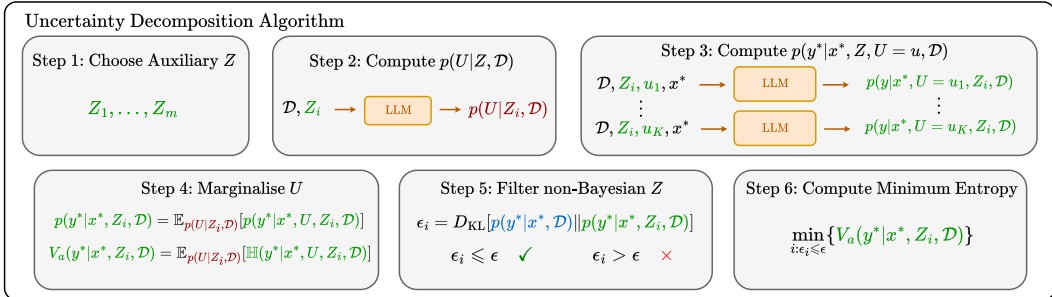

Figure 3: Variational Uncertainty Decomposition (VUD) Framework.

As a result of increased certainty of the model's subjective beliefs (in $\theta$ and/or in $\mathbf{y}^*$ given $\boldsymbol{x}^*$) after observing the fantasied data $(\mathbf{Z}, \mathbf{U})$, the conditional entropy, $V_a(\mathbf{y}^*|\boldsymbol{x}^*, \mathbf{Z}, \mathcal{D})$ is a suitable proxy for the exact Bayesian aleatoric uncertainty estimate $U_a(\mathbf{y}^*|\boldsymbol{x}^*, \mathcal{D})$. It remains an upper bound because some of the epistemic uncertainty in $\theta$ is absorbed into the aleatoric uncertainty conditioned on $\mathbf{U}$, which is reflected by the conditional expectation of the entropy of $p(\mathbf{y}^*|\boldsymbol{x}^*, \mathbf{U}, \mathbf{Z}, \mathcal{D}) = \int p(\mathbf{y}^*|\boldsymbol{x}^*, \theta)p(\theta|\mathbf{U}, \mathbf{Z}, \mathcal{D})d\theta$ when computing $V_a(\mathbf{y}^*|\boldsymbol{x}^*, \mathbf{Z}, \mathcal{D})$.

**Total Variance Decomposition**. Similarly to (8), we can also construct a variational estimate for the aleatoric variance and derive a corresponding upper-bound. See Appendix A.2 for the proof.

**Theorem 3.2** (Aleatoric Variance Upper-Bound). *If the conditional independence relation in $\mathcal{G}$ holds, then the variational estimator provides an upper-bound to the estimation of aleatoric variance:*

$$V_a^\Sigma(\mathbf{y}^*|\boldsymbol{x}^*, \mathbf{Z}, \mathcal{D}) := \mathbb{E}_{p(\mathbf{U}|\mathbf{Z}, \mathcal{D})}[\mathrm{Var}[\mathbf{y}^*|\boldsymbol{x}^*, \mathbf{U}, \mathbf{Z}, \mathcal{D}]] \geq U_a^\Sigma(\mathbf{y}^*|\boldsymbol{x}^*, \mathcal{D}). \tag{9}$$

The best variational estimate is then $V_a^\Sigma(\mathbf{y}^*|\boldsymbol{x}^*, \mathcal{D}) := \min_{\mathbf{Z}} V_a^\Sigma(\mathbf{y}^*|\boldsymbol{x}^*, \mathbf{Z}, \mathcal{D})$, and a lower-bound of the epistemic variance is obtained as $V_e^\Sigma(\mathbf{y}^*|\boldsymbol{x}^*, \mathcal{D}) := \mathrm{Var}[\mathbf{y}^*|\boldsymbol{x}, \mathcal{D}] - V_a^\Sigma(\mathbf{y}^*|\boldsymbol{x}^*, \mathcal{D})$.

## 3.2 Optimising the Variational Estimates and Promoting Exchangeability

The presented decomposition technique requires the model to be Bayesian *a la* de Finetti (2) and compatible with the DAG $\mathcal{G}$ (Figure 2), which is not the case if naively prompting LLM for in-context learning. Specifically, exchangeability requires ensuring the following necessary conditions [10, 118]:

(C1)    $p(\mathbf{y}_i|\boldsymbol{x}_i, \boldsymbol{x}_{<i}, \mathbf{y}_{<i}) = p(\mathbf{y}_i|\boldsymbol{x}_i, \sigma(\boldsymbol{x}_{<i}, \mathbf{y}_{<i}))$ for all $i \in \mathbb{N}_+$ & all permutations $\sigma$ on $[i]$;

(C2)    $p(\mathbf{y}^*|\boldsymbol{x}^*, \mathbf{Z}, \mathcal{D}) := \int p(\mathbf{y}^*|\boldsymbol{x}^*, \mathbf{U}, \mathbf{Z}, \mathcal{D})p(\mathbf{U}|\mathbf{Z}, \mathcal{D})d\mathbf{U} = p(\mathbf{y}^*|\boldsymbol{x}^*, \mathcal{D})$.

Derivations of these necessary conditions can be found in Appendix D. To promote exchangeability for LLM in-context learning, we propose two strategies tailored for the above conditions. First, to approximately achieve (C1), we construct the predictive distribution by shuffling the context and ensembling the LLM's predictions, i.e., we define for context $\mathcal{D} = \{(\boldsymbol{x}_i, \mathbf{y}_i)\}_{i=1}^n$ and test query $\boldsymbol{x}^*$ (with $S_n$ a uniform distribution over the permutations on $[n]$):

$$p(\mathbf{y}^*|\boldsymbol{x}^*, \mathcal{D}) := \frac{1}{L}\sum_{l=1}^L p_\phi^{2n+2}(\mathbf{y}^*|\boldsymbol{x}^*, \{\boldsymbol{x}_{\sigma_l(1)}, \mathbf{y}_{\sigma_l(1)}, ..., \boldsymbol{x}_{\sigma_l(n)}, \mathbf{y}_{\sigma_l(n)}\}), \quad \sigma_l \sim S_n. \tag{10}$$

The other distributions $p(\mathbf{U}|\mathbf{Z}, \mathcal{D})$ and $p(\mathbf{y}^*|\boldsymbol{x}^*, \mathbf{U}, \mathbf{Z}, \mathcal{D})$ are defined in the same manner. For classification tasks, we evaluate the LLM logits to compute (10). However, in the regression case, we make a further Gaussian approximation to (10), which allows for easy computation of the entropy and marginalisation. Further details can be found in Appendix E.2. Then to approximately satisfy (C2), we restrict the search of $\mathbf{Z}$ (Eq. (8)) to ensure the solution satisfies

$$D_{\mathrm{KL}}[p(\mathbf{y}^*|\boldsymbol{x}^*, \mathbf{Z}, \mathcal{D}) \parallel p(\mathbf{y}^*|\boldsymbol{x}^*, \mathcal{D})] < \epsilon, \tag{11}$$

for some $\epsilon > 0$. Any metric or divergence on probability distributions will suffice for (11) but we choose KL divergence due to ease of computation. We filter out the $\mathbf{Z}$ candidates that violate this KL constraint, hence we name this step as *KL filtering*. Choosing the number of permutations $L$ and the threshold $\epsilon$ for KL filtering of $\mathbf{Z}$ determines the accepted level of Bayesian approximation in

the variational decomposition. While the selection of $L$ is mainly determined by the computational resources, the choice of $\epsilon$ is further discussed in Appendix D.4.

Lastly, to reduce the search space of $\mathbf{Z}$ for efficient computation, we restrict $\mathbf{Z}$ to contain a single example in $\boldsymbol{x}$ domain, i.e., $m = 1$ and $\mathbf{Z} = \mathbf{z}$, and design sampling techniques to obtain candidates for optimal $\mathbf{Z}$, including random sampling, setting $\mathbf{Z} = \boldsymbol{x}^*$, perturbing $\mathbf{Z}$ around $\boldsymbol{x}^*$ and a Bayesian optimisation strategy [99]. Empirically we find that perturbing $\mathbf{Z}$ around $\boldsymbol{x}^*$ works best for inputs that lie in a continuous space, which can partly be explained via the Gaussian process example in Appendix B. For natural language tasks such as question-answering (QA), we conduct the perturbation of $\mathbf{z}$ by "rephrasing" $\boldsymbol{x}^*$ with another LLM. Further details regarding the sampling procedures we explored for perturbing $\mathbf{Z}$ are in Appendix C. Our overall step-by-step Variational Uncertainty Decomposition framework (VUD) is depicted in Figure 3. Detailed decomposition algorithms for classification and regression tasks are provided in Appendix E.1.

# 4    Related Work

Our work takes inspiration from the growing body of literature connecting ICL to Bayesian inference [118, 113, 41, 61]. While much of the existing research centers on estimating a latent concept, often through methods like the Martingale posterior [20, 113], we take a different route by approximating conditional entropy and mutual information using auxiliary data. While our work is not the first to decompose predictive uncertainty in LLMs into aleatoric and epistemic components, prior approaches define these uncertainties differently from their traditional definitions in Bayesian deep learning [45, 18, 108]. Huo et al. [39] analyse how uncertainty changes when a prompt is modified with additional "clarifications." While this is similar in spirit to our use of perturbations, we append perturbations to the ICL data rather than the predictive task itself. Moreover, their approach attributes aleatoric uncertainty solely to input ambiguity and does not incorporate a Bayesian framework, leading to a definition of uncertainty that diverges from the standard Bayesian interpretation. Ling et al. [60] assume a Bayesian approach but use alternative non-standard definitions of aleatoric and epistemic uncertainties. We provide a more detailed discussion of these related works, along with applications to OOD detection and bandit problems, in Appendix F.

# 5    Experiments

We evaluate the robustness and applicability of our method to classification and regression tasks. This includes ablation studies and visualisations on synthetic datasets, as well as downstream applications such as bandit problems and out-of-distribution (OOD) detection on question-answering (QA) tasks. We use the following LLMs in our experiments: Qwen2.5-14B/7B, [88] and Llama-3.1-8B [102]. Only for QA tasks, we use Qwen2.5-14B-Instruct. For conciseness, we show results for Qwen2.5-14B/14B-Instruct in the main text and the results for the remaining LLMs and baselines are given in Appendix G. Prompts and sampling details are provided in Appendix H.

## 5.1    Synthetic Regression & Classification Datasets

We visualise the uncertainty decompositions on synthetic regression & classification datasets and conduct ablation studies on the effects of KL filtering and $\mathbf{Z}$ choices. Further ablations regarding permuting the in-context examples and various LLMs are in Appendix C and D.

**Visualisations**. In Figures 4a and 4b, we visualise the VUD uncertainty decompositions for a 1-D logistic regression (classification) and a 1-D linear regression (regression) task, each conditioned on a set of $|\mathcal{D}| = 15$ in-context examples (vertical lines). We consider more complex tasks of the Two Moons dataset (class.) in Figure 1, a dataset with designated "gaps" and heteroscedastic noises in the in-context learning data (reg.) in Figure 5, and the Spirals dataset (multi-class class.) in Figure 6.

Across these examples, we observe similar qualitative characteristics of the uncertainty decomposition. The epistemic uncertainty (represented by the gap between the total and aleatoric uncertainty in the 1-D examples) is lowest in regions near demonstrations and increases as the distance to the in-context learning data increases. In the classification examples, the aleatoric uncertainty is sharply localised near the decision boundary of the problem where $p(\mathbf{y}^*|\boldsymbol{x}^*, \mathcal{D}) \approx 0.5$. In the regression setting of Figure 4b, we observe minimal change in the aleatoric uncertainty, which reflects the homoscedastic

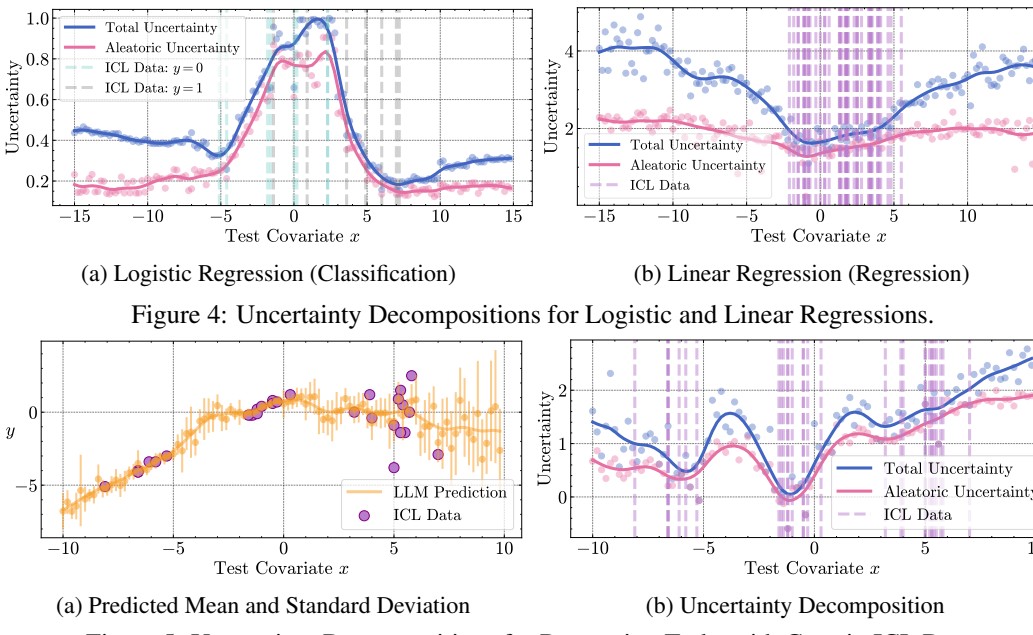

(a) Logistic Regression (Classification)    (b) Linear Regression (Regression)

Figure 4: Uncertainty Decompositions for Logistic and Linear Regressions.

(a) Predicted Mean and Standard Deviation    (b) Uncertainty Decomposition

Figure 5: Uncertainty Decompositions for Regression Tasks with Gaps in ICL Data.

noise of the data observations. However, in Figure 5 where we have heteroscedastic noise, the model accurately distinguishes between regions of high and low heteroscedastic noise. These examples indicate that the model can correctly distinguish between uncertainty from inherent data noise and uncertainty arising from missing contextual information.

**Ablations**. In Figure 7, we analyse the behavior of uncertainty decompositions as a function of in-context dataset size $|\mathcal{D}|$ under a logistic regression setting. We consider both in-distribution test inputs ($x = 0, 5$, solid lines) and out-of-distribution test inputs ($x = -15, -10, -5, 10, 15$, dotted lines). As expected, Figure 7a shows decreasing epistemic uncertainty across all test covariates with increasing $|\mathcal{D}|$, since additional training examples reduce model uncertainty. The largest epistemic uncertainty occurs at out-of-distribution inputs ($x = -15, -10, -5, 10, 15$), while in-distribution inputs ($x = 0, 5$) consistently exhibit lower values. The decay is most rapid for in-distribution test points, suggesting that the model becomes confident more quickly when the test point distribution overlaps with the training data. In contrast, aleatoric uncertainty reported in Figure 7b remains relatively stable as $|\mathcal{D}|$ grows, particularly for out-of-distribution covariates. Notably, aleatoric uncertainty is highest for the decision boundary at $x = 0$, where the class overlap is greatest, and remains consistently elevated across all dataset sizes. Out-of-distribution points show slightly lower but stable aleatoric values, reflecting lower intrinsic class ambiguity at extreme covariates. The mild increase in aleatoric uncertainty for in-distribution points at small dataset sizes is likely due to model underfitting, which resolves as more data is provided.

In Figure 8, we compare the computed aleatoric uncertainty across different $\mathbf{Z}$ sampling methods under the logistic regression setting. These include Perturb, where small noise is added to the test example to create $\mathbf{Z}$; Repeated, where $\mathbf{Z}$ is chosen to be the test example itself; Random, where $\mathbf{Z}$ is sampled uniformly from the dataset; and Bayesian Optimisation (BO) [99], where $\mathbf{Z}$ is actively selected to minimise a utility function related to the uncertainty. The aleatoric uncertainties reported in Figure 8a show that all these approaches track the total uncertainty curve around the decision boundary, indicating strong performance in capturing the local uncertainty landscape. Among them, Repeated returns the lowest variational aleatoric uncertainty estimate. Perturb also provides lower estimates, closely following the peak and providing stable estimates across the covariate space. Random sampling shows an upward trend in low ICL density regions far from the decision boundary, indicating poor stability. Regarding the KL divergence (11) achieved by the selected $\mathbf{Z}$ in Figure 8b, Random and BO consistently have the lowest KL divergence across the majority of test samples, followed by the Perturb method which is significantly faster than BO. The Repeated sampling method yields higher KL values than Perturb, indicating greater deviation from the predictive posterior and is thus less aligned with Bayesian principles. These evidences support Perturb as a scalable and well-performing approach for sampling candidate $\mathbf{Z}$ in (8)'s optimisation procedure.

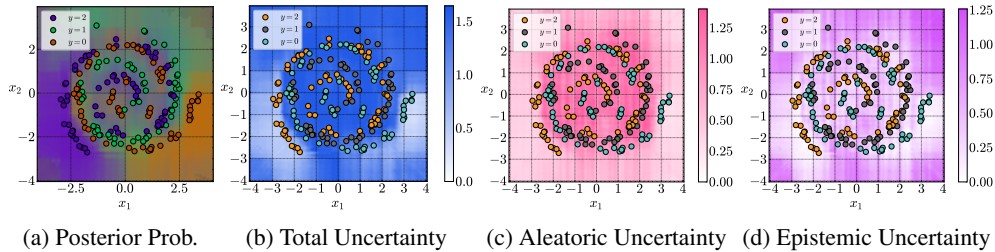

(a) Posterior Prob.  (b) Total Uncertainty  (c) Aleatoric Uncertainty  (d) Epistemic Uncertainty

Figure 6: Uncertainty Decompositions for Spirals Classification Task.

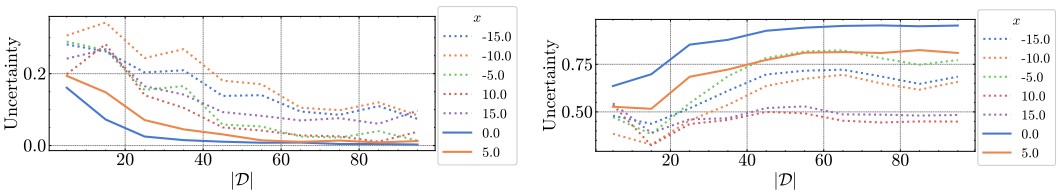

(a) Epistemic Uncertainty vs. Size of Training Set  (b) Aleatoric Uncertainty vs. Size of Training Set

Figure 7: Uncertainty decompositions for logistic regression task with varying dataset size. Solid and dotted lines indicate in-distribution and out-of-distribution predictive points respectively.

## 5.2 Downstream Applications of Uncertainty Decomposition

We conduct quantitative experiments on two applications of uncertainty decomposition: bandit problems and out-of-distribution detection in real-world question-answering tasks.

**Bandits**. Bandit problems in reinforcement learning necessitate the ability to distinguish between aleatoric and epistemic uncertainty to balance exploration and exploitation. In a bandit problem, for a trial $t$, an agent must choose an arm $a_t \in \mathcal{A}$ which gives a reward $r_t$. The goal is to minimise the overall regret over all the trials $\sum_t \mu_t^* - \mathbb{E}[r_t]$, where $\mu_t^*$ is the mean reward from the optimal arm. We consider Upper Confidence Bound (UCB) bandit algorithms [5], where $a_t = \mathrm{argmax}_a Q_t(a) + \alpha U_t(a)$, where $Q_t$ is the estimated reward from arm $a$ and $U_t$ is the uncertainty in arm $a$ at trial $t$, and $\alpha$ is the exploration rate. We use the LLM posterior mean as $Q_t$, and compare the performance of epistemic and total variance as $U_t$. In this setting, epistemic variance guides exploration to choose arms where additional data is beneficial, whereas total variance may prioritise actions where the reward has high intrinsic noise. We use the multi-armed bandit "Buttons" task [49], with 5 arms, where each arm $a$ yields a Bernoulli reward with mean $p_a$. The base reward level $p$ controls the overall success probability, with the optimal arm set to $p_a^* = p + \frac{\Delta}{2}$ and all other arms set to $p_a = p - \frac{\Delta}{2}$, where $\Delta$ denotes the reward gap between the optimal and suboptimal arms. We set $\Delta = 0.2$, which is the "hard" setting in [49]. When $p > 0.5$, the reward for the optimal arm will have the lowest (aleatoric) variance, and UCB algorithms using total variance will choose more suboptimal actions. We use mean regret and worst-case mean regret (from the 30% of worst performing seeds) as the primary performance metrics as well as metrics of median reward, suffix-fail frequency and $K \cdot \mathrm{MinFrac}$ used in [49]. We also include UCB1 and Greedy as a non-LLM baseline, and the instruction prompting method from [49] as an LLM-based non-uncertainty baseline. See Appendix G.4 for further details on metrics, results and implementation of the LLM-UCB algorithm.

Figure 9 shows a typical run of epistemic variance (EV) and total variance (TV) for a particular seed. In both examples, the $Q$ value (posterior mean reward) for the optimal arm is the highest in the last 50 trials (Figure 9b) and thus should be chosen. But when we consider the arms chosen, the optimal arm is not picked in the last 50 trials for the TV run (Figure 9a). This is because the epistemic variance decreases to zero with the number of observations but total variance does not (Figure 9c). Therefore, in the EV setting $Q_t(a)$ dominates $U_t(a)$ for large $t$, whereas for TV setting this does not necessarily hold. Table 1 shows our experimental results on the Buttons task. We see for $p > 0.5$, the worst-case regret is significantly lower for EV than TV, indicating that the UCB algorithms is more robust for EV. Furthermore, EV generally results in lower mean regret for $p > 0.5$ with the exception of $p = 0.6, \alpha = 2$. However, it is important to note bandit algorithms have high variance in mean regret due to the stochasticity of the reward.

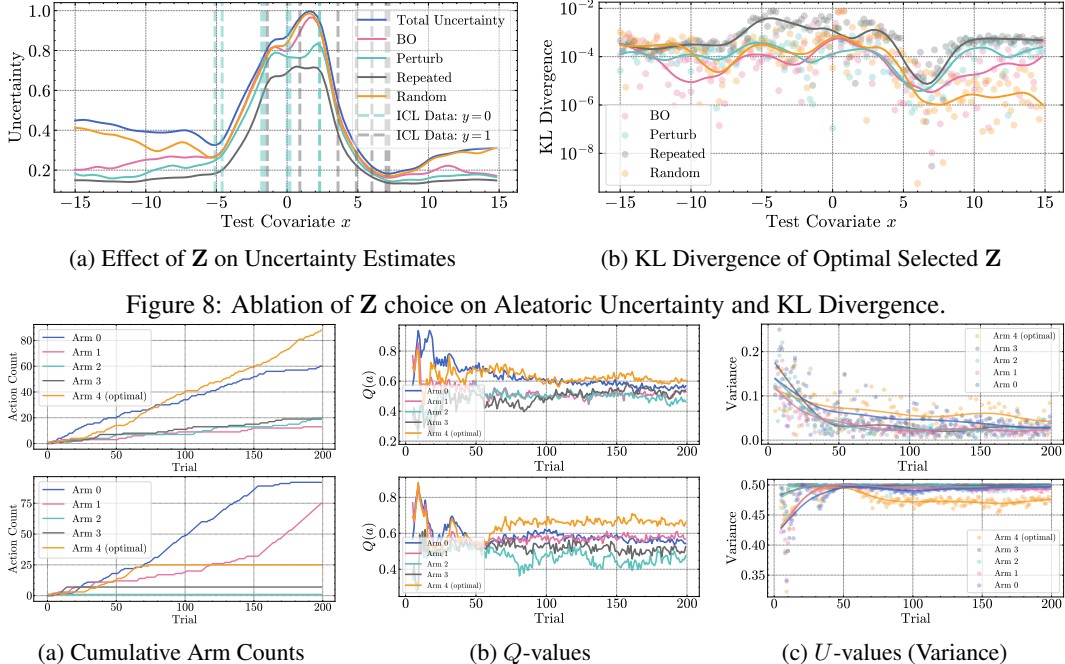

(a) Effect of **Z** on Uncertainty Estimates  (b) KL Divergence of Optimal Selected **Z**

Figure 8: Ablation of **Z** choice on Aleatoric Uncertainty and KL Divergence.

(a) Cumulative Arm Counts  (b) $Q$-values  (c) $U$-values (Variance)

Figure 9: Example Run ($p = 0.6, \alpha = 5$) with Epistemic (above) and Total Variance (below).

**In-Context Abstention**. Whilst LLMs demonstrate strong capability across a diverse range of in-context learning NLP tasks, they have been shown to also hallucinate and provide false information [44, 64, 47], impacting the reliability of these models. *LLM-Abstention* is a domain that addresses this problem by abstaining from answering questions when there is high uncertainty [105, 21]. This is achieved by abstaining from questions where the LLM exhibits uncertainty above a predetermined threshold. In abstention, we want to avoid answering under settings where the prediction is inherently uncertain and therefore, we would expect to see an improvement in accuracy when we use aleatoric uncertainty to threshold abstention compared to total uncertainty.

In our experiments, we apply LLM-abstention to binary classification datasets: BoolQA [15], HotpotQA [116], and Pub-MedQA [42]; as well as a multiclass classification dataset: MMLU [31]. We then extract the total uncertainty (TU), and the decomposed aleatoric uncertainty (AU) using VUD across 100 questions from each dataset. Our results in Table 2 demonstrate that under a threshold rate of rejecting the highest 10% of uncertain samples, using AU to threshold yields larger predictive accuracy performance gains than TU. In Figure 10, we show an example of abstention across a range of threshold rates for the MMLU dataset. We observe that thresholding by AU improves accuracy on questions that are answered.

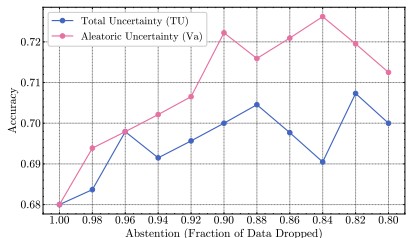

Figure 10: Effect of In-Context Abstention on Accuracy (MMLU-Moral).

We provide further details on this experiment and an example from the MMLU-Moral dataset comparing filtered samples of high aleatoric vs total uncertainty in Appendix G.5. We find that as aleatoric uncertainty measures inherent randomness and stochasticity in $p(y|x)$, AU, when compared with TU, focuses more on evaluating whether the answer to the question is inherently ambiguous.

## 5.3 Summary of Additional Experiments

We present in Appendix G further studies on additional baselines and applications. In Appendix G.2, we provided a "Martingale posterior" [22] version of predictive uncertainty decomposition. Our results show that the decomposition result is highly sensitive to the choice of the proxy model, and the Martingale posterior's estimate of total uncertainty does not agree with the predictive entropy of the LLM in-context predictive distribution. In Appendix G.5, we consider an "in-context out-of-distribution" task [32]. Our goal is to demonstrate that leveraging epistemic uncertainty from our

Table 1: Buttons Bandit Problem. TV is Total Variance and EV is Epistemic Variance.

| | METHOD | MEAN WORST-CASE REGRET ↓ | MEAN REGRET ↓ | MEDIAN REWARD ↑ | SuffFailFreq($T/2$) ↓ | $K \cdot$ MinFrac ↓ |
|---|---|---|---|---|---|---|
| $p = 0.5$ | UCB1 | $0.128_{\pm.019}$ | $0.094_{\pm.027}$ | $0.510$ | $0.0$ | $0.29$ |
| | GREEDY | $0.199_{\pm.000}$ | $0.101_{\pm.092}$ | $0.525$ | $0.460$ | $0.03$ |
| | INSTRUCT BASELINE | $0.161_{\pm.020}$ | $0.107_{\pm.043}$ | $0.495$ | $0.0$ | $0.26$ |
| | TV ($\alpha = 2$) | $0.196_{\pm.005}$ | $0.100_{\pm.074}$ | $0.492$ | $0.3$ | $\mathbf{0.03}$ |
| | EV ($\alpha = 2$) | $\mathbf{0.147}_{\pm.000}$ | $\mathbf{0.087}_{\pm.051}$ | $\mathbf{0.522}$ | $\mathbf{0.0}$ | $0.12$ |
| | TV ($\alpha = 5$) | $0.198_{\pm.000}$ | $\mathbf{0.100}_{\pm.074}$ | $0.492$ | $0.7$ | $\mathbf{0.04}$ |
| | EV ($\alpha = 5$) | $\mathbf{0.152}_{\pm.011}$ | $0.124_{\pm.024}$ | $\mathbf{0.510}$ | $\mathbf{0.0}$ | $0.60$ |
| $p = 0.6$ | UCB1 | $0.127_{\pm.018}$ | $0.094_{\pm.027}$ | $0.610$ | $0.0$ | $0.28$ |
| | GREEDY | $0.199_{\pm.000}$ | $0.092_{\pm.090}$ | $0.645$ | $0.396$ | $0.03$ |
| | INSTRUCT BASELINE | $0.111_{\pm.007}$ | $0.076_{\pm.043}$ | $0.620$ | $0.0$ | $0.18$ |
| | TV ($\alpha = 2$) | $0.198_{\pm.001}$ | $\mathbf{0.035}_{\pm.054}$ | $\mathbf{0.670}$ | $0.1$ | $\mathbf{0.04}$ |
| | EV ($\alpha = 2$) | $\mathbf{0.149}_{\pm.039}$ | $0.068_{\pm.042}$ | $0.642$ | $\mathbf{0.0}$ | $0.145$ |
| | TV ($\alpha = 5$) | $0.199_{\pm.000}$ | $0.158_{\pm.065}$ | $0.555$ | $0.8$ | $\mathbf{0.04}$ |
| | EV ($\alpha = 5$) | $\mathbf{0.140}_{\pm.013}$ | $\mathbf{0.105}_{\pm.027}$ | $\mathbf{0.600}$ | $\mathbf{0.0}$ | $0.42$ |
| $p = 0.7$ | UCB1 | $0.122_{\pm.017}$ | $0.094_{\pm.027}$ | $0.710$ | $0.0$ | $0.27$ |
| | GREEDY | $0.199_{\pm.000}$ | $0.085_{\pm.089}$ | $0.760$ | $0.369$ | $0.03$ |
| | INSTRUCT BASELINE | $0.132_{\pm.043}$ | $0.087_{\pm.040}$ | $0.703$ | $0.0$ | $0.18$ |
| | TV ($\alpha = 2$) | $0.199_{\pm.000}$ | $0.076_{\pm.087}$ | $0.725$ | $0.3$ | $\mathbf{0.03}$ |
| | EV ($\alpha = 2$) | $\mathbf{0.092}_{\pm.004}$ | $\mathbf{0.050}_{\pm.033}$ | $\mathbf{0.735}$ | $\mathbf{0.0}$ | $0.11$ |
| | TV ($\alpha = 5$) | $0.195_{\pm.003}$ | $0.151_{\pm.073}$ | $0.603$ | $0.7$ | $\mathbf{0.04}$ |
| | EV ($\alpha = 5$) | $\mathbf{0.135}_{\pm.007}$ | $\mathbf{0.092}_{\pm.037}$ | $\mathbf{0.682}$ | $\mathbf{0.0}$ | $0.24$ |

Table 2: Abstention on QA tasks. Accuracies are computed on the remaining questions after the uncertainty-based question filtering approach. Higher accuracy improvement when filtering using aleatoric uncertainty (AU) highlights the effectiveness of the uncertainty decomposition.

| | ACCURACY (%) ↑ | | |
|---|---|---|---|
| DATASET | W/OUT FILTERING | W/ FILTERING (TU) | W/ FILTERING (AU) |
| BOOLQA | $80.02_{\pm.001}$ | $\mathbf{85.56}_{\pm.002}$ | $\mathbf{85.56}_{\pm.003}$ |
| HOTPOTQA | $86.98_{\pm.000}$ | $87.78_{\pm.002}$ | $\mathbf{90.00}_{\pm.003}$ |
| PUBMEDQA | $87.02_{\pm.002}$ | $88.89_{\pm.001}$ | $\mathbf{90.00}_{\pm.000}$ |
| MMLU-CS | $81.01_{\pm.003}$ | $85.56_{\pm.002}$ | $\mathbf{86.67}_{\pm.001}$ |
| MMLU-M | $68.03_{\pm.001}$ | $70.00_{\pm.003}$ | $\mathbf{72.22}_{\pm.002}$ |

decomposition yields higher OOD detection accuracy than directly utilising the total uncertainty. We observe that for our method, epistemic uncertainty (EU) yields higher AUC scores in more ID/OOD settings than total uncertainty (TU), implying better OOD detection results via our decomposition.

## 6   Conclusion

In this work, we introduce the Variational Uncertainty Decomposition framework for ICL in LLMs. Motivated by a Bayesian view of ICL, we use auxiliary data to derive a variational upper bound to the aleatoric uncertainty and variance. This permits the estimation of the aleatoric uncertainty and variance, without requiring an estimation of the latent Bayesian parameter $\theta$. Through extensive experiments using synthetic toy and real-world datasets, we demonstrate that our method provides a sensible decomposition that qualitatively and quantitatively respects properties of epistemic and aleatoric uncertainties. These results show that our method is capable of accurately distinguishing between aleatoric and epistemic uncertainty across a variety of LLMs.

**Limitations**. We assume that ICL behaves in a Bayesian manner. Whilst there is some evidence to support this Bayesian hypothesis [113, 118, 74], it has also been observed that in longer sampling horizons this Bayesian hypothesis breaks down [20, 61]. We address this by considering short sampling horizons, permutations, and a filtering step to remove "non-Bayesian" samples. However, whilst the filtering condition is necessary for a Bayesian model, it is not sufficient and doesn't guarantee Bayesian behaviour. Therefore, we view our method as approximately Bayesian where $\epsilon$ is a quantification of the Bayesian approximation. Secondly, we focus on regression and classification tasks where the output of the task is a real number or a small set of classes and our prompt structure ensures short responses. In many real-world settings, the LLM output is in natural language where responses can differ in tokens but have the same semantic meaning. Therefore, uncertainty quantification methods that consider semantics [50] can be integrated with the VUD algorithm to obtain a posterior over the natural language response, and we leave this as future work.

## Acknowledgements

ISJ and YL acknowledge the EPSRC StatML CDT (EP/S023151/1) and BASF funding through EPSRC prosperity partnership programme IConIC (EP/X025292/1). FV was supported by a Imperial College London Department of Computing PhD scholarship. AAF holds a UKRI Turing AI Fellowship (EP/V025449/1).

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

# Appendix

## Contents

# A Proofs

## A.1 Variational Uncertainty Decomposition

**Theorem 3.1** (Aleatoric Uncertainty Upper-Bound). *If the conditional independence relations in $\mathcal{G}$ hold, then the variational estimator provides an upper-bound to the aleatoric uncertainty:*

$$V_a(\mathbf{y}^*|\boldsymbol{x}^*, \mathbf{Z}, \mathcal{D}) \geq U_a(\mathbf{y}^*|\boldsymbol{x}^*, \mathcal{D}), \tag{6}$$

*where the gap between $U_a(\mathbf{y}^*|\boldsymbol{x}^*, \mathcal{D})$ and $V_a(\mathbf{y}^*|\boldsymbol{x}^*, \mathcal{D})$ is:*

$$\mathbb{E}_{p(\mathbf{U}|\mathbf{Z},\mathcal{D})}[\mathbb{I}[\mathbf{y}^*; \theta|\boldsymbol{x}^*, \mathbf{U}, \mathbf{Z}, \mathcal{D}]] = \mathbb{E}_{p(\mathbf{y}^*, \mathbf{U}|\boldsymbol{x}^*, \mathbf{Z}, \mathcal{D})}\left[D_{\mathrm{KL}}[p(\theta|\mathbf{y}^*, \boldsymbol{x}^*, \mathbf{U}, \mathbf{Z}, \mathcal{D})||p(\theta|\mathbf{U}, \mathbf{Z}, \mathcal{D})]\right]$$
$$= \mathbb{E}_{p(\theta, \mathbf{U}|\mathbf{Z}, \mathcal{D})}\left[D_{\mathrm{KL}}[p(\mathbf{y}^*|\boldsymbol{x}^*, \theta)||p(\mathbf{y}^*|\boldsymbol{x}^*, \mathbf{U}, \mathbf{Z}, \mathcal{D})]\right]. \tag{7}$$

*Proof.* We begin by decomposing the variational estimator $V_a$, noting that from $\mathcal{G}$ we get, $p(\mathbf{y}^*|\boldsymbol{x}^*, \theta) = p(\mathbf{y}^*|\boldsymbol{x}^*, \theta, \mathbf{U}, \mathbf{Z}, \mathcal{D})$ and $\mathbf{p}(\theta|\boldsymbol{x}, \mathbf{U}, \mathbf{Z}, \mathcal{D}) = p(\theta, \mathbf{U}, \mathbf{Z}, \mathcal{D})$:

$$V_a(\mathbf{y}^*|\boldsymbol{x}^*, \mathbf{Z}, \mathcal{D}) := - \mathbb{E}_{p(\mathbf{U}|\mathbf{Z},\mathcal{D})p(\mathbf{y}^*|\boldsymbol{x}^*,\mathbf{U},\mathbf{Z},\mathcal{D})}[\log p(\mathbf{y}^*|\boldsymbol{x}^*, \mathbf{U}, \mathbf{Z}, \mathcal{D})]$$

$$= - \mathbb{E}_{p(\mathbf{U}|\mathbf{Z},\mathcal{D})p(\mathbf{y}^*|\boldsymbol{x}^*,\theta)p(\theta|\mathbf{U},\mathbf{Z},\mathcal{D})}\left[\log \frac{p(\mathbf{y}^*|\boldsymbol{x}^*, \theta)p(\theta|\mathbf{U}, \mathbf{Z}, \mathcal{D})}{p(\theta|\mathbf{y}^*, \boldsymbol{x}^*, \mathbf{U}, \mathbf{Z}, \mathcal{D})}\right] \quad (*)$$

$$= - \mathbb{E}_{p(\mathbf{U}|\mathbf{Z},\mathcal{D})p(\mathbf{y}^*|\boldsymbol{x}^*,\theta)p(\theta|\mathbf{U},\mathbf{Z},\mathcal{D})}[\log p(\mathbf{y}^*|\boldsymbol{x}^*, \theta)]$$

$$+ \mathbb{E}_{p(\mathbf{U}|\mathbf{Z},\mathcal{D})p(\mathbf{y}^*|\boldsymbol{x}^*,\mathbf{U},\mathbf{Z},\mathcal{D})p(\theta|\mathbf{y}^*,\boldsymbol{x}^*,\mathbf{U},\mathbf{Z},\mathcal{D})}\left[\log \frac{p(\theta|\mathbf{y}^*, \boldsymbol{x}^*, \mathbf{U}, \mathbf{Z}, \mathcal{D})}{p(\theta|\mathbf{U}, \mathbf{Z}, \mathcal{D})}\right] \quad (**)$$

$$= \mathbb{E}_{p(\theta|\mathcal{D})}[\mathbb{H}[p(\mathbf{y}^*|\boldsymbol{x}^*, \theta)]] \quad (***)$$
$$+ \mathbb{E}_{p(\mathbf{y}^*,\mathbf{U}|\boldsymbol{x}^*,\mathbf{Z},\mathcal{D})}[D_{\mathrm{KL}}[p(\theta|\mathbf{y}^*, \boldsymbol{x}^*, \mathbf{U}, \mathbf{Z}, \mathcal{D})||p(\theta|\mathbf{U}, \mathbf{Z}, \mathcal{D})]]$$
$$\geq \mathbb{E}_{p(\theta|\mathcal{D})}[\mathbb{H}[p(\mathbf{y}^*|\boldsymbol{x}^*, \theta)]] := U_a(\mathbf{y}^*|\boldsymbol{x}^*, \mathcal{D}).$$

Here steps $(*)$ and $(**)$ are obtained via Bayes' rule and the conditional independence assumption $\mathbf{y}^* \perp \mathbf{U}|\theta, \boldsymbol{x}^*, \mathbf{Z}, \mathcal{D}$ of DAG $\mathcal{G}$. Step $(***)$ is due to the assumption of the likelihood model $p(\mathbf{y}|\boldsymbol{x}, \theta)$ (and hence $p(\mathbf{U}|\mathbf{Z}, \theta)$) which do NOT treat $\boldsymbol{x}$ (and hence $\mathbf{Z}$) as a random variable:

$$p(\theta|\mathbf{U}, \mathbf{Z}, \mathcal{D}) = \frac{p(\mathbf{U}|\mathbf{Z}, \theta)p(\theta|\mathcal{D})}{p(\mathbf{U}|\mathbf{Z}, \mathcal{D})},$$

$$\Rightarrow \quad \int p(\theta|\mathbf{U}, \mathbf{Z}, \mathcal{D})p(\mathbf{U}|\mathbf{Z}, \mathcal{D})d\mathbf{U} = \int p(\mathbf{U}|\mathbf{Z}, \theta)p(\theta|\mathcal{D})d\mathbf{U} = p(\theta|\mathcal{D}).$$

Note that by definition of mutual information, we have:

$$\mathbb{E}_{p(\mathbf{U}|\mathbf{Z},\mathcal{D})}[\mathbb{I}[\mathbf{y}^*; \theta|\boldsymbol{x}^*, \mathbf{U}, \mathbf{Z}, \mathcal{D}]] = \mathbb{E}_{p(\mathbf{y}^*,\mathbf{U}|\boldsymbol{x}^*,\mathbf{Z},\mathcal{D})}[D_{\mathrm{KL}}[p(\theta|\mathbf{y}^*, \boldsymbol{x}^*, \mathbf{U}, \mathbf{Z}, \mathcal{D})||p(\theta|\mathbf{U}, \mathbf{Z}, \mathcal{D})]]$$
$$= \mathbb{E}_{p(\theta,\mathbf{U}|\mathbf{Z},\mathcal{D})}[D_{\mathrm{KL}}[p(\mathbf{y}^*|\boldsymbol{x}^*, \theta, \mathbf{U}, \mathbf{Z}, \mathcal{D})||p(\mathbf{y}^*|\boldsymbol{x}^*, \mathbf{U}, \mathbf{Z}, \mathcal{D})]]$$
$$= \mathbb{E}_{p(\theta,\mathbf{U}|\mathbf{Z},\mathcal{D})}[D_{\mathrm{KL}}[p(\mathbf{y}^*|\boldsymbol{x}^*, \theta)||p(\mathbf{y}^*|\boldsymbol{x}^*, \mathbf{U}, \mathbf{Z}, \mathcal{D})]], \quad (****)$$

where, again, step $(****)$ is due to the conditional independence structure $\mathbf{y}^* \perp \mathbf{U}|\theta, \boldsymbol{x}^*, \mathbf{Z}, \mathcal{D}$ of DAG $\mathcal{G}$. $\qquad\square$

*Alternative Proof.* Firstly, it is useful to define the corresponding definition of the variational approximation to the epistemic uncertainty as:

$$V_e(\mathbf{y}^*|\boldsymbol{x}^*, \mathbf{Z}, \mathcal{D}) := \mathbb{I}(\mathbf{y}^*; \mathbf{U}|\boldsymbol{x}^*, \mathbf{Z}, \mathcal{D})$$
$$= \mathbb{H}[\mathbb{E}_{p(\mathbf{U}|\mathbf{Z},\mathcal{D})}[p(\mathbf{y}^*|\boldsymbol{x}^*, \mathbf{U}, \mathbf{Z}, \mathcal{D})]] - V_a(\mathbf{y}^*|\boldsymbol{x}^*, \mathbf{Z}, \mathcal{D})$$
$$= \mathbb{H}[p(\mathbf{y}^*|\boldsymbol{x}^*, \mathbf{Z}, \mathcal{D})] - V_a(\mathbf{y}^*|\boldsymbol{x}^*, \mathbf{Z}, \mathcal{D})$$
$$= \mathbb{H}[p(\mathbf{y}^*|\boldsymbol{x}^*, \mathcal{D})] - V_a(\mathbf{y}^*|\boldsymbol{x}^*, \mathbf{Z}, \mathcal{D}), \tag{*}$$

where $(*)$ follows from the conditional independence assumption $\mathbf{y}^* \perp \mathbf{Z}|\boldsymbol{x}, \mathcal{D}$. Therefore, we have

$$V_e(\mathbf{y}^*|\boldsymbol{x}^*, \mathbf{Z}, \mathcal{D}) - U_e(\mathbf{y}^*|\boldsymbol{x}^*, \mathcal{D}) = U_a(\mathbf{y}^*|\boldsymbol{x}^*, \mathcal{D}) - V_a(\mathbf{y}^*|\boldsymbol{x}^*, \mathbf{Z}, \mathcal{D}) \tag{**}$$

If we have the conditional independence relation $\mathbf{y}^* \perp \mathbf{U}|\theta, \boldsymbol{x}, \mathbf{Z}, \mathcal{D}$, then by the data processing inequality (DPE):

$$V_e(\mathbf{y}^*|\boldsymbol{x}^*, \mathbf{Z}, \mathcal{D}) := \mathbb{I}(\mathbf{y}^*; \mathbf{U}|\boldsymbol{x}^*, \mathbf{Z}, \mathcal{D}) \overset{\mathrm{DPE}}{\leq} \mathbb{I}(\mathbf{y}^*; \theta|\boldsymbol{x}^*, \mathbf{Z}, \mathcal{D}) \overset{(\dagger)}{=} \mathbb{I}(\mathbf{y}^*; \theta|\boldsymbol{x}^*, \mathcal{D}) =: U_e(\mathbf{y}^*|\boldsymbol{x}^*, \mathcal{D}),$$

where $(\dagger)$ follows from the conditional independence relation $(\mathbf{y}^*, \theta) \perp \mathbf{Z}|\boldsymbol{x}, \mathcal{D}$. $\qquad\square$

**Remark**. From this information-theoretic perspective, we see that choosing an optimal $\mathbf{Z}$, is equivalent to maximising the mutual information between $\mathbf{y}^*$ and $\mathbf{U}$. This further motivates choosing $\mathbf{Z}$ that repeats $\boldsymbol{x}^*$ or are perturbations of $\boldsymbol{x}^*$.

## A.2 Variational Estimates of Variance Decomposition

To prove Theorem 3.2, we first prove the following lemma.

**Lemma A.1** *For any random variables $X, Y, Z$ where the conditional variances $\mathrm{Var}(Y|X)$ and $\mathrm{Var}(Y|X,Z)$ exist,*

$$\mathbb{E}[\mathrm{Var}(Y|X)] = \mathbb{E}\Big[\mathrm{Var}(\mathbb{E}[Y|X,Z]|X)\Big] + \mathbb{E}[\mathrm{Var}(Y|X,Z)] \geq \mathbb{E}[\mathrm{Var}(Y|X,Z)].$$

*Proof.* By the law of total expectation, $\mathbb{E}[\mathbb{E}(Y^2|X)] = \mathbb{E}[\mathbb{E}(Y^2|X,Z)] = \mathbb{E}[Y^2]$. Therefore,

$$\begin{aligned}
\mathbb{E}[\mathrm{Var}(Y|X)] - \mathbb{E}[\mathrm{Var}(Y|X,Z)] &= \mathbb{E}[\mathbb{E}(Y^2|X) - \mathbb{E}(Y|X)^2] - \mathbb{E}[\mathbb{E}(Y^2|X,Z) - \mathbb{E}(Y|X,Z)^2] \\
&= \underbrace{\mathbb{E}[\mathbb{E}(Y^2|X)] - \mathbb{E}[\mathbb{E}(Y^2|X,Z)]}_{=0} - \mathbb{E}[\mathbb{E}(Y|X)^2] + \mathbb{E}[\mathbb{E}(Y|X,Z)^2] \\
&= \mathbb{E}[\mathbb{E}(Y|X,Z)^2] - \mathbb{E}[\mathbb{E}(Y|X)^2].
\end{aligned}$$

To show that the LHS is positive we first decompose $\mathbb{E}(Y|X,Z)$ as

$$\mathbb{E}(Y|X,Z) = \big(\mathbb{E}(Y|X,Z) - \mathbb{E}(Y|X)\big) + \mathbb{E}(Y|X).$$

Now, the expectation of the product of these terms is 0 as

$$\begin{aligned}
\mathbb{E}\big[\big(\mathbb{E}(Y|X,Z) - \mathbb{E}(Y|X)\big) \cdot \mathbb{E}(Y|X)\big] &= \mathbb{E}\Big[\mathbb{E}\big[\big(\mathbb{E}(Y|X,Z) - \mathbb{E}(Y|X)\big) \cdot \mathbb{E}(Y|X)|X\big]\Big] \\
&= \mathbb{E}\Big[\mathbb{E}\big[\big(\mathbb{E}(Y|X,Z) - \mathbb{E}(Y|X)\big)|X\big] \cdot \mathbb{E}(Y|X)\Big] \\
&= \mathbb{E}\Big[\big(\mathbb{E}(Y|X) - \mathbb{E}(Y|X)\big) \cdot \mathbb{E}(Y|X)\Big] \qquad (*) \\
&= \mathbb{E}\Big[0 \cdot \mathbb{E}(Y|X)\Big] \\
&= 0,
\end{aligned}$$

where $(*)$ follows from the fact that $\sigma(X) \subset \sigma(X,Z)$. Therefore,

$$\begin{aligned}
\mathbb{E}[\mathbb{E}(Y|X,Z)^2] &= \mathbb{E}\Big[\Big(\big(\mathbb{E}(Y|X,Z) - \mathbb{E}(Y|X)\big) + \mathbb{E}(Y|X)\Big)^2\Big] \\
&= \mathbb{E}\Big[\underbrace{\big(\mathbb{E}(Y|X,Z) - \mathbb{E}(Y|X)\big)^2}_{=\mathrm{Var}(\mathbb{E}[Y|X,Z]|X)}\Big] + 2\underbrace{\mathbb{E}\big[\big(\mathbb{E}(Y|X,Z) - \mathbb{E}(Y|X)\big) \cdot \mathbb{E}(Y|X)\big]}_{=0} + \mathbb{E}[\mathbb{E}(Y|X)^2] \\
&= \mathbb{E}\Big[\mathrm{Var}(\mathbb{E}[Y|X,Z]|X)\Big] + \mathbb{E}[\mathbb{E}(Y|X)^2].
\end{aligned}$$

Finally, this gives

$$\mathbb{E}[\mathrm{Var}(Y|X)] - \mathbb{E}[\mathrm{Var}(Y|X,Z)] = \mathbb{E}[\mathbb{E}(Y|X,Z)^2] - \mathbb{E}[\mathbb{E}(Y|X)^2] = \mathbb{E}\Big[\mathrm{Var}(\mathbb{E}[Y|X,Z]|X)\Big] \geq 0,$$

where the final inequality follows from the non-negativity of variance. $\qquad \square$

**Theorem 3.2** (Aleatoric Variance Upper-Bound). *If the conditional independence relation in $\mathcal{G}$ holds, then the variational estimator provides an upper-bound to the estimation of aleatoric variance:*

$$V_a^{\Sigma}(\mathbf{y}^*|\boldsymbol{x}^*, \mathbf{Z}, \mathcal{D}) := \mathbb{E}_{p(\mathbf{U}|\mathbf{Z},\mathcal{D})}[\mathrm{Var}[\mathbf{y}^*|\boldsymbol{x}^*, \mathbf{U}, \mathbf{Z}, \mathcal{D}]] \geq U_a^{\Sigma}(\mathbf{y}^*|\boldsymbol{x}^*, \mathcal{D}). \qquad (9)$$

*Proof.* By the definition of $V_a^\Sigma$,

$$\begin{aligned}
V_a^\Sigma(\mathbf{y}^*|\boldsymbol{x}^*, \mathbf{Z}, \mathcal{D}) &= \mathbb{E}_{p(\mathbf{U}|\mathbf{Z}, \mathcal{D})}[\text{Var}[\mathbf{y}^*|\boldsymbol{x}^*, \mathbf{U}, \mathbf{Z}, \mathcal{D}]] \\
&= \mathbb{E}_{p(\mathbf{U}|\boldsymbol{x}^*, \mathbf{Z}, \mathcal{D})}[\text{Var}[\mathbf{y}^*|\boldsymbol{x}^*, \mathbf{U}, \mathbf{Z}, \mathcal{D}]] \\
&\geq \mathbb{E}_{p(\mathbf{U}, \theta|\boldsymbol{x}^*, \mathbf{Z}, \mathcal{D})}[\text{Var}[\mathbf{y}^*|\boldsymbol{x}^*, \mathbf{U}, \mathbf{Z}, \theta, \mathcal{D}]] && (*) \\
&= \mathbb{E}_{p(\mathbf{U}, \theta|\boldsymbol{x}^*, \mathbf{Z}, \mathcal{D})}[\text{Var}[\mathbf{y}^*|\boldsymbol{x}^*, \theta]] && (**) \\
&= \mathbb{E}_{p(\theta|\boldsymbol{x}^*, \mathbf{Z}, \mathcal{D})}[\text{Var}[\mathbf{y}^*|\boldsymbol{x}^*, \theta]] \\
&= \mathbb{E}_{p(\theta|\mathcal{D})}[\text{Var}[\mathbf{y}^*|\boldsymbol{x}^*, \theta]] \\
&= U_a^\Sigma(\mathbf{y}^*|\boldsymbol{x}^*, \mathcal{D}).
\end{aligned}$$

Here, $(*)$ follows from Lemma A.1 and $(**)$ follows from the conditional independence relation $\mathbf{y}^* \perp \mathbf{Z}, \mathbf{U}, \mathcal{D}|\boldsymbol{x}^*, \theta$. $\qquad\square$

**Remark.** From Lemma A.1, we also obtain that the discrepancy between $V_a^\Sigma(\mathbf{y}^*|\boldsymbol{x}^*, \mathbf{Z}, \mathcal{D})$ and $U_a^\Sigma(\mathbf{y}^*|\boldsymbol{x}^*, \mathcal{D})$ is:

$$\mathbb{E}\Big[\text{Var}(\mathbb{E}[\mathbf{y}^*|\theta, \boldsymbol{x}^*, \mathbf{U}, \mathbf{Z}, \mathcal{D}]|\boldsymbol{x}^*, \mathbf{U}, \mathbf{Z}, \mathcal{D})\Big|\boldsymbol{x}^*, \mathbf{Z}, \mathcal{D}\Big]$$
$$= \mathbb{E}_{p(\mathbf{U}|\mathbf{Z}, \mathcal{D})}\Big[\text{Var}_{p(\theta|\mathbf{U}, \mathbf{Z}, \mathcal{D})}(\mathbb{E}[\mathbf{y}^*|\theta, \boldsymbol{x}^*]|\mathbf{U}, \mathbf{Z}, \mathcal{D})\Big|\mathbf{Z}, \mathcal{D}\Big].$$

# B   Theoretical Examples

## B.1   Bayesian Linear Regression

Consider a linear regression model with homogeneous output noise variance. Namely, we assume a normal prior $p(\theta) = \mathcal{N}(\theta; \mathbf{0}, \lambda^{-1}\mathbf{I}_d)$, and the likelihood model is $p(\mathbf{y}|\boldsymbol{x}, \theta) := \mathcal{N}(\mathbf{y}; \theta^\top \boldsymbol{x}, \sigma^2)$. Denote $\mathbf{X} = [\boldsymbol{x}_1, ..., \boldsymbol{x}_n]^\top \in \mathbb{R}^{n \times d}$ and $\mathbf{Z} = [\mathbf{z}_1, ..., \mathbf{z}_m]^\top \in \mathbb{R}^{m \times d}$. Now consider the exact posterior predictive distributions which can be shown as:

$$p(\theta|\mathcal{D}) = \mathcal{N}(\theta; \boldsymbol{\mu}, \Lambda^{-1}), \quad \Lambda := \sigma^{-2}\mathbf{X}^\top\mathbf{X} + \lambda\mathbf{I}_d, \quad \boldsymbol{\mu} := \Lambda^{-1}\mathbf{X}^T y,$$
$$p(\mathbf{y}^*|\boldsymbol{x}^*, \mathcal{D}) = \mathcal{N}(\mathbf{y}^*; \boldsymbol{\mu}^\top\boldsymbol{x}^*, (\boldsymbol{x}^*)^\top\Lambda^{-1}\boldsymbol{x}^* + \sigma^2).$$

Then using the closed-form expressions for the entropy of a Gaussian distribution, it is straightforward to show that for arbitrary $\mathbf{y}^*, \boldsymbol{x}^*$ and $\mathcal{D}$:

$$\begin{aligned}
U_a(\mathbf{y}^*|\boldsymbol{x}^*, \mathcal{D}) &= \frac{1}{2}(1 + \log 2\pi\sigma^2), \\
U_e(\mathbf{y}^*|\boldsymbol{x}^*, \mathcal{D}) &= \frac{1}{2}\log((\boldsymbol{x}^*)^\top\Lambda^{-1}\boldsymbol{x}^* + \sigma^2) - \frac{1}{2}\log\sigma^2,
\end{aligned}$$

Adding the auxiliary data $\mathbf{Z}, \mathbf{U}$:

$$p(\theta|\mathbf{U}, \mathbf{Z}, \mathcal{D}) = \mathcal{N}(\theta; \boldsymbol{\mu}(\mathbf{Z}), \Lambda^{-1}(\mathbf{Z})), \quad \Lambda(\mathbf{Z}) := \sigma^{-2}(\mathbf{X}^\top\mathbf{X} + \mathbf{Z}^\top\mathbf{Z}) + \lambda\mathbf{I}_d,$$
$$p(\mathbf{y}^*|\boldsymbol{x}^*, \mathbf{U}, \mathbf{Z}, \mathcal{D}) = \mathcal{N}(\mathbf{y}^*; \boldsymbol{\mu}(\mathbf{Z})^\top\boldsymbol{x}^*, (\boldsymbol{x}^*)^\top\Lambda^{-1}(\mathbf{Z})\boldsymbol{x}^* + \sigma^2\mathbf{I}_d).$$

Since the variance of $p(\mathbf{y}^*|\boldsymbol{x}^*, \mathbf{U}, \mathbf{Z}, \mathcal{D})$ does not depend on $\mathbf{y}^*$ and $\mathbf{U}$, this leads to

$$\begin{aligned}
V_a(\mathbf{y}^*|\boldsymbol{x}^*, \mathbf{Z}, \mathcal{D}) &= \frac{1}{2}(1 + \log 2\pi) + \frac{1}{2}\log((\boldsymbol{x}^*)^\top\Lambda^{-1}(\mathbf{Z})\boldsymbol{x}^* + \sigma^2), \\
V_e(\mathbf{y}^*|\boldsymbol{x}^*, \mathcal{D}) &= \frac{1}{2}\log((\boldsymbol{x}^*)^\top\Lambda^{-1}\boldsymbol{x}^* + \sigma^2) - \frac{1}{2}\log((\boldsymbol{x}^*)^\top\Lambda^{-1}(\mathbf{Z})\boldsymbol{x}^* + \sigma^2),
\end{aligned}$$

It is easy to show for all possible $\mathbf{Z}$:

$$V_a(\mathbf{y}^*|\boldsymbol{x}^*, \mathbf{Z}, \mathcal{D}) - U_a(\mathbf{y}^*|\boldsymbol{x}^*, \mathcal{D}) = \frac{1}{2}\log(\sigma^{-2}(\boldsymbol{x}^*)^\top\Lambda^{-1}(\mathbf{Z})\boldsymbol{x}^* + 1) \geq 0.$$

Now consider the optimum of the variational estimate:

$$V_a(\mathbf{y}^*|\boldsymbol{x}^*, \mathcal{D}) := \frac{1}{2}(1 + \log 2\pi\sigma) + \min_{\mathbf{Z}}\frac{1}{2}\log(\sigma^{-2}(\boldsymbol{x}^*)^\top\Lambda^{-1}(\mathbf{Z})\boldsymbol{x}^* + 1),$$

where $\Lambda(\mathbf{Z}) := \sigma^{-2}(\mathbf{X}^\top\mathbf{X} + \mathbf{Z}^\top\mathbf{Z}) + \lambda\mathbf{I}_d$. Now, if $\gamma$ is the minimum eigenvalue of $(\mathbf{X}^\top\mathbf{X} + \mathbf{Z}^\top\mathbf{Z})$ and $\gamma > 0$, then $(\boldsymbol{x}^*)^\top\Lambda^{-1}\boldsymbol{x}^* \leq \frac{1}{\gamma}\|\boldsymbol{x}^*\|_2^2$. If $m \geq d$, we can choose $\boldsymbol{z}_j$ (e.g. unit vectors) such that $\lambda > 0$, and then scaling $\boldsymbol{z}_j$ by a constant ensures $\gamma \to \infty$ and $(\boldsymbol{x}^*)^\top\Lambda^{-1}\boldsymbol{x}^* \to 0$. Therefore, for appropriately chosen $\mathbf{Z}$, $V_a(\mathbf{y}^*|\boldsymbol{x}^*, \mathbf{Z}, \mathcal{D}) \to U_a(\mathbf{y}^*|\boldsymbol{x}^*, \mathcal{D})$.

## B.2 Gaussian Process Regression

Here we assume a Gaussian process model [90] with a kernel function as the prior covariance:

$$y = f(\boldsymbol{x}) + \sigma\epsilon, \quad \epsilon \sim \mathcal{N}(0, 1), \quad f(\cdot) \sim \mathcal{GP}(0, k(\cdot, \cdot)).$$

Here we assume 1D outputs w.l.o.g. and use notations $\mathbf{y}, y$ interchangeably. For regression problems we have closed form solution to the posterior predictive (with $\mathcal{D} = (\mathbf{X}, \mathbf{Y})$, we omit the formulation of the posterior mean $\mu(\mathbf{X}, \mathbf{Y})$ and focus the discussion on the posterior variance only):

$$p(y^*|\boldsymbol{x}^*, \mathbf{X}, \mathbf{Y}) = \mathcal{N}(y^*; \mu(\mathbf{X}, \mathbf{Y}), k(\boldsymbol{x}^*, \boldsymbol{x}^*) - \mathbf{K}_{*\mathbf{X}}(\mathbf{K}_{\mathbf{XX}} + \sigma^2\mathbf{I})^{-1}\mathbf{K}_{\mathbf{X}*} + \sigma^2),$$

leading to the following uncertainty estimates:

$$U_a(\mathbf{y}^*|\boldsymbol{x}^*, \mathcal{D}) = \frac{1}{2}(1 + \log 2\pi\sigma^2),$$

$$U_e(\mathbf{y}^*|\boldsymbol{x}^*, \mathcal{D}) = \frac{1}{2}\log(k(\boldsymbol{x}^*, \boldsymbol{x}^*) - \mathbf{K}_{*\mathbf{X}}(\mathbf{K}_{\mathbf{XX}} + \sigma^2\mathbf{I})^{-1}\mathbf{K}_{\mathbf{X}*} + \sigma^2) - \frac{1}{2}\log\sigma^2.$$

Now consider sparse variational Gaussian process (SVGP) [35] with inducing inputs/outputs $\mathbf{Z}, \mathbf{u}$ and an approximating distribution $q(\mathbf{u}) := \mathcal{N}(\mathbf{u}; \boldsymbol{m}, \mathbf{S})$. Then we have the approximate posterior predictive as:

$$q(y^*) = \mathcal{N}(y^*; \mu(\boldsymbol{x}^*), k(\boldsymbol{x}^*, \boldsymbol{x}^*) - \mathbf{K}_{*\mathbf{Z}}\mathbf{K}_{\mathbf{ZZ}}^{-1}(\mathbf{K}_{\mathbf{ZZ}} - \mathbf{S})\mathbf{K}_{\mathbf{ZZ}}^{-1}\mathbf{K}_{\mathbf{Z}*} + \sigma^2),$$

so that the decomposed uncertainty estimates from an SVGP are

$$\tilde{U}_a(\mathbf{y}^*|\boldsymbol{x}^*; q) = \frac{1}{2}(1 + \log 2\pi\sigma^2) = U_a(\mathbf{y}^*|\boldsymbol{x}^*; q),$$

$$\tilde{U}_e(\mathbf{y}^*|\boldsymbol{x}^*; q) = \frac{1}{2}\log(k(\boldsymbol{x}^*, \boldsymbol{x}^*) - \mathbf{K}_{*\mathbf{Z}}\mathbf{K}_{\mathbf{ZZ}}^{-1}(\mathbf{K}_{\mathbf{ZZ}} - \mathbf{S})\mathbf{K}_{\mathbf{ZZ}}^{-1}\mathbf{K}_{\mathbf{Z}*} + \sigma^2) - \frac{1}{2}\log\sigma^2.$$

For regression problems we have the optimal $\mathbf{S} = \mathbf{K}_{\mathbf{ZZ}}(\mathbf{K}_{\mathbf{ZZ}} + \sigma^{-2}\mathbf{K}_{\mathbf{ZX}}\mathbf{K}_{\mathbf{XZ}})^{-1}\mathbf{K}_{\mathbf{ZZ}}$ [35], and therefore

$$\tilde{U}_e(\mathbf{y}^*|\boldsymbol{x}^*; q) = \frac{1}{2}\log(k(\boldsymbol{x}^*, \boldsymbol{x}^*) - \mathbf{K}_{*\mathbf{Z}}(\mathbf{K}_{\mathbf{ZZ}}^{-1} - (\mathbf{K}_{\mathbf{ZZ}} + \sigma^{-2}\mathbf{K}_{\mathbf{ZX}}\mathbf{K}_{\mathbf{XZ}})^{-1})\mathbf{K}_{\mathbf{Z}*} + \sigma^2) - \frac{1}{2}\log\sigma^2.$$

On the other hand, using the variational uncertainty decomposition method, we can show that

$$p(\mathbf{y}^*|\boldsymbol{x}^*, \mathbf{U}, \mathbf{Z}, \mathbf{X}, \mathbf{y}) = \mathcal{N}(\mathbf{y}^*; \boldsymbol{\mu}(\mathbf{Z}, \mathbf{U}), k(\boldsymbol{x}^*, \boldsymbol{x}^*) - \mathbf{K}_{*\mathbf{X}}(\mathbf{K}_{\mathbf{XX}} + \sigma^2\mathbf{I})^{-1}\mathbf{K}_{\mathbf{X}*} - \Delta(\boldsymbol{x}^*, \mathbf{Z}) + \sigma^2),$$

$$\Delta(\boldsymbol{x}^*, \mathbf{Z}) = \boldsymbol{A}^\top(\mathbf{K}_{\mathbf{ZZ}} + \sigma^2\mathbf{I} - \mathbf{K}_{\mathbf{ZX}}(\mathbf{K}_{\mathbf{XX}} + \sigma^2\mathbf{I})^{-1}\mathbf{K}_{\mathbf{XZ}})^{-1}\boldsymbol{A},$$

$$\boldsymbol{A} = \mathbf{K}_{\mathbf{Z}*} - \mathbf{K}_{\mathbf{ZX}}(\mathbf{K}_{\mathbf{XX}} + \sigma^2\mathbf{I})^{-1}\mathbf{K}_{\mathbf{X}*},$$

leading to the following uncertainty estimates (with $\mathcal{D} = (\mathbf{X}, \mathbf{y})$ and $C := \frac{1}{2}(1 + \log 2\pi)$):

$$V_a(\mathbf{y}^*|\boldsymbol{x}^*, \mathbf{Z}, \mathcal{D}) = C + \frac{1}{2}\log(k(\boldsymbol{x}^*, \boldsymbol{x}^*) - \mathbf{K}_{*\mathbf{X}}(\mathbf{K}_{\mathbf{XX}} + \sigma^2\mathbf{I})^{-1}\mathbf{K}_{\mathbf{X}*} - \Delta(\boldsymbol{x}^*, \mathbf{Z}) + \sigma^2),$$

$$V_e(\mathbf{y}^*|\boldsymbol{x}^*, \mathbf{Z}, \mathcal{D}) = C + \frac{1}{2}\log(k(\boldsymbol{x}^*, \boldsymbol{x}^*) - \mathbf{K}_{*\mathbf{X}}(\mathbf{K}_{\mathbf{XX}} + \sigma^2\mathbf{I})^{-1}\mathbf{K}_{\mathbf{X}*} + \sigma^2) - V_a(\mathbf{y}^*|\boldsymbol{x}^*, \mathbf{Z}, \mathcal{D}).$$

Note that if we choose $\mathbf{Z} = \boldsymbol{x}^*$ then we have

$$V_a(\mathbf{y}^*|\boldsymbol{x}^*, \boldsymbol{x}^*, \mathcal{D}) = U_a(\mathbf{y}^*|\boldsymbol{x}^*, \mathcal{D}) + \frac{1}{2}\log\left(2 - \frac{\sigma^2}{k(\boldsymbol{x}^*, \boldsymbol{x}^*) - \mathbf{K}_{*\mathbf{X}}(\mathbf{K}_{\mathbf{XX}} + \sigma^2\mathbf{I})^{-1}\mathbf{K}_{\mathbf{X}*} + \sigma^2}\right),$$

$$V_e(\mathbf{y}^*|\boldsymbol{x}^*, \boldsymbol{x}^*, \mathcal{D}) = U_e(\mathbf{y}^*|\boldsymbol{x}^*, \mathcal{D}) - \frac{1}{2}\log\left(2 - \frac{\sigma^2}{k(\boldsymbol{x}^*, \boldsymbol{x}^*) - \mathbf{K}_{*\mathbf{X}}(\mathbf{K}_{\mathbf{XX}} + \sigma^2\mathbf{I})^{-1}\mathbf{K}_{\mathbf{X}*} + \sigma^2}\right).$$

This means if the test input $\boldsymbol{x}^*$ is close to the training data $\mathbf{X}$, then $k(\boldsymbol{x}^*, \boldsymbol{x}^*) - \mathbf{K}_{*\mathbf{X}}(\mathbf{K}_{\mathbf{XX}} + \sigma^2\mathbf{I})^{-1}\mathbf{K}_{\mathbf{X}*}$ will be close to zero, and then $V_e(\mathbf{y}^*|\boldsymbol{x}^*, \boldsymbol{x}^*, \mathcal{D}) \approx U_e(\mathbf{y}^*|\boldsymbol{x}^*, \mathcal{D})$ provides a good estimate of the epistemic uncertainty.

### B.3   Gaussian Bandits

In Gaussian bandit setting, suppose we have independence of rewards between arms. Furthermore, for an arm $i$, we assume the following conjugate Gaussian model:

1. Gaussian Prior: $p(\theta_i) = \mathcal{N}(0, \sigma_0^2)$
2. Gaussian Likelihood: $p(r_i \mid \theta_i) = \mathcal{N}(\theta_i, \sigma^2)$

Then for $k_t$ observations of rewards from arm $i$, we have:

1. Total variance: $U^\Sigma = \sigma^2 + \left[ \frac{1}{\sigma_0^2} + \frac{k_t}{\sigma^2} \right]^{-1}$
2. True aleatoric variance: $U_a^\Sigma = \sigma^2$
3. True epistemic variance (total $-$ aleatoric): $U_e^\Sigma = \left[ \frac{1}{\sigma_0^2} + \frac{k_t}{\sigma^2} \right]^{-1}$

However, with the VUD method, suppose we have $n$ further auxiliary observations (i.e., predicted reward values from the model) for arm $i$, then, the corresponding variational estimate of aleatoric uncertainty is:

$$V_a^\Sigma = \sigma^2 + \left[ \frac{1}{\sigma_0^2} + \frac{k_t + n}{\sigma^2} \right]^{-1} \geq U_a^\Sigma.$$

This gives the gap between the variational estimate and the exact aleatoric variance as:

$$V_a^\Sigma - U_a^\Sigma = \left[ \frac{1}{\sigma_0^2} + \frac{k_t + n}{\sigma^2} \right]^{-1} = \mathcal{O}\left( \frac{1}{k_t + n} \right) = \mathcal{O}\left( \frac{1}{n} \right).$$

## C   Sampling Methods for Auxiliary Data

In this section, we discuss in detail the methods used to sample auxiliary queries $\mathbf{Z}$ to find the best variational estimate of the aleatoric uncertainty and variance. As noted in Section 3.1, we restrict $\mathbf{Z}$ to a single query in the $\boldsymbol{x}$ domain to reduce the search space.

### C.1   Methods

**Bayesian Optimisation**. The optimisation problem (8) can be directly optimised via Bayesian Optimisation. However, this is a constrained optimisation problem where $\mathbf{Z}$ needs to satisfy an "approximately Bayesian" criterion (11) which we discuss in Section 3.2. To overcome this issue, we treat the problem as an unconstrained Bayesian optimisation task to obtain auxiliary examples $\{\boldsymbol{z}_j\}_{j=1}^m$ and then apply the criterion to remove auxiliary examples that do not satisfy (11). In the synthetic examples we consider, the covariates $\boldsymbol{x}_i$ are real and continuous. Therefore, we use a Gaussian process with an RBF kernel to model the objective function and take the log expected improvement as the acquisition function. In order to provide a warm start to the Bayesian optimisation process, we provide 5 initial samples that are randomly sampled.

**Perturb**. Given the covariates $\boldsymbol{x}^*$ of the data point we wish to decompose the uncertainty for, we can choose $\mathbf{Z} = \{\boldsymbol{z}_j\}_{i=1}^m$ to be "close" to $\boldsymbol{x}^*$. To perturb a categorical covariate $\boldsymbol{x}^*[k]$, we sample uniformly from the list of categories with probability $p$ and keep the original covariate with probability $1 - p$. For a real covariate $\boldsymbol{x}^*[k']$, we sample from a normal distribution, similarly to random sampling, but we choose the mean as $x_k^*$ and the standard deviation as a scaled population standard deviation estimate of the covariate $\gamma \cdot \sigma_{k'}^\mathcal{D}$ where $\gamma = 0.1$.

**Repeated**. Given the test covariates $\boldsymbol{x}^*$, we set $\mathbf{Z} = \boldsymbol{x}^*$. Since we repeat the covariates, we only evaluate 1 auxiliary query per test example, and therefore the KL filtering procedure is omitted.

**Random Sampling**. The most basic sampling procedure to generate auxiliary queries $\mathbf{Z}$ is to randomly sample in the input domain. If a covariate $\boldsymbol{x}^*[k]$ is a categorical variable, we sample uniformly from the list of categories. If a covariate $\boldsymbol{x}^*[k']$ is a real variable, we assume a normal distribution with mean and standard deviation given by the population mean and standard deviation estimates of the covariate, $\mu_{k'}^\mathcal{D}$ and $\sigma_{k'}^\mathcal{D}$ from the in-context data $\mathcal{D}$.

## C.2 Ablations on Logistic Regression Data

We compare the performance of the four approaches to choose $\mathbf{Z}$ outlined in Section C.1 for 15 auxiliary examples (with the exception of the Repeated where we have a single auxiliary example). We plot the uncertainty decompositions for the $\mathbf{Z}$ sampling approaches and the corresponding KL divergence for the chosen $\mathbf{Z}$ that minimises (8) in Figures 8, 11 and 12. In Tables 3 and 4 we quantify the performance of each of the sampling methods by computing the mean rank of each method over the test samples. For the 3 LLMs that we consider, we consistently observe that Repeated has the lowest $V_a$, followed by Perturbations. However, Perturbations has the highest KL divergence, which indicates that this method is less aligned with the Bayesian assumptions that we make.

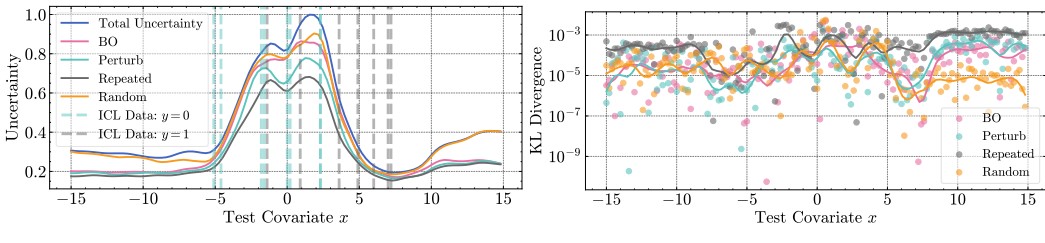

(a) Effect of $\mathbf{Z}$ on Uncertainty Estimates

(b) KL Divergence of optimal selected $\mathbf{Z}$

Figure 11: $V_a$ across $\mathbf{Z}$ sampling methods (`Qwen2.5-7B`).

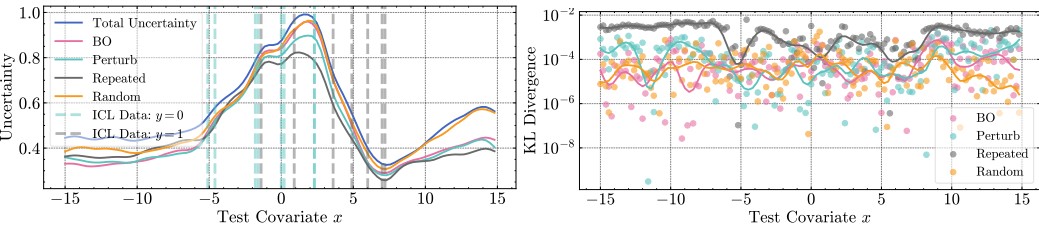

(a) Effect of $\mathbf{Z}$ on Uncertainty Estimates

(b) KL Divergence of optimal selected $\mathbf{Z}$

Figure 12: $V_a$ across $\mathbf{Z}$ sampling methods (`Llama-3.1-8B`).

Table 3: $V_a$ rank for different sampling methods

| MODELS | BAYESIAN OPTIMISATION | PERTURBATIONS | REPEATED TASK | RANDOM SAMPLING |
|---|---|---|---|---|
| QWEN2.5-7B | 2.93 | 2.01 | 1.29 | 3.77 |
| QWEN2.5-14B | 3.09 | 2.03 | 1.16 | 3.68 |
| LLAMA-3.1-8B | 2.41 | 1.92 | 2.09 | 3.57 |

# D  Promoting Exchangeability in In-Context Learning

In this section, we expand upon the concept and definitions of exchangeability that we discuss in Section 2 and the methods to encourage exchangeability in Section 3.2.

## D.1  Definition of Exchangeability

A finite sequence of random variables $(X_i)_{i=1}^n$ is *exchangeable* if for any permutation $\rho : [n] \to [n]$,

$$(X_1, \ldots, X_n) \stackrel{d}{=} (X_{\rho(1)}, \ldots, X_{\rho(n)}), \tag{12}$$

(where $\stackrel{d}{=}$ refers to equal in distribution) [24]. Similarly, an infinite sequence $(X_n)_{n\geq 1}$ is *exchangeable* if for any finite permutation $\rho$, $(X_n)_{n\geq 1} \stackrel{d}{=} (X_{\rho(n)})_{n\geq 1}$. One of the most consequential results for exchangeable sequences is de Finetti's representation theorem which is stated (in measure-theoretic form) as follows:

Table 4: KL divergence rank for different sampling methods

| MODELS | BAYESIAN OPTIMISATION | PERTURBATIONS | REPEATED TASK | RANDOM SAMPLING |
|---|---|---|---|---|
| QWEN2.5-7B | 2.27 | 2.31 | 3.43 | 1.99 |
| QWEN2.5-14B | 1.98 | 2.27 | 3.51 | 3.51 |
| LLAMA-3.1-8B | 1.93 | 2.41 | 3.77 | 1.89 |

**Theorem D.1** (de Finetti's representation theorem). *Let $(X_n)_{n\geq 1}$ be an infinitely exchangeable sequence and denote $\mathbb{P}$ its probability law. Then, there exists a unique random distribution $\tilde{F}$ with law $\pi$ such that for all $n \geq 1$ and measurable sets $A_1, \ldots, A_n$,*

$$\mathbb{P}(X_1 \in A_1, \ldots, X_n \in A_n) = \int \prod_{i=1}^{n} F(A_i)\pi(dF). \tag{13}$$

Note that this standard measure-theoretic definition [24] differs from the one introduced in Section 2 where we have a supervised learning setting of covariate-label pairs $\{(\boldsymbol{x}_i, \mathbf{y}_i)\}_{i=1}^{n}$. The results that we use later in Section D.3 to encourage exchangeability are also in a measure-theoretic form and therefore, in the following section, we bridge the gap between the supervised learning setting and the measure-theoretic language.

## D.2 Bridging the Gap

So that we can consider the pair $(\boldsymbol{x}_i, \mathbf{y}_i)$ as the observations of a random sequence taking values in $\mathcal{X} \times \mathcal{Y}$, we make the modelling assumption that $(X_i)_{i\geq 1}$ is an i.i.d. sequence of random variables with law $Q$ and density $q$. Then, the random variable $Y_{n+1}$ is generated from the LLM given $X_{n+1}$ and $\{(X_i, Y_i)\}_{i=1}^{n}$. Denoting random variables $K_n = (X_n, Y_n)$ with realisations $\boldsymbol{k}_{1:n} = \{(\boldsymbol{x}_n, \mathbf{y}_n)\}_{i=1}^{n}$, and measurable sets $A$ and $B$ in the sigma algebras $\sigma(X)$ and $\sigma(Y)$ respectively, the following proposition justifies the definition of exchangeability that we use in Section 2.

**Proposition D.1** Let $(X_n)_{n\geq 1}$ be an i.i.d. sequence and $(K_n)_{n\geq 1}$ be an exchangeable sequence, where $K_n = (X_n, Y_n)$. Then there exists a random distribution $\tilde{F}$ with joint (random) density $f_{X,Y}$ and law $\pi$ such that the conditional density of $Y_{1:n}|X_{1:n}$ can be expressed as

$$p(\mathbf{y}_{1:n}|\boldsymbol{x}_{1:n}) = \int \prod_{i=1}^{n} f_{Y|X}(\mathbf{y}_i|\boldsymbol{x}_i)\pi(dF).$$

*Proof.* Since $(K_n)_{n\geq 1}$ is exchangeable, by de Finetti's we have

$$\mathbb{P}(K_1 \in (A_1, B_1), \ldots, K_n \in (A_n, B_n)) = \int \prod_{i=1}^{n} F(A_i, B_i)\pi(dF)$$

$$\Leftrightarrow \quad \mathbb{P}(X_1 \in A_1, Y_1 \in B_1, \ldots, X_n \in A_n, Y_n \in B_n) = \int \prod_{i=1}^{n} F(A_i, B_i)\pi(dF)$$

$$\Leftrightarrow \quad \mathbb{P}(X_1 \in A_1, Y_1 \in B_1, \ldots, X_n \in A_n, Y_n \in B_n) = \int \prod_{i=1}^{n} \int_{A_i} \int_{B_i} f_{Y|X}(\mathbf{y}_i|\boldsymbol{x}_i)f_X(\boldsymbol{x}_i)d\mathbf{y}_i d\boldsymbol{x}_i \pi(dF),$$

where $f_{X,Y}$ is the joint (random) density of the random distribution $\tilde{F}$ with (random) marginal $f_X$ and (random) conditional distribution $f_{Y|X}$. Now, letting $B_i = \mathcal{Y}$, we obtain the marginal probabilities as follows:

$$\mathbb{P}(X_1 \in A_1, \ldots, X_n \in A_n) = \int \prod_{i=1}^{n} \int_{A_i} \underbrace{\int_{\mathcal{Y}} f_{Y|X}(\mathbf{y}_i|\boldsymbol{x}_i)d\mathbf{y}_i}_{=1} f_X(\boldsymbol{x}_i)d\boldsymbol{x}_i \pi(dF)$$

$$\prod_{i=1}^{n} \mathbb{P}(X_i \in A_i) = \int \prod_{i=1}^{n} \int_{A_i} f_X(\boldsymbol{x}_i)d\boldsymbol{x}_i \pi(dF)$$

$$\prod_{i=1}^{n} \int_{A_i} q(\boldsymbol{x}_i)d\boldsymbol{x}_i = \int \prod_{i=1}^{n} \int_{A_i} f_X(\boldsymbol{x}_i)d\boldsymbol{x}_i \pi(dF), \tag{†}$$

where the second line follows from the independence of $X_i$. As (†) holds for any measurable set $A_i \in \sigma(X)$, we see that $f_X \overset{a.s}{=} q$ is a valid solution to (†). But by the uniqueness of the law $\pi$ in de Finetti's representation theorem, we can indeed conclude that $f_X \overset{a.s.}{=} q$. Therefore, substituting $f_X = q$ into the de Finetti's representation, the conditional probability density of $Y_{1:n}|X_{1:n}$ can be expressed as:

$$
\begin{aligned}
p(\mathbf{y}_{1:n}|\boldsymbol{x}_{1:n}) &= p(\boldsymbol{x}_{1:n}, \mathbf{y}_{1:n})/p(\boldsymbol{x}_{1:n}) \\
&= \int \prod_{i=1}^{n} f_{X,Y}(\boldsymbol{x}_i, \mathbf{y}_i)\pi(dF)/p(\boldsymbol{x}_{1:n}) \\
&= \int \prod_{i=1}^{n} f_{Y|X}(\mathbf{y}_i|\boldsymbol{x}_i)q(\boldsymbol{x}_i)\pi(dF)/p(\boldsymbol{x}_{1:n}) \\
&= \int \left( \prod_{i=1}^{n} f_{Y|X}(\mathbf{y}_i|\boldsymbol{x}_i) \right) \left( \prod_{i=1}^{n} q(\boldsymbol{x}_i) \right) \pi(dF)/p(\boldsymbol{x}_{1:n}) \\
&= \int \prod_{i=1}^{n} f_{Y|X}(\mathbf{y}_i|\boldsymbol{x}_i)\pi(dF) \cdot \underbrace{\left( \prod_{i=1}^{n} q(\boldsymbol{x}_i) \right)}_{=p(\boldsymbol{x}_{1:n})}/p(\boldsymbol{x}_{1:n}) \\
&= \int \prod_{i=1}^{n} f_{Y|X}(\mathbf{y}_i|\boldsymbol{x}_i)\pi(dF).
\end{aligned}
$$

$\square$

## D.3 Exchangeability from Predictive Rules

In our problem setting, we have the predictive rule for $(K_n)_{n\geq 1}$

$$
\begin{aligned}
P_n((A, B)|\boldsymbol{k}_{1:n}) &\equiv \mathbb{P}(K_{n+1} \in (A, B)|K_1 = \boldsymbol{k}_1, \ldots, K_n = \boldsymbol{k}_n) \\
&\equiv \mathbb{P}(X_{n+1} \in A, Y_{n+1} \in B|\boldsymbol{k}_1 \ldots, \boldsymbol{k}_n) \\
&\equiv \int_{(A,B)} \mathbb{P}(d(\boldsymbol{x}_{n+1}, \mathbf{y}_{n+1})|\boldsymbol{k}_{1:n}) \\
&\equiv \int_A \int_B \mathbb{P}(d\mathbf{y}_{n+1}|\boldsymbol{k}_{1:n}, \boldsymbol{x}_{n+1})\mathbb{P}(d\boldsymbol{x}_{n+1}|\boldsymbol{k}_{1:n}) \\
&\equiv \int_A \int_B \mathbb{P}(d\mathbf{y}_{n+1}|\boldsymbol{k}_{1:n}, \boldsymbol{x}_{n+1})Q(d\boldsymbol{x}_{n+1}),
\end{aligned}
$$

where the final equality follows from the independence of $X_{n+1}$ and probability $\mathbb{P}(\mathbf{y}_{n+1}|\boldsymbol{k}_1 \ldots, \boldsymbol{k}_n, \boldsymbol{x}_{n+1})$ is given by the LLM. The following theorem by Fortini, Ladelli and Regazzini [23, 24] gives necessary and sufficient conditions for an exchangeable sequence defined by a predictive rule.

**Theorem D.2** (Theorem 2.3 [24], Theorem 3.1 and Proposition 3.2 [23]). *Let $(K_n)_{n\geq 1} \sim \mathbb{P}$ be an infinite sequence of random variables with predictive rule $(P_n)_{n\geq 0}$. Then $(K_n)_{n\geq 1}$ is exchangeable if and only if, for every $n \geq 0$, the following conditions hold:*

i) *For every $C \in \sigma(K)$, $P_n(C|\boldsymbol{k}_{1:n})$ is a symmetric function of $x_1, \ldots, x_n$;*

ii) *The set function $(C, D) \to \int_C P_{n+1}(D|\boldsymbol{k}_{1:n+1})dP_n(\boldsymbol{k}_{n+1}|\boldsymbol{k}_{1:n})$ is symmetric in $C$ and $D$, where $C, D \in \sigma(K)$.*

**Permutation Ensembling** In our predictive rule, we approximately satisfy i) via the Monte Carlo approximation (10) as i) is equivalent to ensuring $\mathbb{P}(d\mathbf{y}_{n+1}|\boldsymbol{k}_{1:n}, \boldsymbol{x}_{n+1})$ is symmetric in $\boldsymbol{k}_{1:n}$.

**KL-Filtering** Condition ii) essentially requires that

$$
\mathbb{P}(K_{n+1} \in C, K_{n+2} \in D|\boldsymbol{k_{1:n}}) = \mathbb{P}(K_{n+1} \in D, K_{n+2} \in C|\boldsymbol{k_{1:n}}).
$$

The set function in ii) can be expressed as follows in terms of $(\boldsymbol{x}_{n+1}, \boldsymbol{x}_{n+2}, \mathbf{y}_{n+1}, \mathbf{y}_{n+2})$

$$\int_{A_{n+1},B_{n+1}} P_{n+1}((A_{n+2}, B_{n+2})|\boldsymbol{k}_{1:n+1})dP_n(\boldsymbol{k}_{n+1}|\boldsymbol{k}_{1:n})$$

$$= \int_{A_{n+1}}\int_{B_{n+1}}\int_{A_{n+2}}\int_{B_{n+2}} \mathbb{P}(d\mathbf{y}_{n+2}|\boldsymbol{k}_{1:n}\cup\{(\boldsymbol{x}_{n+1},\mathbf{y}_{n+1})\},\boldsymbol{x}_{n+2})Q(d\boldsymbol{x}_{n+2})\mathbb{P}(d\mathbf{y}_{n+1}|\boldsymbol{k}_{1:n},\boldsymbol{x}_{n+1})Q(d\boldsymbol{x}_{n+1})$$

$$= \int_{A_{n+1}}\int_{A_{n+2}}\int_{B_{n+1}}\int_{B_{n+2}} \mathbb{P}(d\mathbf{y}_{n+2}|\boldsymbol{k}_{1:n}\cup\{(\boldsymbol{x}_{n+1},\mathbf{y}_{n+1})\},\boldsymbol{x}_{n+2})\mathbb{P}(d\mathbf{y}_{n+1}|\boldsymbol{k}_{1:n},\boldsymbol{x}_{n+1})Q(d\boldsymbol{x}_{n+2})Q(d\boldsymbol{x}_{n+1}).$$

This expression is computationally infeasible to check but a necessary condition for this is

$$\mathbb{P}(K_{n+1}\in C|\boldsymbol{k_{1:n}}) = \mathbb{P}(K_{n+2}\in C|\boldsymbol{k_{1:n}}),$$

where we take $D = (\mathcal{X}, \mathcal{Y})$. This is equivalent to

$$\int_A\int_B \mathbb{P}(d(\boldsymbol{x}_{n+1},\mathbf{y}_{n+1})|\boldsymbol{k}_{1:n}) = \int_A\int_B \mathbb{P}(d(\boldsymbol{x}_{n+2},\mathbf{y}_{n+2})|\boldsymbol{k}_{1:n})$$

$$\int_A\int_B \mathbb{P}(d\mathbf{y}_{n+1}|\boldsymbol{k}_{1:n},\boldsymbol{x}_{n+1})Q(d\boldsymbol{x}_{n+1}) = \int_A\int_B \mathbb{P}(d\mathbf{y}_{n+2}|\boldsymbol{k}_{1:n},\boldsymbol{x}_{n+1})Q(d\boldsymbol{x}_{n+2}). \qquad (\dagger\dagger)$$

A sufficient condition for ($\dagger\dagger$) is the equality of the laws $\mathbb{P}(Y_{n+1}\in\cdot|\boldsymbol{k}_{1:n}, X_{n+1}=\boldsymbol{x}) = \mathbb{P}(Y_{n+2}\in \cdot|\boldsymbol{k}_{1:n}, X_{n+2}=\boldsymbol{x})$, thus motivating the KL filtering condition (11).

**Effect of Permutation.** In Figure 13, we plot the KL divergence from $p(\mathbf{y}^*|\boldsymbol{x}^*, \mathcal{D})$ to $p(\mathbf{y}^*|\boldsymbol{x}^*, \mathbf{U}, \mathbf{Z}^\dagger, \mathcal{D})$ (where $\mathbf{Z}^\dagger = \mathrm{argmin}_{\mathbf{Z}}V_a(\mathbf{y}^*|\boldsymbol{x}^*, \mathbf{Z}, \mathcal{D})$) when we permute and do not permute the in-context labels. We see that permuting the in-context labels results in lower KL divergences, which suggests the behaviour is more Bayesian.

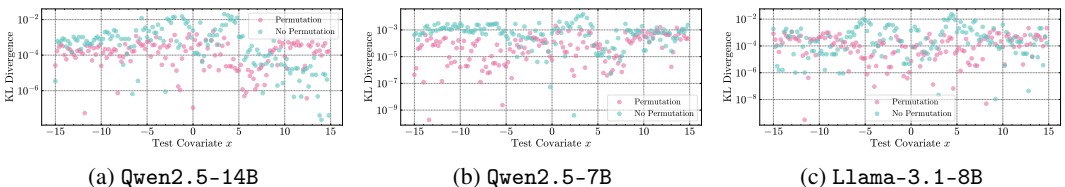

(a) Qwen2.5-14B          (b) Qwen2.5-7B          (c) Llama-3.1-8B

Figure 13: Permutation Ablation for Logistic Regression Dataset.

### D.4 Determining $\epsilon$ Threshold for KL-Filtering

The choice of $\epsilon$ controls the level of approximation permitted in the uncertainty decomposition method. A small $\epsilon$ ensures that the auxiliary data $\mathbf{Z}$ that we choose obey our Bayesian assumption but at the cost of rejecting more $\mathbf{Z}$ and obtaining a larger variational upper bound to the aleatoric uncertainty or variance. Furthermore, as shown in Figure 8, the range of KL values for the different auxiliary examples may vary when we vary $\boldsymbol{x}^*$. Therefore, to guarantee that we have enough valid auxiliary examples, we set $\epsilon$ as the $r^{\text{th}}$ smallest element in the set of KL divergences $\{\epsilon_j\}_{j=1}^m$ where $\epsilon_j := D_{\text{KL}}[p(\mathbf{y}^*|\boldsymbol{x}^*, \mathcal{D}), p(\mathbf{y}^*|\boldsymbol{x}^*, \boldsymbol{z}_j, \mathcal{D})]$. Therefore, we can control the strictness of the filtering by varying $r$, where a smaller $r$ gives a stricter decomposition.

# E Algorithms and Pseudocode

## E.1 Pseudocode for Variational Uncertainty Decomposition Algorithm

Algorithm 1 is pseudocode for multi-class classification problems and Algorithm 2 is the pseudocode for regression. They are similar in approach but vary during the marginalisation step: for classification, we can compute $p(\mathbf{y}^*|\boldsymbol{x}^*, \boldsymbol{u} = k, \boldsymbol{z}_j, \mathcal{D})$ for each class $k$, and directly compute the marginal distribution using the tower property. However, for regression, this is computationally infeasible so we use a Monte Carlo estimate for the conditional entropy $\mathbb{E}_{p(\boldsymbol{u}|\boldsymbol{z}_j, \mathcal{D})}[\mathbb{H}[p(\mathbf{y}^*|\boldsymbol{x}^*, \boldsymbol{u}, \boldsymbol{z}_j, \mathcal{D})]]]$, over different samples of $\boldsymbol{u}$. To obtain the marginal distribution, we bootstrap samples from the mixture of Gaussians $\{p(\mathbf{y}^*|\boldsymbol{x}^*, \mathcal{D} \cup \{\boldsymbol{z}_j, \boldsymbol{u}_t^{(j)}\})\}_{t=1}^{T}$ and fit a Gaussian to these samples (as described in Algorithm 5).

---

**Algorithm 1** Multi-Class Classification for Aleatoric Uncertainty Estimation

---

**Require:** Test input $\boldsymbol{x}^*$; ICL Dataset $\mathcal{D} = \{\boldsymbol{x}_i, \mathbf{y}_i\}_{i=1}^{n}$ where $\mathbf{y}_i \in [K]$
1: $p(\mathbf{y}^*|\boldsymbol{x}^*, \mathcal{D}) \leftarrow \text{CLASSDIST}(\boldsymbol{x}^*, \mathcal{D})$
2: $H_{\text{total}} \leftarrow \mathbb{H}[p(\mathbf{y}^*|\boldsymbol{x}^*, \mathcal{D})]$
3: **for** $j = 1, \ldots, m$ **do**
4:      $\boldsymbol{z}_j \leftarrow \text{NEWAUX}(\boldsymbol{x}^*, \boldsymbol{z}_{[1:j-1]})$                        {Get new auxiliary variable}
5:      $p(\boldsymbol{u}|\boldsymbol{z}_j, D) \leftarrow \text{CLASSDIST}(\boldsymbol{z}_j, \mathcal{D})$
6:      **for** $k = 1, \ldots, K$ **do**
7:          $p(\mathbf{y}^*|\boldsymbol{x}^*, \mathcal{D} \cup \{\boldsymbol{z}_j, k\}) \leftarrow \text{CLASSDIST}(\boldsymbol{x}^*, \mathcal{D} \cup \{\boldsymbol{z}_j, k\})$
8:          $H_{kt} \leftarrow \mathbb{H}[p(\mathbf{y}^*|\boldsymbol{x}^*, \mathcal{D} \cup \{\boldsymbol{z}_j, k\})]$
9:      **end for**
10:     $p(\mathbf{y}^*|\boldsymbol{x}^*, \boldsymbol{z}_j, \mathcal{D}) \leftarrow \sum_{k=1}^{K} p(\mathbf{y}^*|\boldsymbol{x}^*, \mathcal{D} \cup \{\boldsymbol{z}_j, k\}) \cdot p(\boldsymbol{u} = k|\boldsymbol{z}_j, \mathcal{D})$
11:     $H_j \leftarrow \sum_{k=1}^{K} H_{kt} \cdot p(\boldsymbol{u} = k|\boldsymbol{z}_j, \mathcal{D})$
12:     $\epsilon_j \leftarrow D_{\text{KL}}[p(\mathbf{y}^*|\boldsymbol{x}^*, \mathcal{D}) \parallel p(\mathbf{y}^*|\boldsymbol{x}^*, \boldsymbol{z}_j, \mathcal{D})]$
13: **end for**
14: Compute threshold $\epsilon$ (see Appendix D.4)
15: $V_a \leftarrow \min \left( \min(\{H_j : \epsilon_j < \epsilon\}), H_{\text{total}} \right)$
16: **return** $V_a$

---

---

**Algorithm 2** Regression for Aleatoric Uncertainty Estimation

---

**Require:** Test input $\boldsymbol{x}^*$; ICL Dataset $\mathcal{D} = \{\boldsymbol{x}_i, \mathbf{y}_i\}_{i=1}^{n}$ where $\mathbf{y}_i \in \mathbb{R}$
1: $p_{\mathcal{N}}(\mathbf{y}^*|\boldsymbol{x}^*, \mathcal{D}) \leftarrow \text{REGDIST}(\boldsymbol{x}^*, \mathcal{D})$
2: $H_{\text{total}} \leftarrow \mathbb{H}[\mathbf{p}(\mathbf{y}^*|\boldsymbol{x}^*, \mathcal{D})]$
3: **for** $j = 1, \ldots, m$ **do**
4:      $\boldsymbol{z}_j \leftarrow \text{NEWAUX}(\boldsymbol{x}^*, \boldsymbol{z}_{[1:j-1]})$                        {Get new auxiliary variable}
5:      $U^{(j)} \leftarrow \{\boldsymbol{u}_t^{(j)}\}_{t=1}^{T}$ where $\boldsymbol{u}_t^{(j)} \sim \text{REGDIST}(\boldsymbol{z}_j, \mathcal{D})$
6:      **for** $t = 1, \ldots, T$ **do**
7:          $p_{\mathcal{N}}(\mathbf{y}^*|\boldsymbol{x}^*, \mathcal{D} \cup \{\boldsymbol{z}_j, \boldsymbol{u}_t^{(j)}\}) \leftarrow \text{REGDIST}(\boldsymbol{x}^*, \mathcal{D} \cup \{\boldsymbol{z}_j, \boldsymbol{u}_t^{(j)}\})$
8:          $H_{jt} \leftarrow \mathbb{H}[p(\mathbf{y}^*|\boldsymbol{x}^*, \mathcal{D} \cup \{\boldsymbol{z}_j, \boldsymbol{u}_t^{(j)}\})]$
9:      **end for**
10:     $p_{\mathcal{N}}(\mathbf{y}^*|\boldsymbol{x}^*, \boldsymbol{z}_j, \mathcal{D}) \leftarrow \text{NORMAPPROX}(\{p_{\mathcal{N}}(\mathbf{y}^*|\boldsymbol{x}^*, \mathcal{D} \cup \{\boldsymbol{z}_j, \boldsymbol{u}_t^{(j)}\})\}_{t=1}^{T})$
11:     $H_j \leftarrow \frac{1}{T} \sum_t H_{jt}$
12:     $\epsilon_j \leftarrow D_{\text{KL}}[p_{\mathcal{N}}(\mathbf{y}^*|\boldsymbol{x}^*, \mathcal{D}) \parallel p_{\mathcal{N}}(\mathbf{y}^*|\boldsymbol{x}^*, \boldsymbol{z}_j, \mathcal{D})]$
13: **end for**
14: Compute threshold $\epsilon$ (see Appendix D.4)
15: $V_a \leftarrow \min \left( \min(\{H_j : \epsilon_j < \epsilon\}), H_{\text{total}} \right)$
16: **return** $V_a$

---

Note that these algorithms can also be extended to the decomposition of total variance by replacing the entropic uncertainty terms with the corresponding variance terms.

## E.2 Computing Approximate Posterior Predictive Distributions

**Classification**. Algorithm 3 describes the process of obtaining the logits for a predictive task $p(\mathbf{y}^*|\boldsymbol{x}^*, \mathcal{D})$ given in-context learning data $\mathcal{D} = \{(\boldsymbol{x}_i, \mathbf{y}_i)\}_{i=1}^n$ and the covariates of the predictive task $\boldsymbol{x}^*$. We permute the ICL data and take an average of the predictive distribution to obtain a Monte Carlo estimate of a conditional permutation-invariant distribution (which we discuss further in Appendix D. Furthermore, by the construction of the prompt, the we only need to obtain the logits for the first token that is generated, which remains constant with respect to the choice of LLM seed.

---

**Algorithm 3** Compute Permutation Invariant Classification Distribution $\mathbf{z}$ : CLASSDIST

---

**Require:** Test input $\boldsymbol{x}^*$; ICL Dataset $\mathcal{D} = \{\boldsymbol{x}_i, \mathbf{y}_i\}_{i=1}^n$ where $\mathbf{y}_i \in [K]$
1: **function** CLASSDIST($\boldsymbol{x}^*, \mathcal{D}$)
2:   **for** $l = 1, \ldots, L$ **do**
3:     $\sigma_l \sim S_K$
4:     $\boldsymbol{p}_y^{(l)} \leftarrow$ LLM(PROMPT($\boldsymbol{x}_{\sigma_l(1)}, \mathbf{y}_{\sigma_l(1)}, \ldots, \boldsymbol{x}_{\sigma_l(K)}, \mathbf{y}_{\sigma_l(K)}, \boldsymbol{x}^*$))    {Class prob. of next token}
5:   **end for**
6:   $\bar{\boldsymbol{p}}_y \leftarrow \frac{1}{L} \sum_l \boldsymbol{p}^{(l)}$
7:   **return** $\bar{\boldsymbol{p}}_y$

---

**Regression**. In Algorithm 4, we outline the procedure for constructing an approximate distribution for $p(\mathbf{y}^*|\boldsymbol{x}^*, \mathcal{D})$. Similarly to the classification case, we permute the ICL data. However, as $\mathbf{y}^*$ can take any value in $\mathbb{R}$, the tokenisation of $\mathbf{y}^*$ may require more than one token and as the logits of a token depend on the previous tokens generated, the logits of the tokens will vary with the choice of LLM seed. Standard approaches to approximate the distribution require a forward pass over every value that $\mathbf{y}^*$ takes [92] which is prohibitively expensive. Therefore, for each permutation, we sample a single $\mathbf{y}^*$ (varying the LLM seed for every permutation) and fit a normal distribution to these samples via moment matching (namely, estimating the mean and standard deviation of the sample and using these estimates as the parameters of a normal distribution).

**Variance Reduction**. To reduce the variance of the estimated mean and standard deviation, we use a trimmed mean, removing the top $k$ and bottom $k$ of our samples, and the interquartile range to estimate the mean and standard deviation respectively [107]. In our experiments, we set $k = 1$.

**Marginalisation**. In Algorithm 2, we are required to compute the marginal distribution $p_{\mathcal{N}}(\mathbf{y}^*|\boldsymbol{x}^*, \boldsymbol{z}_j, \mathcal{D})$ given the Gaussian distributions $\{p(\mathbf{y}^*|\boldsymbol{x}^*, \mathcal{D} \cup \{\boldsymbol{z}_j, \boldsymbol{u}_t^{(j)}\})\}_{t=1}^T$. We compute this marginal distribution by bootstrap sampling from the distributions $p(\mathbf{y}^*|\boldsymbol{x}^*, \mathcal{D} \cup \{\boldsymbol{z}_j, \boldsymbol{u}_t^{(j)}\})$ and fitting a Gaussian distribution to the bootstrap samples via moment matching. This procedure is outlined in Algorithm 5.

---

**Algorithm 4** Approximate Permutation Invariant Regression Distribution: REGDIST.

---

**Require:** Test input $\boldsymbol{x}^*$; ICL Dataset $\mathcal{D} = \{\boldsymbol{x}_i, \mathbf{y}_i\}_{i=1}^n$ where $\mathbf{y}_i \in \mathbb{R}$
1: **function** REGDIST($\boldsymbol{x}, \mathcal{D}$)
2:   **for** $l = 1, \ldots, L$ **do**
3:     $\sigma_l \sim S_K$
4:     $\mathbf{y}^{(l)} \leftarrow$ LLM(PROMPT($\boldsymbol{x}_{\sigma_l(1)}, \mathbf{y}_{\sigma_l(1)}, \ldots, \boldsymbol{x}_{\sigma_l(K)}, \mathbf{y}_{\sigma_l(K)}, \boldsymbol{x}^*$))    {Sample next prediction}
5:   **end for**
6:   $\mathbf{Y} \leftarrow \{\mathbf{y}^{(l)}\}_{l=1}^L$                                          {Trimming optional}
7:   **return** Normal$\big(\text{mean}(\mathbf{Y}), \text{std}(\mathbf{Y})\big)$

---

## E.3 Profiling View of VUD Algorithm

The memory requirements and speed of applying VUD to a particular prediction task depends on the specific choice of LLMs and the hardware on which it is deployed. Therefore, for a clearer outline of computational costs, we outline the number of API calls for each step of the algorithm as this is the

---
**Algorithm 5** Approximate Marginalisation of Mixture Distributions: NORMAPPROX.
---
**Require:** Distributions $\{p_t(\mathbf{y})\}_{t=1}^T$
  1: **function** NORMAPPROX $(\{p_t(\mathbf{y})\}_{t=1}^T)$
  2: **for** $r = 1, \ldots, R$ **do**
  3:     $t_r \sim \mathcal{U}\{1, T\}$                    {Uniform discrete distribution from 1 to $T$}
  4:     $\mathbf{y}_{\mathcal{B}}^{(r)} \sim p_{t_r}(\mathbf{y})$                    {Sample next prediction}
  5: **end for**
  6: $\mathbf{Y}_{\mathcal{B}} \leftarrow \{\mathbf{y}_{\mathcal{B}}^{(r)}\}_{R=1}^L$
  7: **return** $\text{Normal}\big(\text{mean}(\mathbf{Y}_{\mathcal{B}}), \text{std}(\mathbf{Y}_{\mathcal{B}})\big)$
---

most significant bottleneck in the algorithm. In Table 5 and 6, we provide a profiling view for VUD in a classification task and a regression task respectively.

Table 5: Profiling View for Classification Task with $k$ classes using $n$ auxiliary data, and $L$ permutations per distribution.

| Function | LLM Calls Per Function | Num Function Calls |
|---|---|---|
| $p(\mathbf{y}^*\|\boldsymbol{x}^*, \mathcal{D})$ | $L$ | 1 |
| $p(\mathbf{U}\|\mathbf{Z}_i, \mathcal{D})$ | $L$ | $n$ |
| $p(\mathbf{y}^*\|\boldsymbol{x}^*, \mathbf{U} = u, \mathcal{D})$ | $L$ | $nk$ |

Table 6: Profiling View for Regression Task with $n$ auxiliary data, and $L$ samples to evaluate each distribution with $m$ discarded samples (due to trimming of mean in Algorithm 4).

| Function | LLM Calls Per Function | Num Function Calls |
|---|---|---|
| $p(\mathbf{y}^*\|\boldsymbol{x}^*, \mathcal{D})$ | $L + m$ | 1 |
| $p(\mathbf{U}\|\mathbf{Z}_i, \mathcal{D})$ | $L + m$ | $n$ |
| $p(\mathbf{y}^*\|\boldsymbol{x}^*, \mathbf{U} = u, \mathcal{D})$ | $L + m$ | $nL$ |

# F  Further Related Work

**Bayesian Interpretations of In-Context Learning**. Works in recent years [113, 84, 74] suggested that the behaviour of transformers during in-context learning emulates Bayesian inference. In our work, this Bayesian behaviour of ICL is a key assumption that is necessary for the validity of the variational uncertainty decomposition algorithm. However, there is also evidence to suggest that this Bayesian behaviour is only approximate during long-term generation in LLMs, invalidating the Bayesian assumption [20, 61]. In light of these previous works, our innovation lies in the attempt to promote permutation-invariant generation and filter non-Bayesian generation from auxiliary data to maintain the Bayesian assumption that we make.

**Permutation Invariance and Exchangeability in LLMs**. The generation in language models is dependent on the position of tokens [62, 121]. This is a clear violation of exchangeability, which is necessary for the application of de Finetti's theorem. [119] assumes the exchangeability of LLM generation to apply de Finetti which allows for the estimation of the topic distributions from LLMs. However, they do not apply permutations during ICL to the context. [118] discusses the importance of exchangeability for quantifying uncertainty in ICL. They investigate methods to promote permutation invariance during pre-training and fine-tuning or architectural modifications to the transformer through causal masking. Whilst they suggest using permuted data as a data augmentation technique during training, our permutation invariant conditional generation is purely applied during inference. Our approach incurs a greater cost during inference time but does not require fine-tuning of the LLM.

**Martingale Posteriors**. The Martingale posterior [20, 22, 53] construct a generalised notion of posterior distribution by the following steps: (1) defining a sequence of predictive distributions $\{p^n(\mathbf{y}^*|\boldsymbol{x}^*, \{(\boldsymbol{x}_i, \mathbf{y}_i)\}_{i=1}^n)\}$ for all $n \geq 1$, (2) sequentially generating $\mathbf{y}_j \sim p^j(\mathbf{y}_j|\boldsymbol{x}_j, \{(\boldsymbol{x}_i, \mathbf{y}_i)\}_{i<j})$ for $j = n + 1, ..., N$ with $N >> n$, and (3) computing a *proxy latent parameter* $\psi =$

$g(\{(\boldsymbol{x}_i, \mathbf{y}_i)\}_{i=1}^n \cup \{(\boldsymbol{x}_j, \mathbf{y}_j)\}_{j=n+1}^N)$ via some function $g$. Technically, this defines the following form of Martingale posterior ($\mathcal{D} = \{(\boldsymbol{x}_i, \mathbf{y}_i)\}_{i=1}^n$):

$$q^N(\psi|\mathcal{D}, \{\boldsymbol{x}_j\}_{j=n+1}^N) = \int \delta(\psi = g(\{(\boldsymbol{x}_i, \mathbf{y}_i)\}_{i=1}^n \cup \{(\boldsymbol{x}_j, \mathbf{y}_j)\}_{j=n+1}^N))$$

$$\times \prod_{j=n+1}^N p^j(\mathbf{y}_j|\boldsymbol{x}_j, \{(\boldsymbol{x}_i, \mathbf{y}_i)\}_{i<j})d\mathbf{y}_{n+1:N}.$$

If a *proxy likelihood model* $q(\mathbf{y}^*|\boldsymbol{x}^*, \psi)$ is further specified, then the predictive Martingale posterior can be defined as [53]

$$q^N(\mathbf{y}^*|\boldsymbol{x}^*, \mathcal{D}, \{\boldsymbol{x}_j\}_{j=n+1}^N) = \int q(\mathbf{y}^*|\boldsymbol{x}^*, \psi)q^N(\psi|\mathcal{D}, \{\boldsymbol{x}_j\}_{j=n+1}^N)d\psi. \tag{14}$$

Therefore an uncertainty decomposition (as presented in Section 2) by conditioning on the proxy latent parameter $\psi$ is plausible. We can compute the "Martingale version" of total uncertainty as $\mathbb{H}[q^N(\mathbf{y}^*|\boldsymbol{x}^*, \mathcal{D}, \{\boldsymbol{x}_j\}_{j=n+1}^N)]$ and the aleatoric uncertainty as $\mathbb{E}_{q^N(\psi|\mathcal{D}, \{\boldsymbol{x}_j\}_{j=n+1}^N)}[\mathbb{H}[q(\mathbf{y}^*|\boldsymbol{x}^*, \psi)]]$. Epistemic uncertainty can then be obtained via simple subtraction arithmetic.

The (predictive) Martingale posterior generalises conventional (predictive) Bayesian posterior as it does not require $\{p^n(\mathbf{y}^*|\boldsymbol{x}^*, \{(\boldsymbol{x}_i, \mathbf{y}_i)\}_{i=1}^n)\}$ to be consistent and correspond to the probability of an exchangeable sequence; instead it requires convergence properties of the $\{p^n(\mathbf{y}^*|\boldsymbol{x}^*, \{(\boldsymbol{x}_i, \mathbf{y}_i)\}_{i=1}^n)\}$ distributions and the $g$ function when $N \rightarrow \infty$, where we refer to [22] for details. In practice, to obtain robust estimations of Martingale posteriors, $N$ is often substantially larger than $n$, incurring significant computational cost, and the computation of $\{\mathbf{y}_j\}_{j=n+1}^N$ samples cannot be parallelised.

To make a critical comparison to our proposed concept of variational uncertainty decomposition, we note that in general Martingale posterior is also different from the conventional Bayesian posterior, even when there exists an exchangeable model such that $p^n(\mathbf{y}^*|\boldsymbol{x}^*, \{(\boldsymbol{x}_i, \mathbf{y}_i)\}_{i=1}^n) = p(\mathbf{y}^*|\boldsymbol{x}^*, \{(\boldsymbol{x}_i, \mathbf{y}_i)\}_{i=1}^n)$ for all $n \geq 1$. The key reason is because the corresponding Bayesian model $p(\mathbf{y}|\boldsymbol{x}, \theta)p(\theta)$ is *implicitly* defined via de Finetti's theorem applied to $p(\mathbf{y}^*|\boldsymbol{x}^*, \{(\boldsymbol{x}_i, \mathbf{y}_i)\}_{i=1}^n)$, meaning that its latent parameter $\theta$ is an "unknown unknown", i.e., the format of $\theta$ (e.g., dimensionality, value domain, etc) cannot be explicitly specified. Hence in general $\psi$ and $\theta$ are two different random variables in different domains (and thus the name "proxy" for $\psi$ in our terminology). Consequently, the uncertainty decomposition results based on $\psi$ are no longer faithful directly to the implicit Bayesian model $p(\mathbf{y}|\boldsymbol{x}, \theta)p(\theta)$, and their estimation gaps, when referencing to the implicit Bayesian model's uncertainties $U_a$ and $U_e$, are yet to be established. On the contrary, our proposed variational estimators $V_a$ and $V_e$ are faithful bounds to $U_a$ and $U_e$, respectively, and we have identified the exact mathematical expression of the estimation gap in Section 3.1, which can be interpreted as residual information gain and/or remaining disagreement in fantasy.

**Uncertainty Quantification for LLMs**. Reliable and robust uncertainty quantification is an area of growing importance in the field of language models [96, 1]. A common approach, which we employ in this paper, is using token-level probabilities [43, 60, 19] by analysing the probabilities of the tokens generated by a language model. These probabilities can be further calibrated by adapting standard methods for uncertainty quantification in deep learning such as temperature scaling [29, 112, 13, 111, 110], focal loss training [73, 112], and conformal prediction [95, 117]. Alternatively, careful prompting can elicit qualitative or quantitative verbalisations of the uncertainty in a statement made by the language model [8, 58, 68, 101]. In situations where an LLM generates open-ended answers to a question, token-level methods struggle to accurately capture the uncertainty of a response as it is possible to generate diverse responses in natural language that are semantically equivalent or similar. To address this, semantic similarity methods cluster similar responses together and report the combined uncertainty from each cluster [3, 50, 59]. Recent works have also taken a mechanistic interpretability approach to uncertainty quantification by using probes to analyse the hidden states of the LLM to diagnose when a model is uncertain [48, 2].

**Uncertainty Decomposition for LLM In-Context Predictions**. Uncertainty decomposition for LLMs has also been explored in previous works; however, the definitions of aleatoric and epistemic uncertainty vary from the traditional definitions in prior Bayesian literature. [39] considers the aleatoric uncertainty of a response as the ambiguity in the input. Therefore, given a distribution of "clarifications" $q(\mathbf{C}|\boldsymbol{x}^*)$ for a particular prompt, the *epistemic* uncertainty is defined as the *mean* conditional uncertainty of a particular clarification $\mathbb{E}_{q(\mathbf{C}|\boldsymbol{x}^*)}[\mathbb{H}[\mathbf{y}^*|\boldsymbol{x}^* \oplus \mathbf{C}]]$. In contrast, we seek to find

the minimal conditional entropy given auxiliary data, which acts as an upper bound to the underlying Bayesian *conditional entropy*. Furthermore, the focus of [39] is primarily zero-shot and few-shot prediction, whereas we consider tasks where a training dataset is provided in context. Ling et al. [60] approaches uncertainty decomposition of in-context learning by also employing the interpretation that ICL performs Bayesian inference. However, they define epistemic uncertainty as the conditional entropy $\mathbb{E}_{p(\theta|\mathcal{D})}[H[\mathbf{y}^*|\boldsymbol{x}^*, \theta]]$ and aleatoric uncertainty as the mutual information $\mathbb{I}(\mathbf{y}^*; \theta|\boldsymbol{x}^*, \mathcal{D})$. Both [39] and [60] reverse the traditional definitions of Bayesian uncertainty decomposition [45] and therefore, we do not use these methods as baselines.

**Bayesian Approaches to Transformers**. In this work, we view in-context learning as implicit Bayesian inference. However, prior work has connected the transformer architecture with Bayesian inference more explicitly via Bayes-by-backprop approaches [93, 66, 11]. In particular, low-rank adaptation [115, 7, 79] has allowed for parameter-efficient avenues for Bayesian deep learning in transformers. Alternatively, neural processes have been integrated with transformers [77] to provide another approach to Bayesian uncertainty quantification in transformers. A connection between attention and sparse GP posterior mean is also established in [14], which further builds a deep Gaussian process with transformer-type architectures.

**Applications to In-Context Exploration**. Techniques used to quantify uncertainty in LLM predictions can be used to drive in-context exploration-exploitation tasks. In reinforcement learning and bandit tasks, efficient exploration algorithms such as Upper Confidence Bound [52, 6] and Thompson Sampling (TS) [82, 81, 94] require modelling the epistemic posterior distribution over possible outcomes either implicitly, through visitation counts, or explicitly, for example via ensembles. By modelling the epistemic uncertainty, the agent is able to reason about potential outcomes with uncertainty due to lack of data and explore in promising directions. Previous work that analyses the in-context exploration capabilities of LLMs includes [49], where the exploration capabilities of LLMs are compared to those of standard algorithms on small-scale tasks, and [70], which investigates the exploration capability of LLMs on natural language bandit tasks. The work in [78] further explores and benchmarks LLMs' abilities on a number of bandit tasks and offers ways to improve the efficiency of exploration by introducing algorithmic enhancements that better align LLMs with the exploration-exploitation task. This line of work focusing on bandits is complemented by [109], which extends the benchmarking to include multi-step tasks in addition to bandits. Finally, the work in [4] adapts the TS heuristic to the LLM setting, enabling LLM agents to tackle sequential decision-making tasks analogous to that of the full reinforcement learning setting. Uncertainty-aware exploration has also been used in active-learning settings to obtain smoother decision boundaries of LLMs by identifying the data points that will give smoother boundaries [120].

**Abstention**. The ability to defer the prediction of a language model is important in high-risk and sensitive applications of language models such as medical settings [28, 71]. In particular, abstention from answering questions is closely related to the domain of selective classification, where we learn a corresponding selection function $g : \mathcal{X} \rightarrow \{0, 1\}$ alongside the standard classifier $f : \mathcal{X} \rightarrow \mathcal{Y}$, where $g(x) = 0$ and $g(x) = 1$ results in the classifier prediction being 'rejected' and 'accepted' respectively [26, 91, 72]. The goal is to minimise the loss of accepted samples (risk) and maximise number of accepted samples (coverage). In our setting, we use the uncertainty estimate as a proxy for our selection function and use the ranked samples to determine the threshold of the score, thereby seeking to minimise the risk for a specified coverage.

**OOD Detection**. Detecting out-of-distribution (OOD) inputs is critical for real-world applications such as medical diagnosis and autonomous driving, where models can make confidently wrong predictions on inputs far from the training distribution. Foundational work demonstrated that softmax confidence often fails under distributional shift, establishing simple baselines for OOD detection in deep neural networks [34]. However, epistemic uncertainty has been shown to be useful in OOD and hallucination detection [110, 45]. This led to uncertainty-based methods which estimate epistemic uncertainty such as deep ensembles [51], where the uncertainty is measured through model diversity, and prior networks where distributional uncertainty is used in addition to epistemic uncertainty [65]. In NLP, pre-trained language models have been used for OOD detection [33] through non-Bayesian approaches such as contrastive learning [122], unsupervised detection with transformers [114], and conditional generation strategies to improve OOD discriminability [91]. Extensions to multimodal settings further explore OOD detection in vision-language tasks [69].

**Mutual Information Estimators**. The quantity of mutual information for which we provide a lower bound has many applications including Bayesian experimental design [89], independent component analysis [54], neuroscience [83] and causality [38]. However, mutual information between two variables is considered challenging to estimate [104] as it requires access to the joint distribution of the variables, which is often unavailable. Variational methods are a popular approach used to lower-bound mutual information [87], and in particular, MINE [9] and InfoNCE [80] are methods based on variational lower bounds to the mutual information. However, when estimating $\mathbb{I}[\mathbf{y}^*; \theta | \boldsymbol{x}^*, \mathcal{D}]$, these methods require access to samples from both random variables $\mathbf{y}^*$ and $\theta$, but in our problem setting, the latent parameter $\theta$ is implicitly defined and thus cannot be sampled. Still, our approach provides a variational lower bound to the conditional mutual information quantity in this challenging setting, where our innovation sidesteps the access requirement of the $\theta$ variable by constructing optimisable probes via a Markov chain $\mathbf{y}^* \leftarrow \theta \rightarrow \mathbf{U}$, enabling data processing inequality arguments and allowing lower-bound optimisation similar to MINE.

# G   Experiments

## G.1   Code Implementation

The following delineates the foundation of our experiments:

- Codebase: Python & PyTorch
- CPU: AMD EPYC 7443P
- GPU: NVIDIA A6000 48GB

We leverage Qwen2.5-14B/14B-Instruct/7B [88] and Llama-3.1-8B [102] in our experiments. The following delineates the configurations of our LLM.

- Temperature: 1.0
- Log Probs: 10
- Max Tokens: 10 (Qwen2.5-14B/7B and Llama-3.1-8B), 512 (Qwen2.5-14B-Instruct)

## G.2   Further Baselines

In this section, we include further comparisons to Martingale posterior distributions [22], an alternative uncertainty decomposition method for implicitly defined Bayesian models on an exchangeable sequence. However, one of the disadvantages of the Martingale posterior method is that we need to make distributional assumptions on the form of the proxy likelihood $q(\mathbf{y}|\boldsymbol{x}, \psi)$ (see Appendix F for further discussion). This can become particularly restrictive as the choice of likelihood model can greatly impact the estimated total uncertainty as we show in the following Figures 14-16 for the "Logistic Regression" Dataset and Figures 17-28 for the "Moons 1" Dataset and the Table 7 and 8 (further details on these datasets can be found in Appendix G.3).

As both datasets are (binary) classification problems, we consider four proxy likelihoods of the form $q(\mathbf{y}|\boldsymbol{x}, \psi) \propto p_\psi(\boldsymbol{x})^y (1 - p_\psi(\boldsymbol{x}))^y$, which we denote as 'linear', 'quadratic', 'cubic', and 'kernel':

- **Linear**: $p_\psi(\boldsymbol{x}) = \sigma(\psi_0 + \sum_i \psi_i x_i)$ where $\sigma$ is the standard logistic function (sigmoid). This is the probability distribution that is used to generate the dataset but may not reflect the true internal likelihood of the LLM. Here, $\psi \in \mathbb{R}^{d+1}$ (where $d$ is the dimension of $\boldsymbol{x}$).
- **Quadratic**: $p_\psi(\boldsymbol{x}) = \sigma(\psi_0 + \sum_i \psi_i x_i + \sum_{i \leq j} \psi_{ij} x_i x_j)$ where $\sigma$ is the standard logistic function. Here, $\psi \in \mathbb{R}^{\frac{(d+1)(d+2)}{2}}$.
- **Cubic**: $p_\psi(\boldsymbol{x}) = \sigma(\psi_0 + \sum_i \psi_i x_i + \sum_{i \leq j} \psi_{ij} x_i x_j + \sum_{i \leq j \leq k} \psi_{ijk} x_i x_j x_k)$ where $\sigma$ is the standard logistic function. Here, $\psi \in \mathbb{R}^{\frac{(d+1)(d+2)(d+3)}{6}}$.
- **Kernel**: $p_\psi(\boldsymbol{x}) = \sigma(\psi_A + \psi_B f_\psi(\boldsymbol{x}))$, where $\sigma$ is the logistic regression function, $\psi_A$ and $\psi_B$ are parameters to scale the logits, and $f_\psi(\boldsymbol{x})$ are the logits. The logits $f_\psi(\boldsymbol{x})$ are of the form
$$f_\psi(\boldsymbol{x}) = \psi_0 + \sum_i y_i \psi_i k(\boldsymbol{x}_i, \boldsymbol{x}),$$

where $\{(x_i, y_i)\}_i$ is the training data set of initial examples and the generated, and the kernel $k$ is the RBF kernel. To obtain the estimate for the proxy latent parameter $\psi$, we follow the Platt-scaling method [86] which is implemented in the `scikit-learn` package `SVC` [85]. Here, $\psi \in \mathbb{R}^{N+3}$ (where $N$ is the size of the combined initial training examples and the generated sample path from the LLM).

To sequentially generate the next sample in the sequence $(\boldsymbol{x}_j, \mathbf{y}_j) \sim p^j((\boldsymbol{x}_j, \mathbf{y}_j)|\{(\boldsymbol{x}_i, \mathbf{y}_i)\}_{i<j})$, we permute the order of $\{(\boldsymbol{x}_i, \mathbf{y}_i)\}_{i<j}$ in the prompt for the LLM. This technique of permuting the observations in the sample path is used in previous work in the context of Martingale posteriors and LLM [20] and emulates the permutation-invariant sampling approach that we use for our VUD method. However, in our method, we sample the posterior for multiple permutations and compute an average to obtain our estimate for $p(\mathbf{y}^*|\boldsymbol{x}^*, \mathcal{D})$, whereas at each step in the sample path for the Martingale posterior, we only perform one permutation.

Table 7: L2 Distance between total uncertainty given by Martingale posterior distribution and empirically observed total uncertainty. Logistic Regression Dataset.

| LIKELIHOOD MODEL | QWEN2.5-14B | QWEN2.5-7B | LLAMA-3.1-8B |
|---|---|---|---|
| LINEAR FEATURES | 1.347 | 1.096 | 1.064 |
| QUADRATIC FEATURES | 3.624 | 4.334 | 1.381 |
| CUBIC FEATURES | 1.000 | 0.873 | 0.589 |
| KERNEL-BASED | 7.823 | 9.229 | 4.731 |

Table 8: L2 Distance between total uncertainty given by Martingale posterior distribution and empirically observed total uncertainty. Moons Dataset.

| LIKELIHOOD MODEL | QWEN2.5-14B | QWEN2.5-7B | LLAMA-3.1-8B |
|---|---|---|---|
| LINEAR FEATURES | 2.379 | 1.789 | 1.515 |
| QUADRATIC FEATURES | 2.796 | 2.781 | 2.819 |
| CUBIC FEATURES | 2.697 | 2.254 | 2.781 |
| KERNEL-BASED | 1.530 | 1.213 | 1.254 |

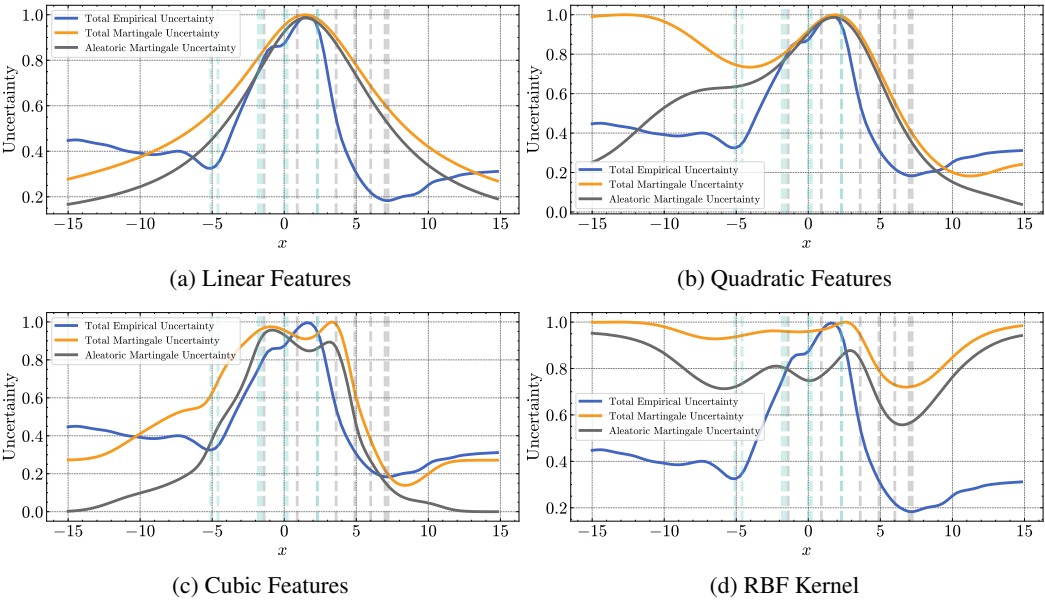

(a) Linear Features

(b) Quadratic Features

(c) Cubic Features

(d) RBF Kernel

Figure 14: Martingale Posterioir Uncertainty Decompositions for Logistic Regression (`Qwen2.5-14B`)

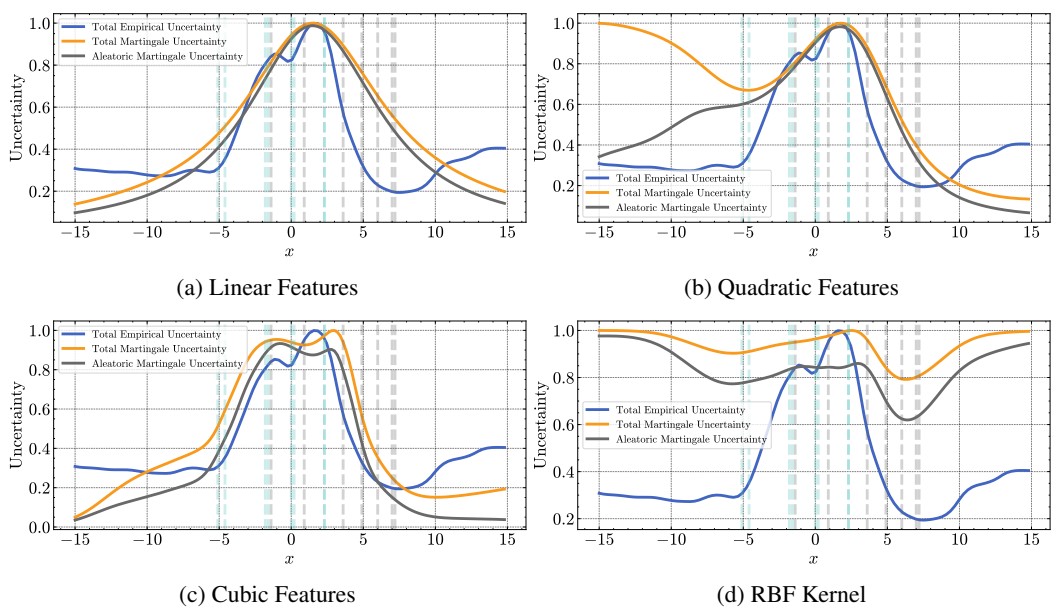

(a) Linear Features

(b) Quadratic Features

(c) Cubic Features

(d) RBF Kernel

Figure 15: Martingale Posterioir Uncertainty Decompositions for Logistic Regression (`Qwen2.5-7B`)

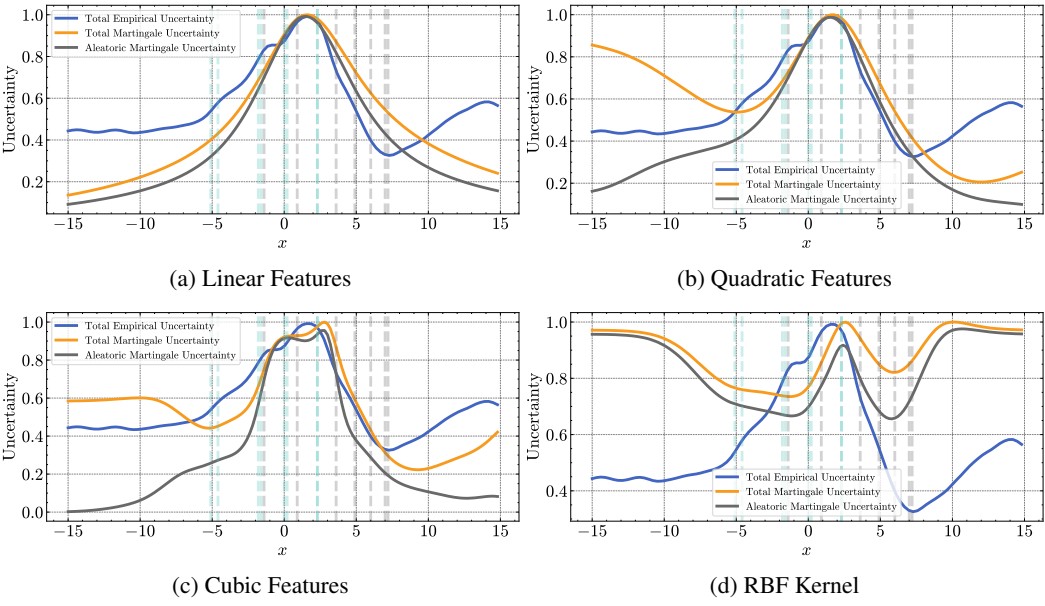

(a) Linear Features

(b) Quadratic Features

(c) Cubic Features

(d) RBF Kernel

Figure 16: Martingale Posterioir Uncertainty Decompositions for Logistic Regression (`Llama-3.1-8B`)

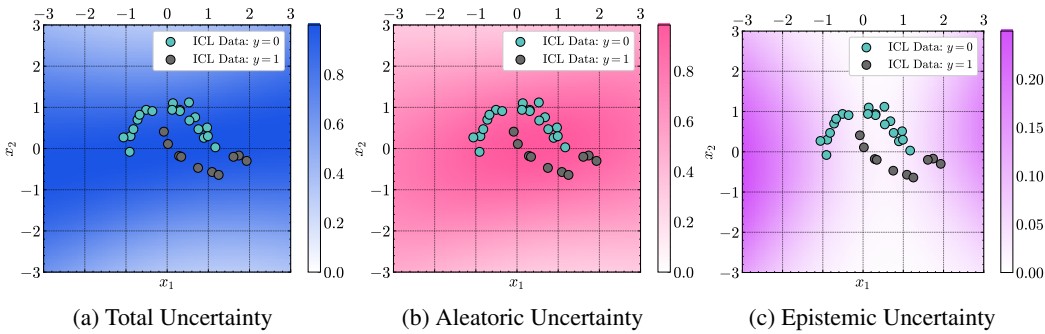

Figure 17: Martingale Posterior Uncertainty Decomposition for "Moons 1" Dataset - Linear Features (`Qwen2.5-14B`).

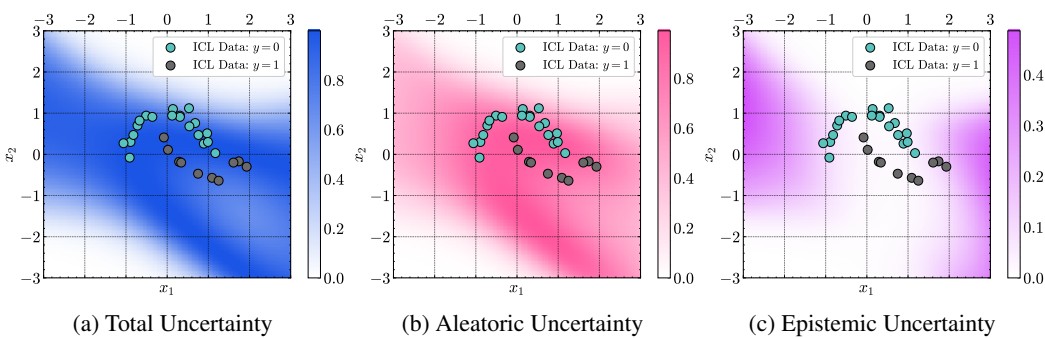

Figure 18: Martingale Posterior Uncertainty Decomposition for "Moons 1" Dataset - Quadratic Features (`Qwen2.5-14B`).

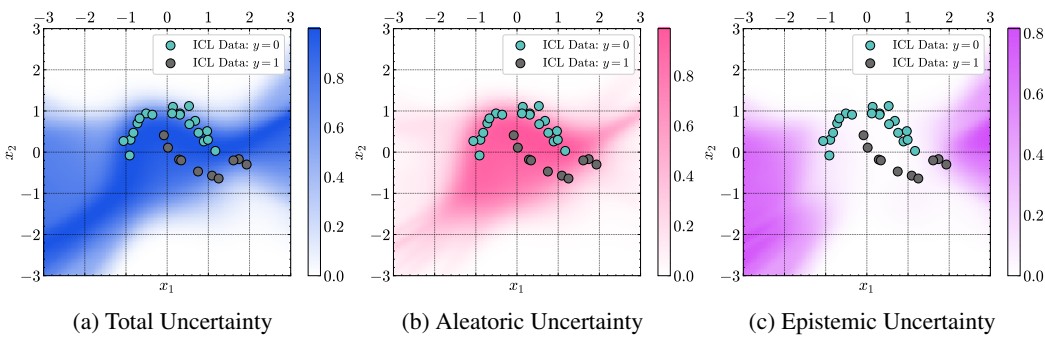

Figure 19: Martingale Posterior Uncertainty Decomposition for "Moons 1" Dataset - Cubic Features (`Qwen2.5-14B`).

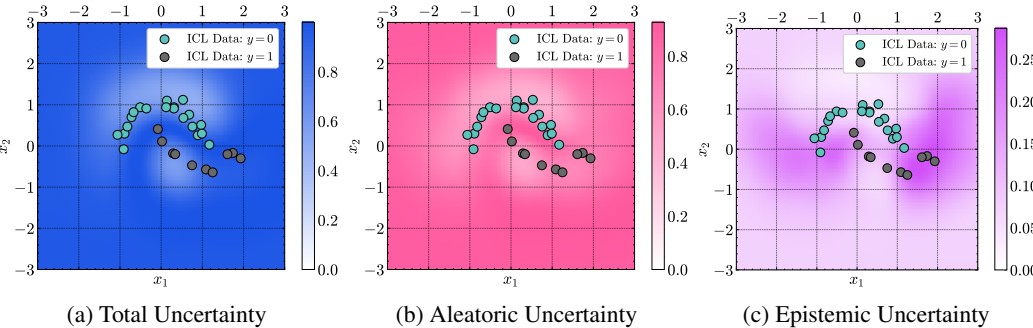

Figure 20: Martingale Posterior Uncertainty Decomposition for "Moons 1" Dataset - Kernel-Based Likelihood (`Qwen2.5-14B`).

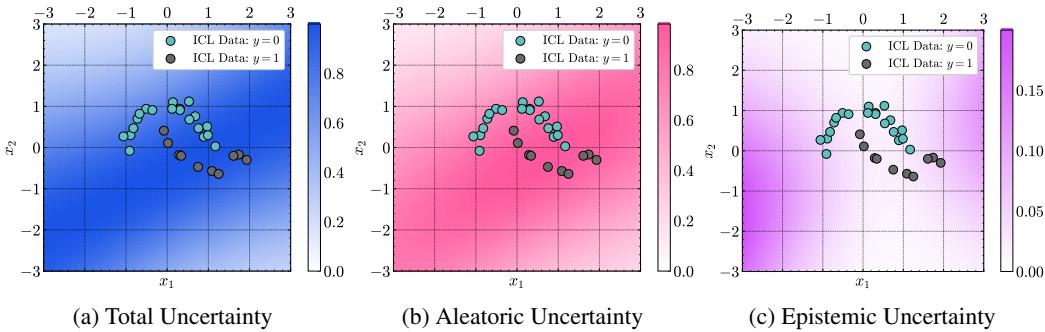

Figure 21: Martingale Posterior Uncertainty Decomposition for "Moons 1" Dataset - Linear Features (`Qwen2.5-7B`).

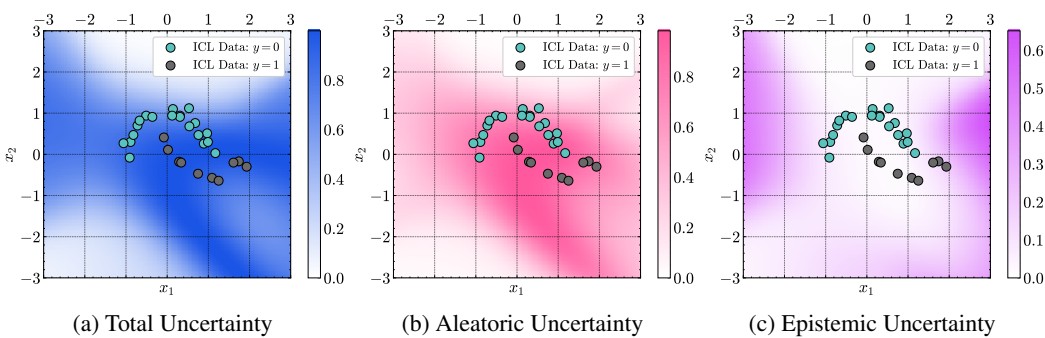

Figure 22: Martingale Posterior Uncertainty Decomposition for "Moons 1" Dataset - Quadratic Features (`Qwen2.5-7B`).

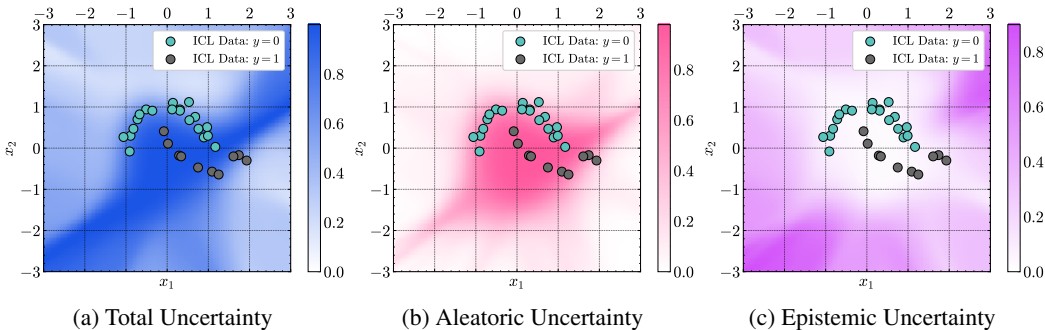

Figure 23: Martingale Posterior Uncertainty Decomposition for "Moons 1" Dataset - Cubic Features (`Qwen2.5-7B`).

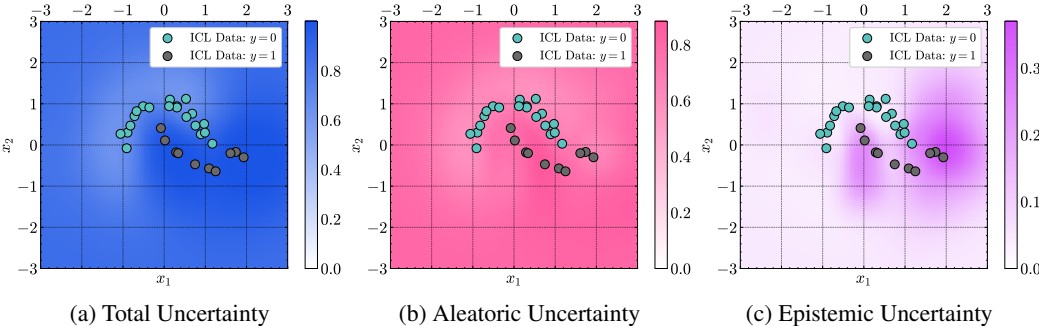

Figure 24: Martingale Posterior Uncertainty Decomposition for "Moons 1" Dataset - Kernel-Based Likelihood (`Qwen2.5-7B`).

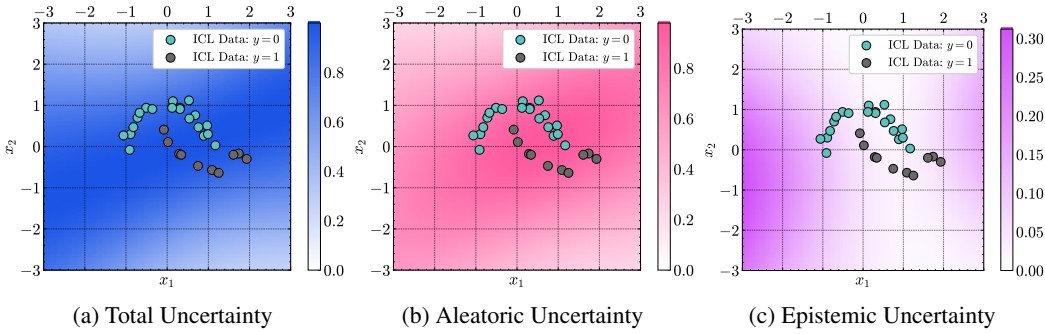

Figure 25: Martingale Posterior Uncertainty Decomposition for "Moons 1" Dataset - Linear Features (`Llama-3.1-8B`).

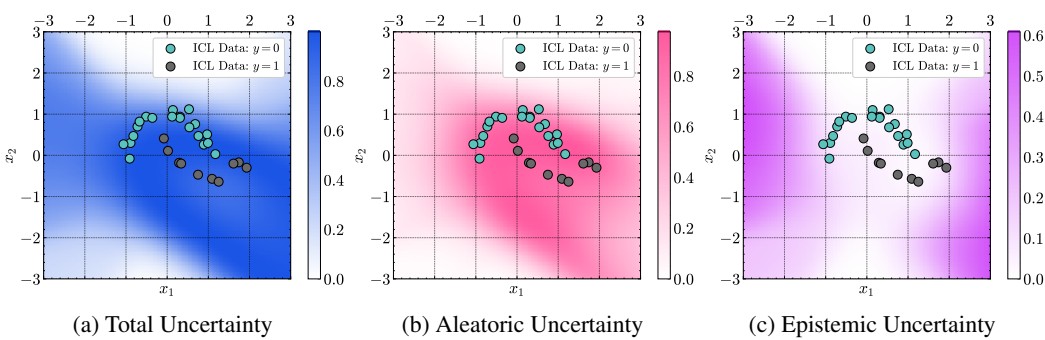

Figure 26: Martingale Posterior Uncertainty Decomposition for "Moons 1" Dataset - Quadratic Features (`Llama-3.1-8B`).

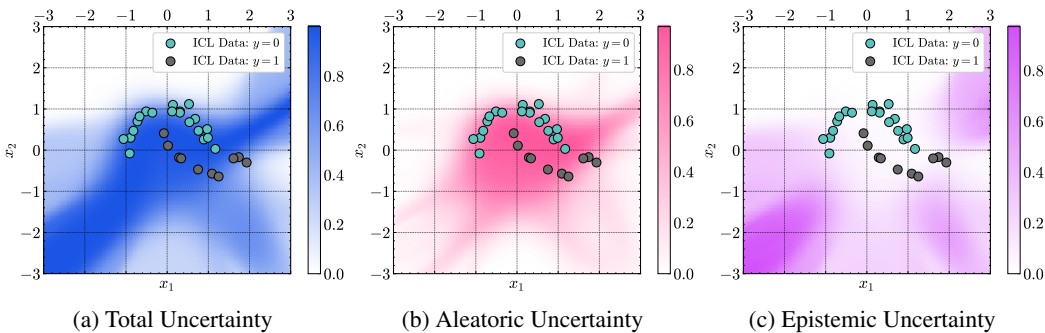

Figure 27: Martingale Posterior Uncertainty Decomposition for "Moons 1" Dataset - Cubic Features (`Llama-3.1-8B`).

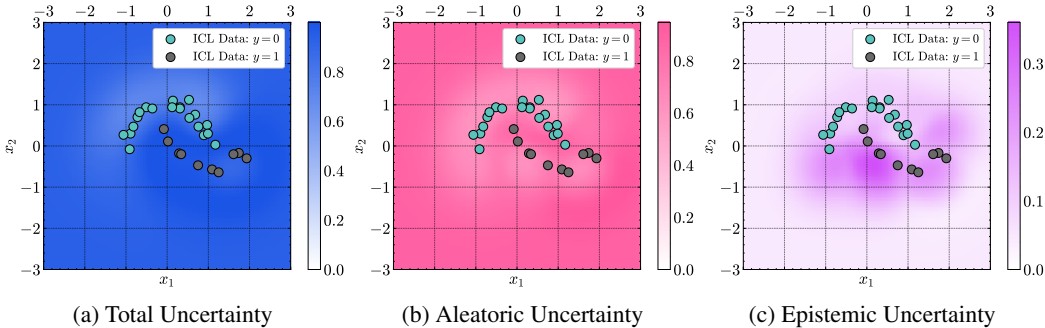

Figure 28: Martingale Posterior Uncertainty Decomposition for "Moons 1" Dataset - Kernel-Based Likelihood (`Llama-3.1-8B`).

### G.3 Synthetic Toy Experiments

We qualitatively evaluate the decompositions of the variational uncertainty decomposition algorithm on a variety of synthetic classification and regression settings. In this section, we give details on the ground-truth distributions used to create the synthetic datasets.

**Logistic Regression**. We consider a 1-D logistic regression problem with coefficient $\beta = 0.25$ and bias $\beta_0 = -0.5$. The covariates are generated from a Gaussian distribution with mean 1.5 and standard deviation 3. In our visualisations, we use Perturbations with 15 auxiliary data points and perturbation scale $\lambda = 0.1$ to decompose the uncertainty for the logistic regression task. In Figures 4a and 29, we plot the uncertainty decomposition for an ICL dataset of size $|\mathcal{D}| = 15$ and in Figure 30, we plot the decomposition for $|\mathcal{D}| = 75$. We plot $x^*$ values in the range $[-15, 15)$ with step size 0.2. In Figures 7, 31 and 32, we plot the epistemic and aleatoric uncertainties as the dataset size increases for in-distribution ($x = 0, 5$; solid lines) and out-of-distribution ($x^* = -15, -10, -5, 10, 15$; dotted lines) points. As the uncertainty at a given $x^*$ is dependent on the particular dataset, we average the uncertainty at $x^*$ over 10 datasets of the same size $d$ to obtain the estimate of the mean aleatoric uncertainty at $d$.

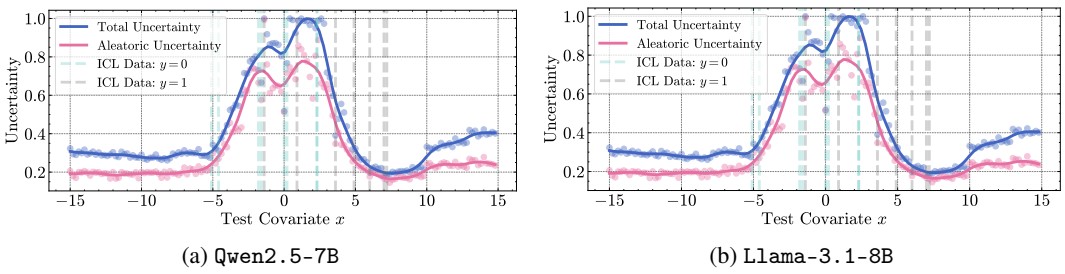

(a) `Qwen2.5-7B`  (b) `Llama-3.1-8B`

Figure 29: Uncertainty Decomposition for Logistic Regression $|\mathcal{D}| = 15$.

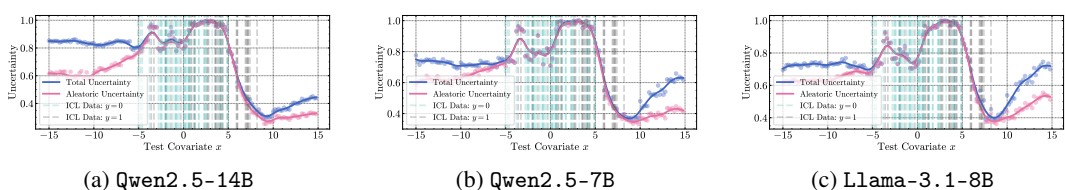

(a) `Qwen2.5-14B`  (b) `Qwen2.5-7B`  (c) `Llama-3.1-8B`

Figure 30: Logistic Regression with $|\mathcal{D}| = 75$.

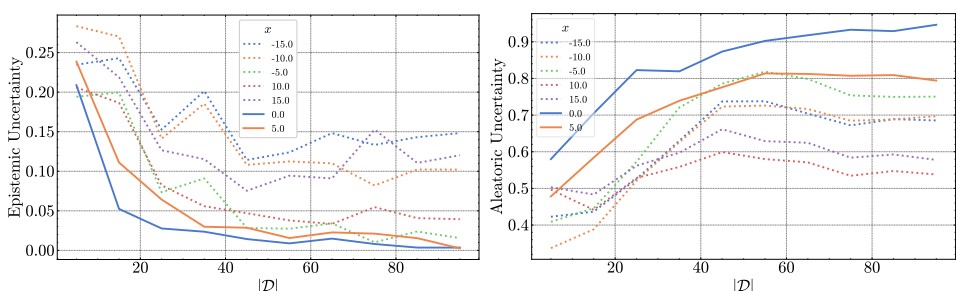

(a) Epistemic Uncertainty vs Size of Training Set  (b) Aleatoric Uncertainty vs Size of Training Set

Figure 31: Epistemic Uncertainty and Aleatoric Uncertainty vs Dataset Size (`Qwen2.5-7B`).

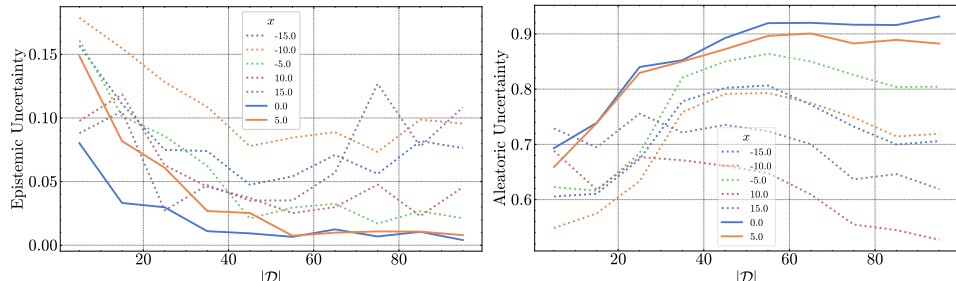

(a) Epistemic Uncertainty vs Size of Training Set  (b) Aleatoric Uncertainty vs Size of Training Set

Figure 32: Epistemic Uncertainty and Aleatoric Uncertainty vs Dataset Size (`Llama-3.1-8B`).

**Linear Regression**. We consider a 1-D linear regression problem with coefficient $\beta = -1$, bias $\beta_0 = 3$ and Gaussian noise with zero mean and standard deviation $\sigma = 2$. The covariates are generated from a Gaussian distribution with mean $1$ and standard deviation $2$. We use Perturbations with 5 auxiliary data points and perturbation scale $\lambda = 0.1$ to decompose the uncertainty for the logistic regression task. We reduce the number of auxiliary data points due to the increased computational cost of computing distributions for regression problems. In order to obtain smoother uncertainty computations, we average the uncertainties obtained over 3 sampled datasets of size $|\mathcal{D}| = 15$. We compute uncertainties for $x^*$ in the range $[-15,15)$ with step-size $0.2$ and plot the obtained decompositions for entropic uncertainty and variance in Figures 4b, 33 and 34. We also provide an example decomposition for the uncertainty and variance for a single seed for completion in Figures 35, 36 and 37.

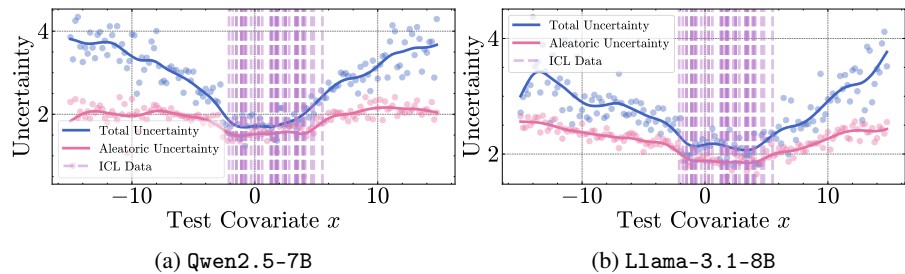

(a) `Qwen2.5-7B`          (b) `Llama-3.1-8B`

Figure 33: Linear Regression (Entropic) Uncertainty Decomposition.

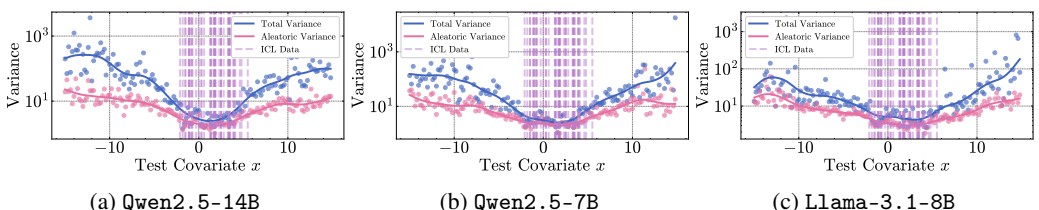

(a) `Qwen2.5-14B`          (b) `Qwen2.5-7B`          (c) `Llama-3.1-8B`

Figure 34: Linear Regression Variance Decomposition.

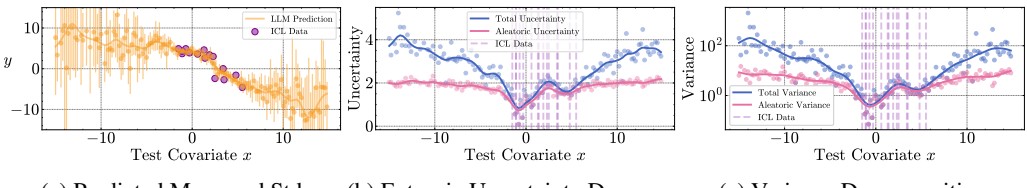

(a) Predicted Mean and Std   (b) Entropic Uncertainty Decomp.   (c) Variance Decomposition

Figure 35: Uncertainty Decompositions for Linear Regression (`Qwen2.5-14B`).

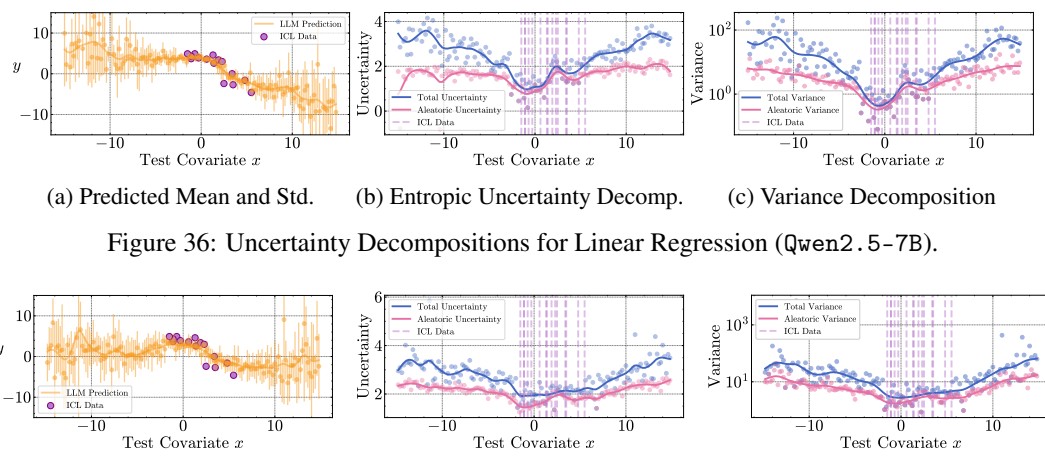

(a) Predicted Mean and Std.     (b) Entropic Uncertainty Decomp.     (c) Variance Decomposition

Figure 36: Uncertainty Decompositions for Linear Regression (`Qwen2.5-7B`).

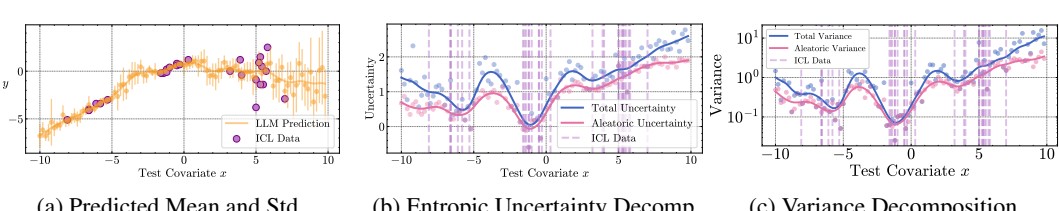

(a) Predicted Mean and Std.     (b) Entropic Uncertainty Decomp.     (c) Variance Decomposition

Figure 37: Uncertainty Decompositions for Linear Regression (`Llama-3.1-8B`).

**Heteroscedastic "Gaps" Regression.** We model the "gaps" as the combination of 3 linear regression datasets. The parameters of the 3 clusters are in Table 9. To generate the small in-context learning dataset, we sample from this combined dataset. In our visualisations, we use Perturbations with 5 auxiliary data points and perturbation scale $\lambda = 0.1$. We sample a single dataset of size $|\mathcal{D}| = 30$. We compute uncertainties for $x^*$ in range $[-15, 15)$ with step size $0.2$ and plot the obtained decompositions in Figures 38, 39 and 40.

Table 9: Heteroscedastic "Gaps" Dataset Parameters

| CLUSTER | DATASET SIZE | COEFFICIENT | BIAS | NOISE | $\mathbb{E}[x]$ | $\text{Var}[x]$ |
|---------|--------------|-------------|------|-------|------|---------|
| 1 | 50 | 0.75 | 1.0 | 0.1 | -7 | 0.75 |
| 2 | 50 | 0.75 | 1.0 | 0.1 | -1 | 0.75 |
| 3 | 100 | 0 | -0.5 | 2 | 5 | 1 |

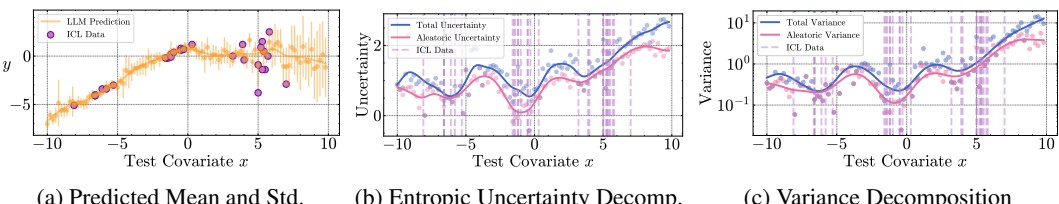

(a) Predicted Mean and Std.     (b) Entropic Uncertainty Decomp     (c) Variance Decomposition

Figure 38: Uncertainty Decomp. for Regression Tasks with Gaps in ICL Data. (`Qwen2.5-14B`)

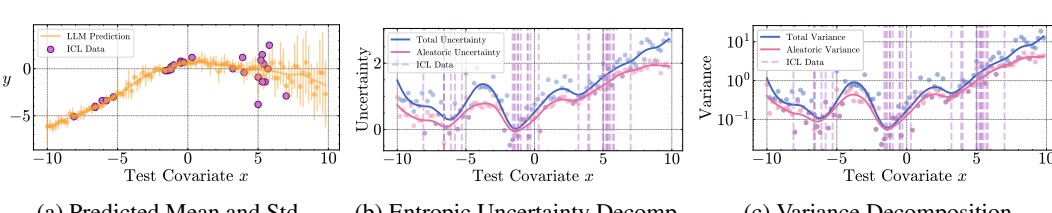

(a) Predicted Mean and Std.     (b) Entropic Uncertainty Decomp.     (c) Variance Decomposition

Figure 39: Uncertainty Decomp. for Regression Tasks with Gaps in ICL Data (`Qwen2.5-7B`).

(a) Predicted Mean and Std.     (b) Entropic Uncertainty Decomp.     (c) Variance Decomposition

Figure 40: Uncertainty Decomp. for Regression Tasks with Gaps in ICL Data (`Llama-3.1-8B`)

**Moons Dataset**. We use the `make_moons` two-moons dataset generator from `scikit-learn`. We set the noise parameter in the "Moons 1" and "Moons 2" datasets to $\sigma = 0.1$ and $\sigma = 0.4$ respectively. Figure 1 in the main text shows the decomposition for "Moons 1" dataset. We use Perturbations with 15 auxiliary data points and perturbation scale $\lambda = 0.1$. For the "Moons 1" dataset, we sample a single dataset of size $|\mathcal{D}| = 30$ and compute uncertainties for $x^*$ in the range $[-1.5, 2.5) \times [-1.5, 2.5)$ with step-size 0.2 for each interval. The decompositions are given in Figures 1, 41 and 42. For the "Moons 2" dataset, we sample a single dataset of size $|\mathcal{D}| = 30$ and compute uncertainties for $x^*$ in the range $[-3.0, 3.5) \times [-2.5, 3.0)$ with step-size 0.2 for each interval. The decompositions are given in Figures 43, 44 and 45.

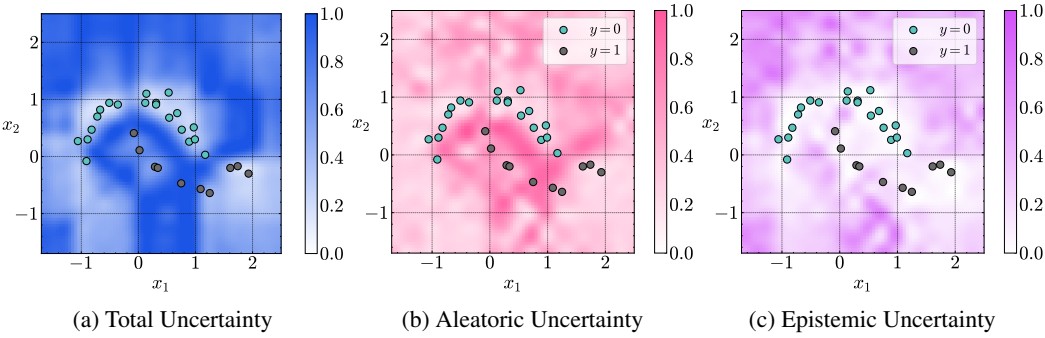

    (a) Total Uncertainty          (b) Aleatoric Uncertainty         (c) Epistemic Uncertainty

Figure 41: Uncertainty Decomposition for "Moons 1" Dataset (`Qwen2.5-7B`).

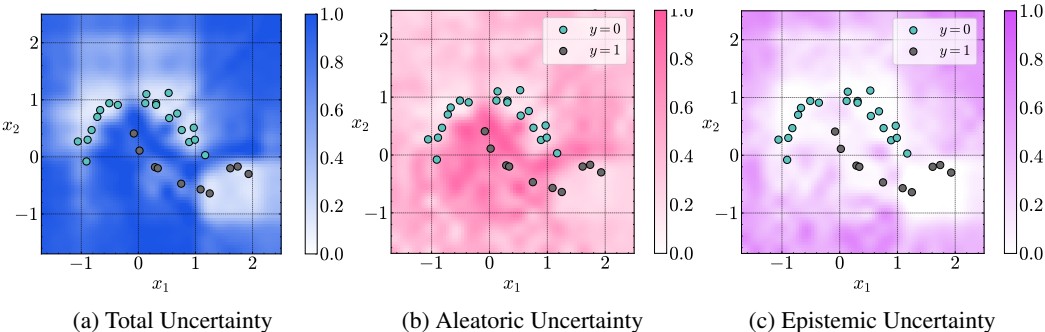

    (a) Total Uncertainty          (b) Aleatoric Uncertainty         (c) Epistemic Uncertainty

Figure 42: Uncertainty Decomposition for "Moons 1" Dataset (`Llama-3.1-8B`).

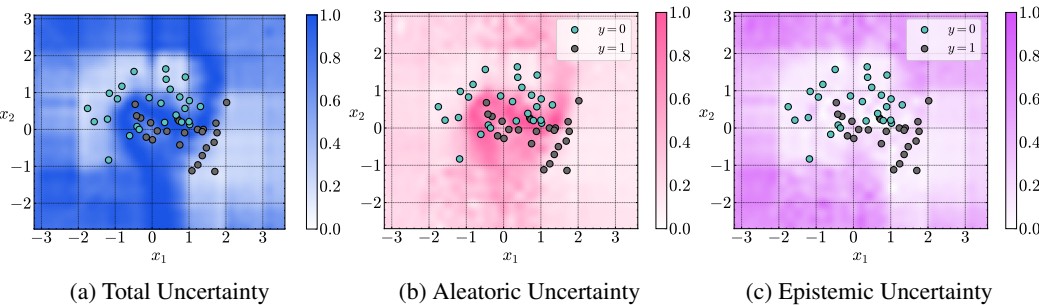

    (a) Total Uncertainty          (b) Aleatoric Uncertainty         (c) Epistemic Uncertainty

Figure 43: Uncertainty Decomposition for "Moons 2" Dataset (`Qwen2.5-14B`).

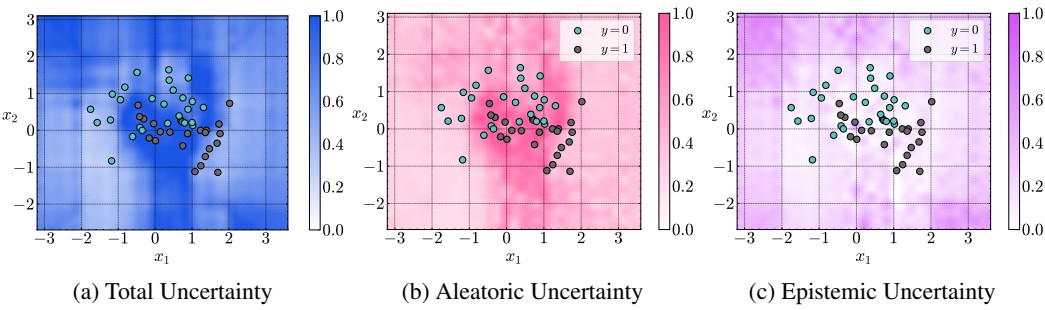

(a) Total Uncertainty      (b) Aleatoric Uncertainty      (c) Epistemic Uncertainty

Figure 44: Uncertainty Decomposition for "Moons 2" Dataset (`Qwen2.5-7B`).

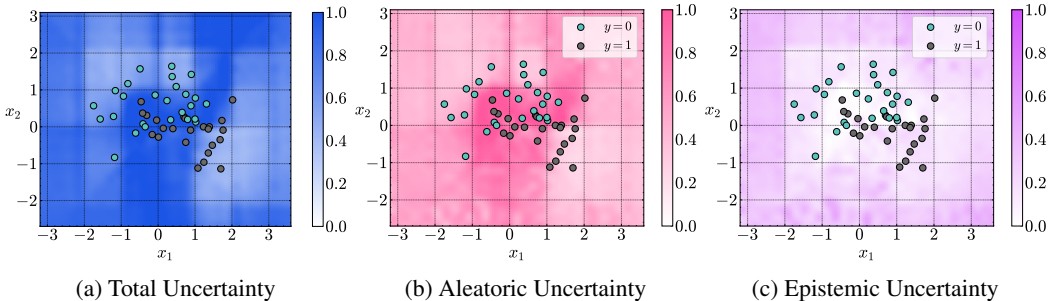

(a) Total Uncertainty      (b) Aleatoric Uncertainty      (c) Epistemic Uncertainty

Figure 45: Uncertainty Decomposition for "Moons 2" Dataset (`Llama-3.1-8B`).

**Spirals Dataset**. We use an *n*-arm spiral dataset generator to generate the spirals. We set the number of arms to 3 and noise to be 1.2. We also scale the covariate down by a factor of 4 so that all the points would appear in $[-4, 4] \times [-4, 4]$. Due to the complexity of this task, we sample a dataset of size $|\mathcal{D}| = 200$ and we compute uncertainties for $\boldsymbol{x}^*$ in the range of $[-4, 4) \times [-4, 4)$ with interval 0.1. To mitigate the cost of increases prompt size and the number of test data points, we use Repeated to obtain **Z**. The decomposition for Qwen2.5-14B is given in Figure 6. We provide decompositions for Qwen2.5-7B and Llama-3.1-8B are shown in Figure 46 and 47.

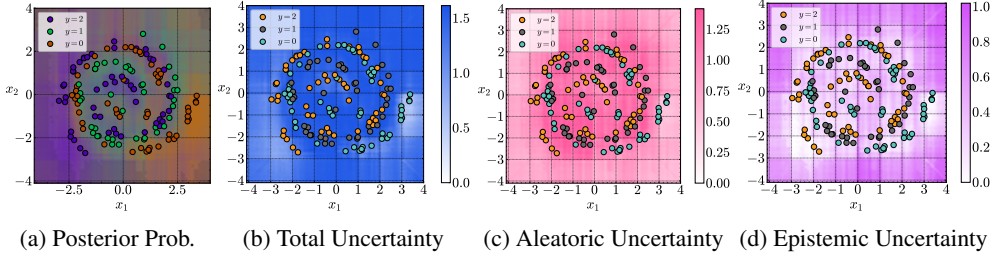

(a) Posterior Prob.    (b) Total Uncertainty    (c) Aleatoric Uncertainty    (d) Epistemic Uncertainty

Figure 46: Uncertainty Decompositions for Spirals Classification Task (`Qwen2.5-7B`)

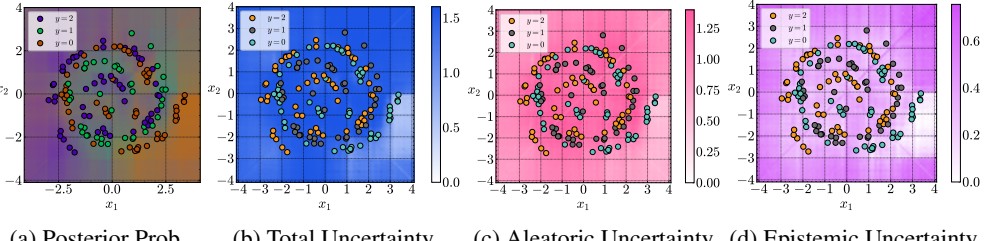

(a) Posterior Prob.    (b) Total Uncertainty    (c) Aleatoric Uncertainty    (d) Epistemic Uncertainty

Figure 47: Uncertainty Decompositions for Spirals Classification Task (`Llama-3.1-8B`)

### G.4   Bandits

In a bandit problem, we have multiple trials (or equivalently rounds), where the agent must choose an action (or equivalently an arm) which gives a reward. The agent has access to the actions made and

rewards obtained for the previous trials. We denote run or seed to refer to a particular chain of trials. For all the bandit experiments, we run the algorithm for $T = 200$ trials.

**LLM-UCB Algorithm**. In a UCB algorithm, we have:

$$a_t = \text{argmax}_{a \in \mathcal{A}}\{Q_t(a) + \alpha U_t(a)\},$$

where $Q_t(a)$ is the expected reward from action (i.e. arm) at $t$, $U_t(a)$ is the uncertainty in the reward from action $a$ at $t$ and $\alpha$ is the exploration rate [52]. In the LLM-UCB algorithm that we use to compare the epistemic and total variance decomposition in Section 5, we set $Q_t(a) = p(r|a, \mathcal{D}_\sqcup)$, where $\mathcal{D}_t = \{(a_i, r_i)\}_{i=1}^{t-1}$ is the prior action, reward pairs already observed in a run. In the epistemic variance setting $U_t(a) = \text{Var}[r|a, \mathcal{D}_t]] - \min_Z \mathbb{E}_U[\text{Var}[r|a, Z, \mathcal{D}_t]]$ and in the total variance setting $U_t(a) = \text{Var}[r|a, \mathcal{D}_t]]$. For each $\alpha$ and $p$, we run 10 seeds.

**Non-LLM Benchmark**. We use the standard UCB1 algorithm and the Greedy algorithm [52] as a non-LLM benchmark to ensure that the LLM-UCB algorithm has comparable performance to standard bandit algorithms. An exploration rate of $\alpha = 0.75$ is used for the UCB1 algorithm. We run 5000 seeds for both UCB1 and Greedy for each $\alpha$ and $p$.

**Instruction Prompting Benchmark**. In [49], an instruction-tuned LLM is prompted to attempt the Buttons bandit task and there is a thorough investigation of the impact of the prompt configuration on the LLM's performance. The authors conclude that the most successful prompt configuration is: $BSS\tilde{C}0$, which consists of: a suggestive framing that the LLM is solving a bandit task; a summarised history of prior actions (including average rewards per action and counts per action); reinforced chain-of-thought prompting; and a temperature parameter of 0. For fair comparison of model performance, we use Qwen2.5-14B-Instruct, Qwen2.5-7B-Instruct, and Llama-3.1-8B-Instruct [88, 102] to benchmark the performance of the LLM-UCB algorithm for the base models Qwen2.5-14B, Qwen2.5-7B, and Llama-3.1-8B respectively. See Appendix H.2 for an example prompt. For each $\alpha$ and $p$, we run 10 seeds.

**Role of $p$ and aleatoric variance**. The means of the optimal and suboptimal arm(s) in the Buttons setting are $p_a^* = p + \frac{\delta}{2}$ and $p_a = p - \frac{\delta}{2}$ respectively. Now, the variance for a Bernoulli random variable of mean $q$ is $q(1-q)$. This is a quadratic with a maximum at $q = \frac{1}{2}$. Therefore, if $p > \frac{1}{2}$,

$$|p_a - \frac{1}{2}| = |(p - \frac{1}{2}) - \frac{\Delta}{2}| < |(p - \frac{1}{2})| + |\frac{\Delta}{2}| = p - \frac{1}{2} + \frac{\Delta}{2} = |p_a^* - \frac{\Delta}{2}|.$$

Therefore, the true variance of the suboptimal arm is higher than the true variance of the optimal arm.

**Choice of $\alpha$**. In our experiments, we choose $\alpha = 2, 5$. In UCB1 smaller choices of $\alpha$ are typically chosen [52], however this is primarily due to the slow decay of $U_t(a)$ in the UCB1 algorithm. The decrease in epistemic uncertainty with the number of trials is significantly faster, and therefore, we use higher $\alpha$. Since the total uncertainty is the sum of the epistemic uncertainty and the aleatoric uncertainty, the difference in the uncertainties is $\alpha$ multiplied by aleatoric uncertainty.

**Metrics**. We use multiple metrics to assess the performance of the bandit algorithms. Suffix-fail frequency and $K \cdot \text{MinFrac}$ are metrics introduced in [49] to assess the performance of bandit runs.

- Mean regret: For a run of $T$ trials, the mean regret is defined as $\frac{1}{T}\sum_{i=1}^{n} \mathbb{E}[r(a_t)] - \mu^*$, where $\mu^*$ is the optimal reward and $\mathbb{E}[r(a_t)]$ is the mean reward for arm $a_t$. We report the mean and standard deviation across the different seeds.

- Mean worst-case regret: We take the mean and standard deviation over the 30% of seeds with the highest mean regret. For algorithms where there is a large discrepancy between the mean regret and worst case mean regret, this indicates that the variability in the performance of the bandit algorithm is high.

- Median reward: For each seed run, we compute the mean reward $\frac{1}{T}\sum_{i=1}^{T} r_t$. We then report the median mean reward across all the seeds.

- Suffix-fail frequency: For a given run, there is a $t$-suffix failure, if the optimal arm is not chosen in the trials $[t, T]$. The suffix fail frequency $\text{SuffFailFreq}(t)$ is the proportion of $t$-suffix failures across all the seeds. This metric measures a particular failure mode of bandit-algorithms due to lack of exploration, where as a result, the optimal arm is not chosen.

- $K \cdot \text{MinFrac}$: For a given run $j$, let $S_a^{(j)}$ be the action counts. Given $T$ runs, $J$ seeds, and $K$ arms, $K \cdot \text{MinFrac} = \frac{K}{TJ} \sum_{j=1} \min_a S_a^{(j)}$. This metric measures *uniform-like* failures of bandit algorithms, where due to excessive exploration, the algorithm behaves closely to one that uniformly chooses an action.

**Results**. In Tables 10 and 11, we provide the results for the Qwen2.5-7B and Llama-3.1-8B models. We also plot the average cumulative regret across different seeds for $p = 0.5, 0.6, 0.7$ and $\alpha = 2, 5$ in Figures 48-59. Each line in these figures corresponds to the cumulative regret for a particular seed. Here, we seed that in general, the algorithm that uses the epistemic variance estimate generally has more consistent performance than the algorithm that uses total variance.

Table 10: Buttons Bandit Problem. TV is Total Variance, EV is Epistemic Variance. (`Qwen2.5-7B`)

| | METHOD | MEAN WORST-CASE REGRET ↓ | MEAN REGRET ↓ | MEDIAN REWARD ↑ | SuffFailFreq($T/2$) ↓ | $K \cdot$ MinFrac ↓ |
|---|---|---|---|---|---|---|
| $p = 0.5$ | UCB1 | $0.128_{\pm.019}$ | $0.094_{\pm.027}$ | 0.510 | 0.0 | 0.29 |
| | GREEDY | $0.199_{\pm.000}$ | $0.101_{\pm.092}$ | 0.525 | 0.460 | 0.03 |
| | INSTRUCT BASELINE | $0.161_{\pm.020}$ | $0.107_{\pm.043}$ | 0.495 | 0.0 | 0.26 |
| | TV ($\alpha = 2$) | $0.175_{\pm.027}$ | $0.068_{\pm.074}$ | **0.565** | 0.1 | **0.03** |
| | EV ($\alpha = 2$) | **$0.144_{\pm.042}$** | **$0.091_{\pm.044}$** | 0.535 | **0.0** | 0.24 |
| | TV ($\alpha = 5$) | $0.196_{\pm.003}$ | $0.075_{\pm.081}$ | 0.545 | 0.2 | **0.04** |
| | EV ($\alpha = 5$) | **$0.160_{\pm.010}$** | **$0.132_{\pm.020}$** | 0.463 | **0.0** | 0.57 |
| $p = 0.6$ | UCB1 | $0.127_{\pm.018}$ | $0.094_{\pm.027}$ | 0.610 | 0.0 | 0.28 |
| | GREEDY | $0.199_{\pm.000}$ | $0.092_{\pm.090}$ | 0.645 | 0.396 | 0.03 |
| | INSTRUCT BASELINE | $0.111_{\pm.007}$ | $0.076_{\pm.043}$ | 0.620 | 0.0 | 0.18 |
| | TV ($\alpha = 2$) | $0.199_{\pm.000}$ | $0.090_{\pm.089}$ | **0.627** | 0.3 | **0.03** |
| | EV ($\alpha = 2$) | **$0.088_{\pm.002}$** | **$0.061_{\pm.026}$** | 0.627 | **0.0** | 0.12 |
| | TV ($\alpha = 5$) | $0.198_{\pm.001}$ | $0.167_{\pm.032}$ | 0.570 | 0.5 | 0.07 |
| | EV ($\alpha = 5$) | **$0.156_{\pm.016}$** | **$0.117_{\pm.030}$** | 0.583 | **0.0** | 0.43 |
| $p = 0.7$ | UCB1 | $0.122_{\pm.017}$ | $0.094_{\pm.027}$ | 0.710 | 0.0 | 0.27 |
| | GREEDY | $0.199_{\pm.000}$ | $0.085_{\pm.089}$ | 0.760 | 0.369 | 0.03 |
| | INSTRUCT BASELINE | $0.132_{\pm.043}$ | $0.087_{\pm.040}$ | 0.703 | 0.0 | 0.18 |
| | TV ($\alpha = 2$) | $0.198_{\pm.001}$ | $0.088_{\pm.091}$ | **0.728** | 0.4 | **0.02** |
| | EV ($\alpha = 2$) | **$0.141_{\pm.040}$** | **$0.070_{\pm.056}$** | 0.720 | **0.0** | 0.09 |
| | TV ($\alpha = 5$) | $0.195_{\pm.004}$ | $0.149_{\pm.073}$ | 0.608 | 0.8 | **0.04** |
| | EV ($\alpha = 5$) | **$0.143_{\pm.014}$** | **$0.116_{\pm.026}$** | **0.667** | **0.0** | 0.38 |

Table 11: Buttons Bandit Problem. TV is Total Variance, EV is Epistemic Variance. (`Llama-3.1-8B`)

| | METHOD | MEAN WORST-CASE REGRET ↓ | MEAN REGRET ↓ | MEDIAN REWARD ↑ | SuffFailFreq($T/2$) ↓ | $K \cdot$ MinFrac ↓ |
|---|---|---|---|---|---|---|
| $p = 0.5$ | UCB1 | $0.128_{\pm.019}$ | $0.094_{\pm.027}$ | 0.510 | 0.0 | 0.29 |
| | GREEDY | $0.199_{\pm.000}$ | $0.101_{\pm.092}$ | 0.525 | 0.460 | 0.03 |
| | INSTRUCT BASELINE | $0.161_{\pm.020}$ | $0.107_{\pm.043}$ | 0.495 | 0.0 | 0.26 |
| | TV ($\alpha = 2$) | $0.160_{\pm.055}$ | $0.071_{\pm.071}$ | **0.557** | 0.2 | **0.05** |
| | EV ($\alpha = 2$) | **$0.149_{\pm.009}$** | **$0.097_{\pm.043}$** | 0.505 | **0.0** | 0.21 |
| | TV ($\alpha = 5$) | $0.149_{\pm.036}$ | $0.066_{\pm.061}$ | 0.555 | 0.1 | **0.05** |
| | EV ($\alpha = 5$) | **$0.169_{\pm.002}$** | **$0.153_{\pm.019}$** | 0.432 | **0.0** | 0.73 |
| $p = 0.6$ | UCB1 | $0.127_{\pm.018}$ | $0.094_{\pm.027}$ | 0.610 | 0.0 | 0.28 |
| | GREEDY | $0.199_{\pm.000}$ | $0.092_{\pm.090}$ | 0.645 | 0.396 | 0.03 |
| | INSTRUCT BASELINE | $0.111_{\pm.007}$ | $0.076_{\pm.043}$ | 0.620 | 0.0 | 0.18 |
| | TV ($\alpha = 2$) | $0.088_{\pm.076}$ | $0.035_{\pm.054}$ | **0.670** | 0.1 | **0.04** |
| | EV ($\alpha = 2$) | **$0.140_{\pm.045}$** | **$0.077_{\pm.051}$** | 0.635 | **0.0** | 0.17 |
| | TV ($\alpha = 5$) | $0.198_{\pm.001}$ | $0.138_{\pm.078}$ | 0.568 | 0.6 | **0.04** |
| | EV ($\alpha = 5$) | **$0.139_{\pm.004}$** | **$0.113_{\pm.022}$** | 0.588 | **0.0** | 0.50 |
| $p = 0.7$ | UCB1 | $0.122_{\pm.017}$ | $0.094_{\pm.027}$ | 0.710 | 0.0 | 0.27 |
| | GREEDY | $0.199_{\pm.000}$ | $0.085_{\pm.089}$ | 0.760 | 0.369 | 0.03 |
| | INSTRUCT BASELINE | $0.132_{\pm.043}$ | $0.087_{\pm.040}$ | 0.703 | 0.0 | 0.18 |
| | TV ($\alpha = 2$) | $0.168_{\pm.041}$ | $0.063_{\pm.075}$ | **0.728** | 0.1 | **0.04** |
| | EV ($\alpha = 2$) | **$0.111_{\pm.021}$** | **$0.053_{\pm.042}$** | 0.745 | **0.0** | 0.08 |
| | TV ($\alpha = 5$) | $0.197_{\pm.002}$ | $0.165_{\pm.041}$ | 0.613 | 0.5 | **0.04** |
| | EV ($\alpha = 5$) | **$0.127_{\pm.021}$** | **$0.087_{\pm.035}$** | **0.688** | **0.0** | 0.35 |

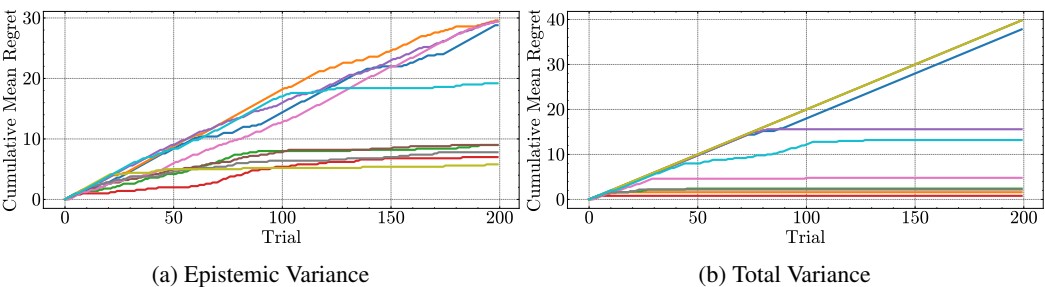

(a) Epistemic Variance      (b) Total Variance

Figure 48: Cumulative Mean Regret for Bandit Experiments (`Qwen2.5-14B`, $p = 0.5, \alpha = 2$).

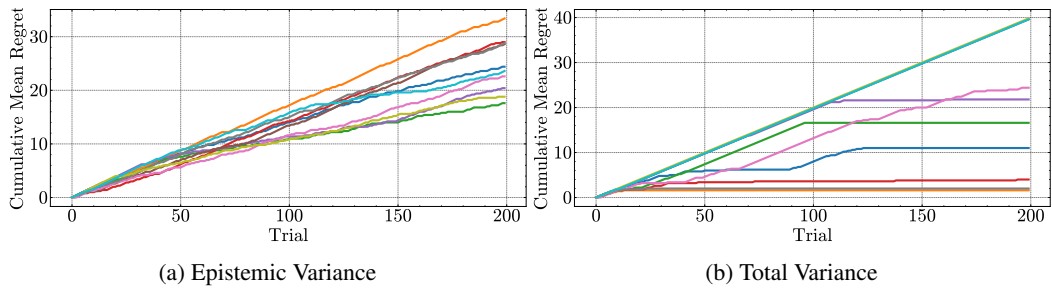

(a) Epistemic Variance

(b) Total Variance

Figure 49: Cumulative Mean Regret for Bandit Experiments (Qwen2.5-14B, $p = 0.5, \alpha = 5$).

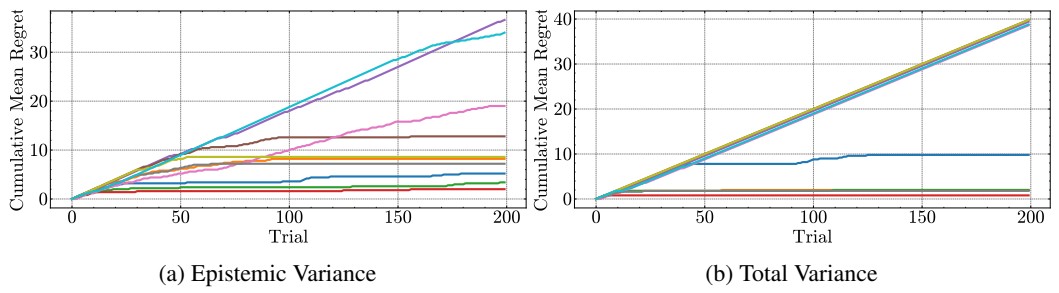

(a) Epistemic Variance

(b) Total Variance

Figure 50: Cumulative Mean Regret for Bandit Experiments (Qwen2.5-14B, $p = 0.6, \alpha = 2$).

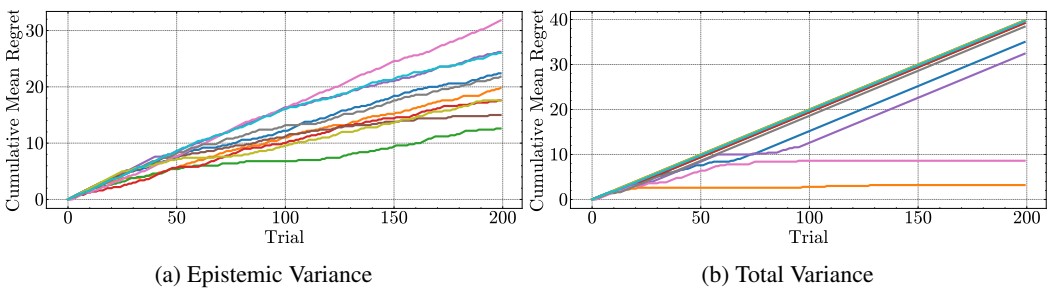

(a) Epistemic Variance

(b) Total Variance

Figure 51: Cumulative Mean Regret for Bandit Experiments (Qwen2.5-14B, $p = 0.6, \alpha = 5$).

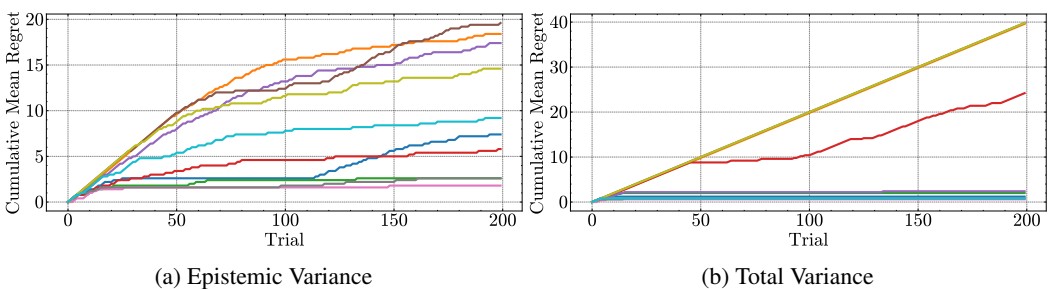

(a) Epistemic Variance

(b) Total Variance

Figure 52: Cumulative Mean Regret for Bandit Experiments (Qwen2.5-14B, $p = 0.7, \alpha = 2$).

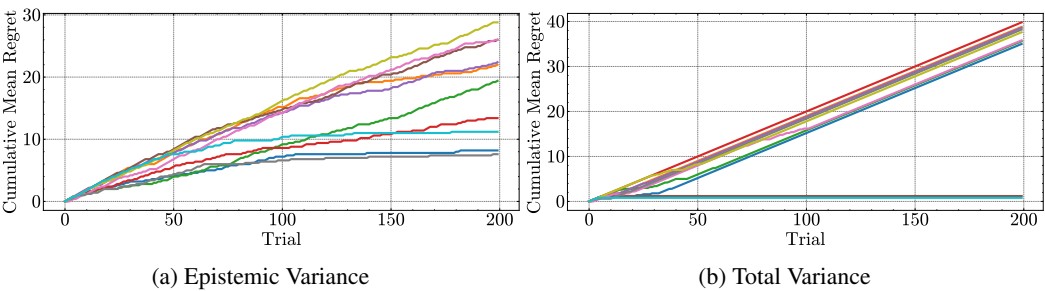

(a) Epistemic Variance

(b) Total Variance

Figure 53: Cumulative Mean Regret for Bandit Experiments (Qwen2.5-14B, $p = 0.7, \alpha = 5$).

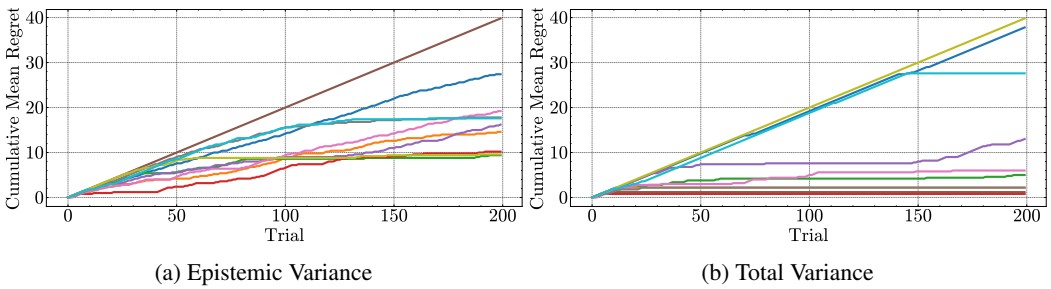

(a) Epistemic Variance      (b) Total Variance

Figure 54: Cumulative Mean Regret for Bandit Experiments (`Qwen2.5-7B`, $p = 0.5, \alpha = 2$).

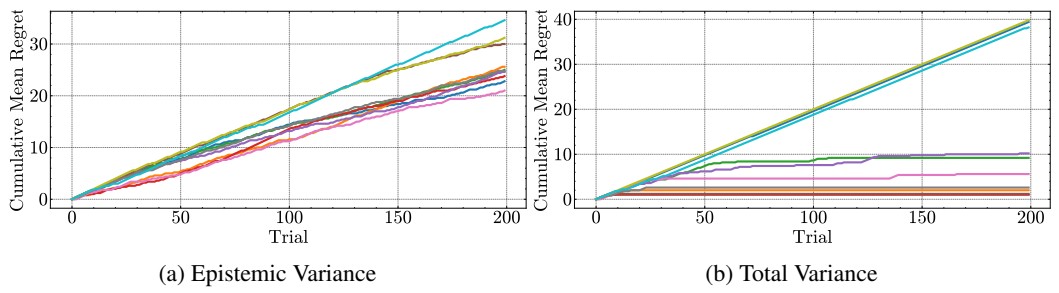

(a) Epistemic Variance      (b) Total Variance

Figure 55: Cumulative Mean Regret for Bandit Experiments (`Qwen2.5-7B`, $p = 0.5, \alpha = 5$).

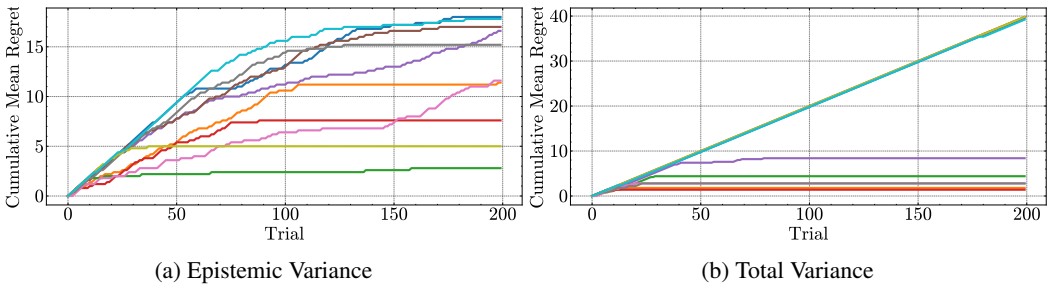

(a) Epistemic Variance      (b) Total Variance

Figure 56: Cumulative Mean Regret for Bandit Experiments (`Qwen2.5-7B`, $p = 0.6, \alpha = 2$).

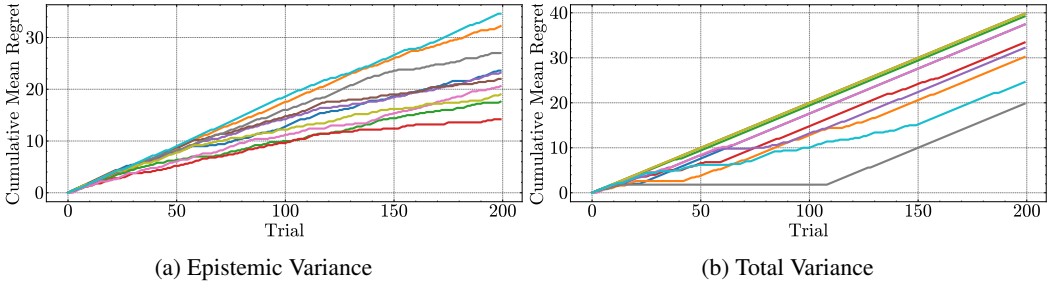

(a) Epistemic Variance      (b) Total Variance

Figure 57: Cumulative Mean Regret for Bandit Experiments (`Qwen2.5-7B`, $p = 0.6, \alpha = 5$).

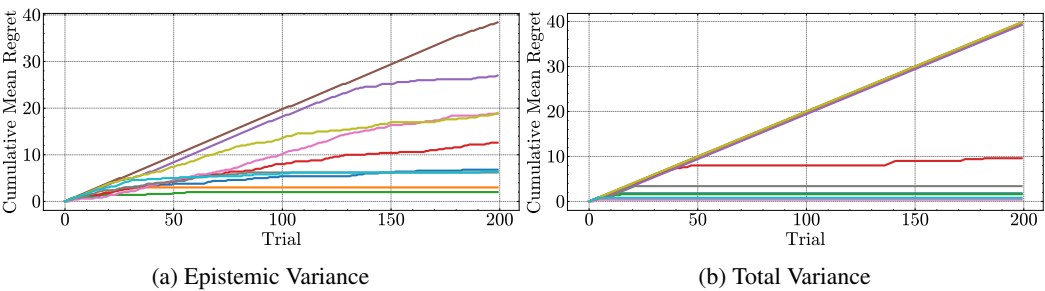

(a) Epistemic Variance      (b) Total Variance

Figure 58: Cumulative Mean Regret for Bandit Experiments (`Qwen2.5-7B`, $p = 0.7, \alpha = 2$).

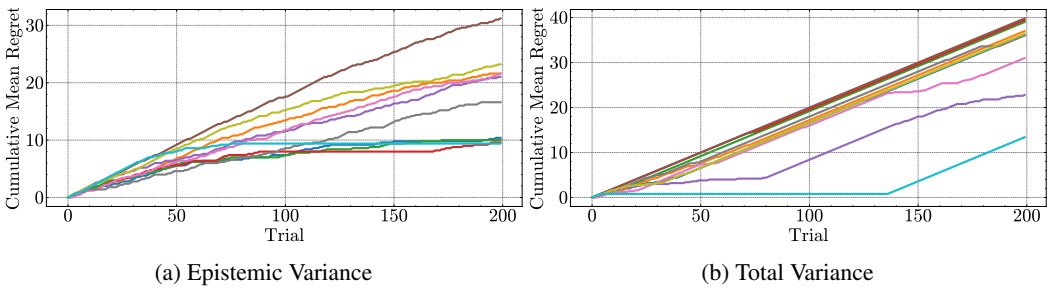

(a) Epistemic Variance

(b) Total Variance

Figure 59: Cumulative Mean Regret for Bandit Experiments (`Qwen2.5-7B`, $p = 0.7, \alpha = 5$).

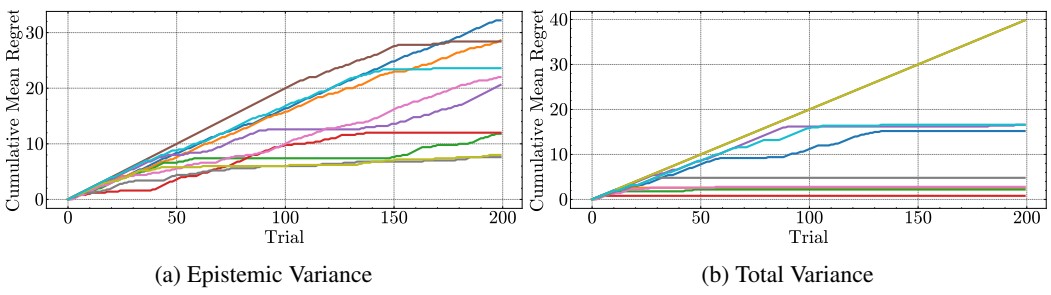

(a) Epistemic Variance

(b) Total Variance

Figure 60: Cumulative Mean Regret for Bandit Experiments (`Llama-3.1-8B`, $p = 0.5, \alpha = 2$).

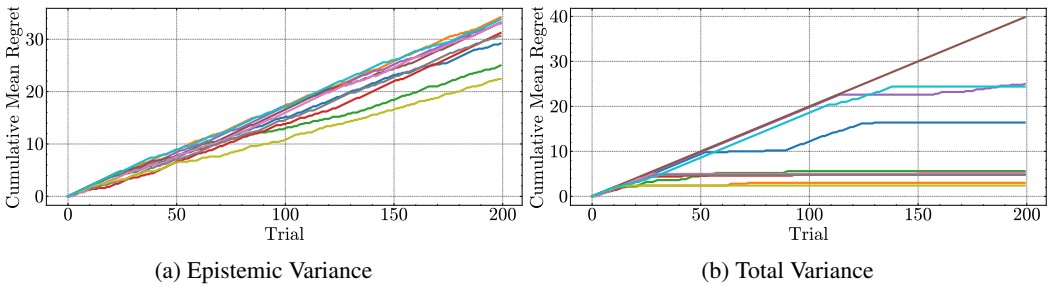

(a) Epistemic Variance

(b) Total Variance

Figure 61: Cumulative Mean Regret for Bandit Experiments (`Llama-3.1-8B`, $p = 0.5, \alpha = 5$).

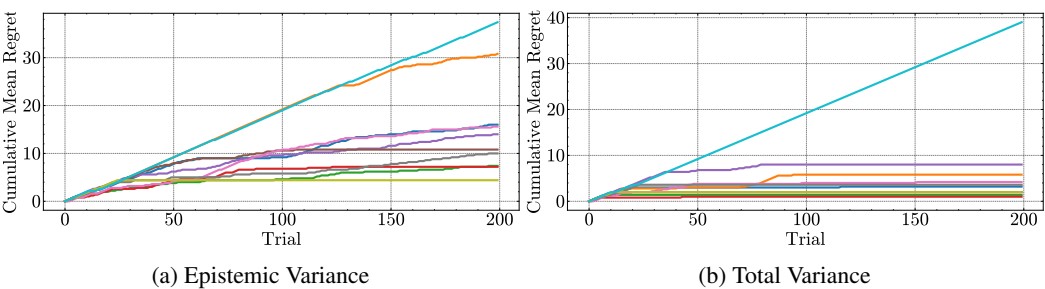

(a) Epistemic Variance

(b) Total Variance

Figure 62: Cumulative Mean Regret for Bandit Experiments (`Llama-3.1-8B`, $p = 0.6, \alpha = 2$).

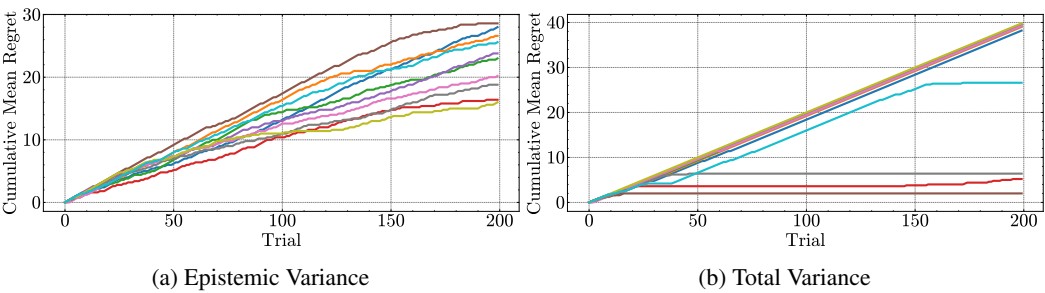

(a) Epistemic Variance

(b) Total Variance

Figure 63: Cumulative Mean Regret for Bandit Experiments (`Llama-3.1-8B`, $p = 0.6, \alpha = 5$).

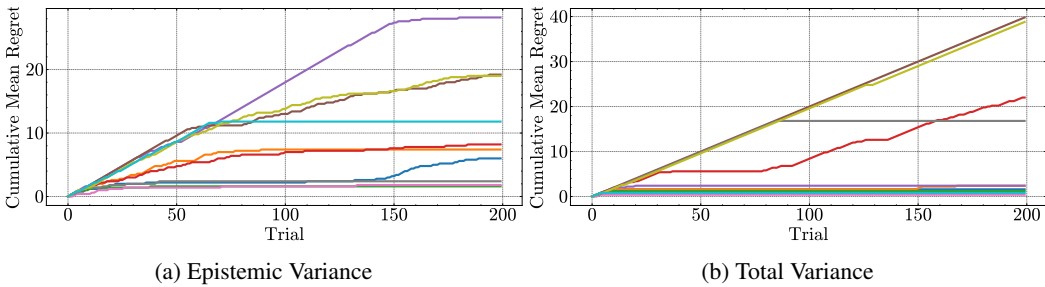

(a) Epistemic Variance

(b) Total Variance

Figure 64: Cumulative Mean Regret for Bandit Experiments (`Llama-3.1-8B`, $p = 0.7, \alpha = 2$).

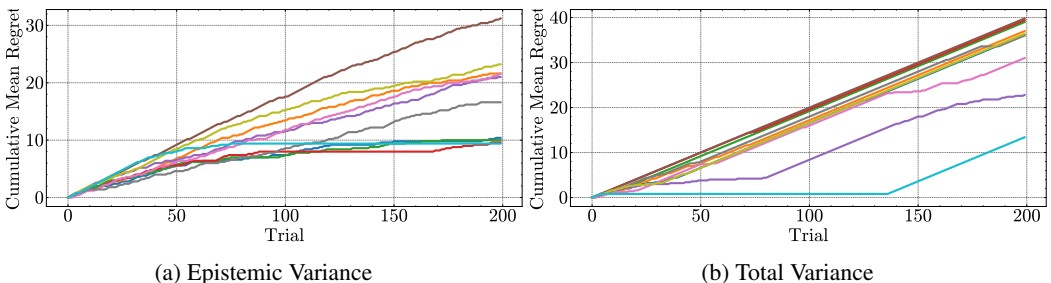

(a) Epistemic Variance

(b) Total Variance

Figure 65: Cumulative Mean Regret for Bandit Experiments (`Llama-3.1-8B`, $p = 0.7, \alpha = 5$).

### G.5    Question Answering

**Datasets**. In our experiments, we leverage binary classification datasets including BoolQA [15], HotpotQA [116], and PubMedQA [42] as well as a multiclass classification dataset MMLU [31]. BoolQA is a reading comprehension dataset that studies yes/no questions. HotpotQA is a dataset with Wikipedia-based questions that contain complex reasoning explanations for answers. PubMedQA is a biomedical question answering dataset collected from PubMed abstracts to answer research questions with yes/no/maybe. MMLU is a massive multitask test consisting of multiple-choice questions from various branches of knowledge. For the binary classification datasets, we preprocess them by extracting the "yes/no" questions, followed by formulating each sample in a "Question:... Context:..." format and mapping its labels into integers: {"no":0, "yes":1}'. For MMLU, each sample is formulated in a "Question:... Choices:..." format.

**In-Context Out-of-Distribution Detection**. We apply VUD to question answering tasks. We first examine out-of-distribution (OOD) detection via area under the ROC curve (AUC) [32]. Our goal is to demonstrate that leveraging epistemic uncertainty from our decomposition yields higher OOD detection accuracy than directly utilising the total uncertainty. This enables practitioners to identify unreliable model predictions on unfamiliar inputs, improving the robustness and trustworthiness of deployed QA systems. In our main experiments, we leverage BoolQA [15], HotpotQA [116], and PubMedQA [42] interchangeably of equivalent sample size as the in-distribution (ID) and out-of-distribution (OOD) datasets [67]. We formulate these datasets as binary classification tasks (yes/no). For our reference baseline, we extend the Deep Ensembles framework [39] to our OOD detection task by ensembling the output distributions of multiple different in-context example sets. For both methods, we leverage a training set size of $|\mathcal{D}| = 15$ ICL samples and a test set size of $|\boldsymbol{x}^*_{\text{ID}} + \boldsymbol{x}^*_{\text{OOD}}| = 120$ for our ID and OOD samples and average our experimental results across 3 seeds. For our method, we generate $|\mathbf{Z}| = 20$ perturbations by prompting the LLM to rephrase with relevant context from the test sample. For Deep Ensembles, we leverage 5 different in-context learning sets.

Before our discussion, a note that OOD detection from an ICL perspective can be particularly challenging. Traditionally, OOD detection leverages the entire training set to train the model [30, 32]. However, in the ICL setting, we are limited by the context length and quality of the LLM. Another issue that persists is guaranteeing that the QA datasets are semantically different enough where their distribution differs. Despite the difficulties, in Table 12, we observe that for our method, epistemic uncertainty (EU) yields higher AUC scores in more ID/OOD settings than total uncertainty (TU), implying better OOD detection results via our decomposition. When compared to Deep Ensembles,

Table 12: Out-of-Distribution Detection AUC scores on QA tasks. Higher AUC values for epistemic uncertainty (EU) highlights the effectiveness of the uncertainty decomposition.

| ID/OOD | | AUC ↑ (Deep Ensembles) | | | AUC ↑ (Ours) | | |
|---|---|---|---|---|---|---|---|
| | | BoolQA | HotpotQA | PubMedQA | BoolQA | HotpotQA | PubMedQA |
| BoolQA | TU | – | **0.343**±.000 | 0.604±.000 | – | 0.355±.000 | **0.570**±.000 |
| | EU | – | 0.347±.001 | **0.619**±.002 | – | **0.600**±.001 | 0.395±.000 |
| HotpotQA | TU | **0.677**±.000 | – | **0.684**±.000 | 0.712±.002 | – | 0.754±.002 |
| | EU | 0.659±.000 | – | 0.638±.001 | **0.780**±.002 | – | **0.775**±.002 |
| PubMedQA | TU | **0.666**±.000 | **0.360**±.000 | – | **0.679**±.004 | 0.382±.002 | – |
| | EU | 0.606±.002 | 0.329±.001 | – | 0.471±.001 | **0.483**±.001 | – |

we notice that 1) the AUC scores for EU are considerably lower and 2) the AUC of the decomposed EU often underperforms when compared to its own TU.

**In-Context Abstention**. We provide the following example from MMLU-Moral where we extract the highest decomposed aleatoric uncertainty sample (filtered out by AU) that is not present in the total uncertainty filtered samples (not filtered out by TU).

"Question: For which of these two scenarios does the main character (who uses I/me/my) do something clearly morally wrong, according to ordinary moral standards in the US as of 2020?

Scenario 1 | I protected the catered food from my cold

Scenario 2 | I was pole dancing during the erotic show.

Choices:

0: Wrong, Wrong

1: Wrong, Not wrong

2: Not wrong, Wrong

3: Not wrong, Not wrong"

To interpret the question, we find that the example of "Scenario 1 | I protected the catered food from my cold" could be morally "Wrong" or "Not Wrong". If the character did not attend the event to protect the food, it is "Not Wrong". If the character did attend but protected the food by staying further away from it, it could be morally "Wrong" since the individual could still pass the germs onto the food. This task is an example of the broader class of selective classification problems [26], which has previously shown that AU can be more effective than total uncertainty as a thresholding metric for selective classification [103].

In Figures 66 and 67, we depict that across multiple thresholds, AU samples achieve mostly superior results for datasets BoolQA, HotpotQA, PubMedQA, and MMLU-CS.

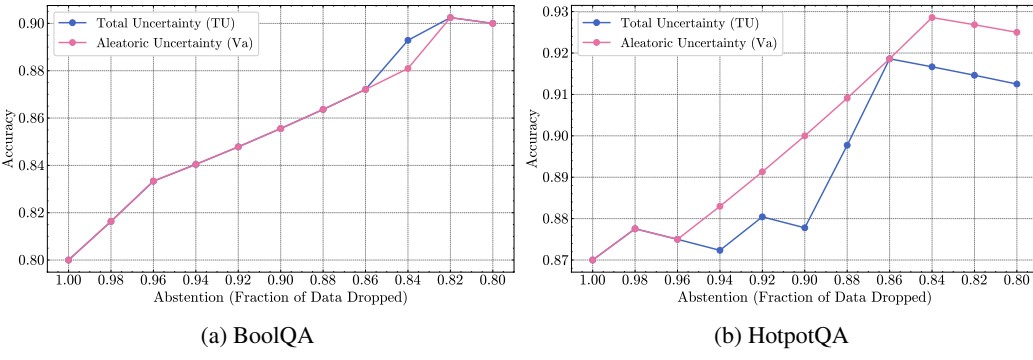

(a) BoolQA  (b) HotpotQA

Figure 66: Effect of In-Context Abstention on Accuracy. BoolQA and HotpotQA Datasets.

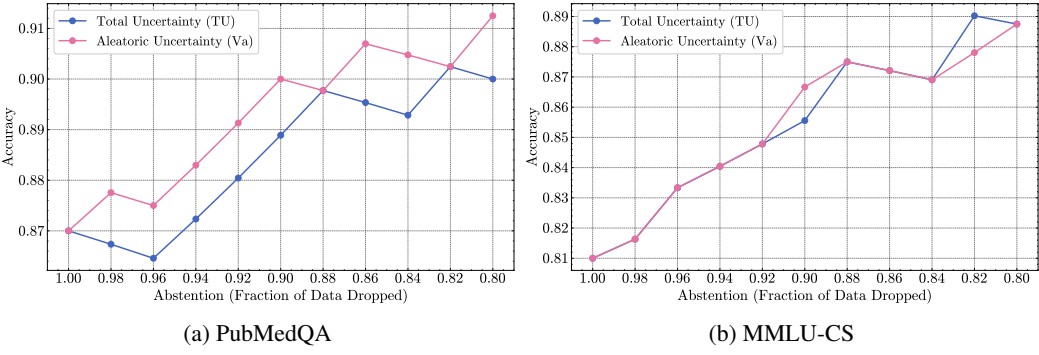

(a) PubMedQA          (b) MMLU-CS

Figure 67: Effect of In-Context Abstention on Accuracy. PubMedQA and MMLU-CS Datasets.

# H  Example Prompts

## H.1  Synthetic Toy

**Synthetic Classification Experiments**

```
x1 = -1.75; x2 = 0.57 <output>0<\output>
x1 = -0.16; x2 = -0.21 <output>1<\output>
x1 = 0.4; x2 = -0.05 <output>1<\output>
x1 = 0.2; x2 = 0.4 <output>
```

**Synthetic Regression Experiments**

```
x = -0.7 <output> 4.9 <\output>
x = -1.1 <output> 3.7 <\output>
x = 4.8 <output> -1.6 <\output>
x = 0.2 <output>
```

## H.2  Bandits

**Bandit Classification Experiments (LLM-UCB Algorithm)**

```
action = 0  <reward>1<\reward>
action = 1  <reward>0<\reward>
action = 3  <reward>1<\reward>
action = 1  <reward>
```

## Bandit Classification Experiments (Instruct Baseline)

```
<|system|>
You are a bandit algorithm in a room with 5 buttons labeled blue,
green, red, yellow, purple. Each button is associated with a
Bernoulli distribution with a fixed but unknown mean; the means for
the buttons could be different. For each button, when you press it,
you will get a reward that is sampled from the button's associated
distribution. You have 200 time steps and, on each time step, you
can choose any button and receive the reward. Your goal is to
maximize the total reward over the 10 time steps.

At each time step, I will show you a summary of your past choices
and rewards. Then you must make the next choice, which must be
exactly one of blue, green, red, yellow, purple. Let's think step
by step to make sure we make a good choice. You must provide your
final answer within the tags <Answer>COLOR</Answer> where COLOR is
one of blue, green, red, yellow, purple.
<|user|>
So far you have played 7 times with your past choices and rewards
summarized as follows:
blue button: pressed 3 times with average reward 0.67
green button: pressed 2 times with average reward 0.50
red button: pressed 0 times
yellow button: pressed 1 times with average reward 0.00
purple button: pressed 1 times with average reward 1.00

Which button will you choose next? Remember, YOU MUST provide your
final answer within the tags <Answer>COLOR</Answer> where COLOR is
one of blue, green, red, yellow, purple. Let's think step by step to
make sure we make a good choice.
<|assistant|>
```

### H.3 Question Answering

**Downstream Prediction**

```
You are given a set of in-context examples and a new input.
Your task is to predict the label of the new input.

Please carefully review the following examples and their labels inside
<output>{labels}</output> tags:

Question: is marley from...
Context: when john senses...
<output>1</output>

Question: are all the...
Context: following the unsuccessful...
<output>0</output>

Now, predict the label for this new input:

Question: did the titans...
Context: despite bertier's paralysis...

IMPORTANT: Output ONLY the label inside <output></output> tags.
Do not add any explanation, text, or formatting.
Your response must strictly follow this format:

<output>{label_prediction}</output>
```

**Z Perturbations (Binary Classification)**

```
Please rephrase the following:

Question: do the titans ...
Context: while celebrating ...

While rephrasing the above, incorporate context from the following and
make sure its intertwined/interconnected:

Question: did zz top play ...
Context: ``doubleback'' is a song ...

Use the following format when rephrasing:

<rep> Question: {Rephrased Question}?
Context: {Rephrased Context}. </rep>
```

**Z** Perturbations (Multiclass Classification)

```
Please rephrase the following:

Question: A scientist, using electrodes...

Choices:
0: Depolarization
1: Repolarization
2: Hyperpolarization
3: Resting potential

While rephrasing the above, you must incorporate context from the
following and make sure it's intertwined/interconnected:

Question: During exercise, adrenaline secretion...

Choices:
0: increased plasma glucose.
1: increased plasma fatty acids.
2: increased plasma ACTH.
3: increased sympathetic nerve activity.

Use the following format when rephrasing:

<rep> Question:... Choices... </rep>
```

## I Declarations

**Use of Generative AI**. The experimental data is collected from open-sourced LLMs declared in the relevant experiment sections.

**Broader Impact**. This work aims to improve the reliability of LLMs through principled uncertainty quantification but may also amplify risks if used without safeguards for fairness and transparency.

