# OpenReview forum: "Variational Uncertainty Decomposition for In-Context Learning"
_NeurIPS.cc/2025/Conference — NeurIPS 2025 poster_

### Official Review · Reviewer_esK3 · 2025-07-01

**Clarity:** 2
**Significance:** 3
**Originality:** 3
**Rating:** 4
**Confidence:** 3

**Summary:**

This paper proposes a novel variational framework for uncertainty decomposition in in-context learning (ICL). It avoids both sampling and explicit computation of the model parameter posterior which are notoriously difficult for ICL. Specifically, it derives an upper bound on aleatoric uncertainty by introducing auxiliary inputs that serve as probes into the underlying parameter posterior. To further enhance the Bayesian property of exchangeability in ICL, the method introduces random shuffling of context permutations combined with KL-based filtering. The effectiveness of the proposed framework is demonstrated through synthetic experiments that highlight the interpretability of the decomposed uncertainties, with additional applications shown in bandit problems and out-of-distribution detection in question-answering tasks.

**Questions:**

1. If I understand correctly, the proposed framework can be applied to estimate aleatoric and epistemic uncertainty for machine learning models with point-estimated parameters. Does this imply that the method is broadly applicable to deep active learning scenarios? It would be helpful if the authors could elaborate on this potential direction.

2. In the KL filtering step, what happens if no candidate satisfies the condition given in Equation (11)? How is this case handled in practice?

3. Restricting the search for $Z$ to a single example from the input domain seems limiting. How might the approach be extended to consider a broader set of $Z$ values, potentially reducing the approximation gap in Theorem 3.1?

4. In the linear regression experiment, the observational noise is modeled as Gaussian with fixed variance. Assuming the regression parameters are estimated via MAP or MLE (i.e., without a Bayesian posterior), why does the total predictive uncertainty vary across different test covariates?

**Ethical Concerns:**

["NO or VERY MINOR ethics concerns only"]

**Final Justification:**

While other reviewers raised concerns about auxiliary information choices and inference time complexity, I believe this paper merits acceptance. Despite its limitations, it offers valuable theoretical novelty in uncertainty quantification for in-context learning and could potentially inspire future research developments.

**Limitations:**

yes

**Paper Formatting Concerns:**

I do not notice any major formatting issues in this paper.

**Quality:**

3

**Strengths And Weaknesses:**

To the best of my knowledge, the proposed framework is novel and offers an interesting contribution. It introduces a new perspective on the uncertainty of model parameter posteriors through the use of probes, such as auxiliary inputs and outputs. The methods of context shuffling and KL filtering are well-justified and conceptually sound. Moreover, the synthetic experiments provide interpretable evidence supporting the proposed uncertainty decomposition.

A key concern pertains to Theorem 3.1, which is based on marginalizing out the model parameters $\theta$. The justification provided in lines 130-132 suggests that "some of the uncertainty in $\theta$ is absorbed into...".
However, in in-context learning (ICL), $\theta$ is treated as a point estimate, implying that there is no posterior uncertainty over model parameters. This discrepancy raises questions about the applicability of the same uncertainty-based interpretation within the ICL framework.

Another point of concern lies in Equation (5). While the framework proposes identifying $Z$ as a perturbation of $x*$, the equation assumes access to $p(U | Z, D)$. This assumption appears problematic, as the ultimate goal is to infer the unknown $p(y* | x*, D)$. If $Z$ is constructed to be close to $x*$, it is unclear how one could reasonably obtain or estimate $p(U | Z, D)$ without already knowing something about $p(y* | x*, D)$.

---

> ### Author Rebuttal · Authors · 2025-07-31
>
> ### Summary Response
>
> We sincerely thank the reviewers for their constructive feedback. We are encouraged that reviewers appreciated the ***novel*** (**bTbz**, **esK3**, **y4yc**) ***principled approach*** (**aVVb**) and ***sound theoretical framework*** (**s63H**) to the ***important and underexplored*** (**y4yc**) problem of uncertainty decomposition in in-context learning.
>
> Our work addresses the fundamental challenge of computing uncertainty decompositions on an intractable and implicitly defined Bayesian model of ICL in LLMs. Specifically:
>
> - We derive an upper bound to the aleatoric uncertainty (conditional entropy/variance) of the Bayesian model.
> - To our knowledge, we are the first achieve this without sampling from the Bayesian model posterior or assumptions on the form of the likelihood, via probing the model with auxiliary inputs.
> - We provide a clear statement of our assumptions and employ techniques such as permutation-based sampling and KL filtering to approximate exchangeability to ensure the validity of our method.
> - Whilst our method is particularly suited to LLMs, it can be applied to any model where there is access to the posterior predictive distribution.
> - We show quantitative and qualitative performance of VUD across a range of synthetic datasets and real-world downstream tasks.
>
> In response to reviewer suggestions, we have expanded our experimental analysis. New results include:
>
> - Table A (see **aVVb**): Additional experimental OOD tasks for natural language on the MMLU dataset
> - Table B (see **aVVb**): New baseline comparisons using Martingale posterior distributions
> - Table C (see **bTbz**): Analysis of the sample efficiency in the number of samples $Z$
>
> We provide detailed responses to individual reviewer comments and questions below.
>
> ---
> ### Individual Response
>
> **Comments: Theorem 3.1**
>
> > A key concern pertains to Theorem 3.1, which is based on marginalizing out the model parameters $\theta$. The justification provided in lines 130-132 suggests that "some of the uncertainty in $\theta$ is absorbed into...". However, in in-context learning (ICL), $\theta$ is treated as a point estimate, implying that there is no posterior uncertainty over model parameters.
> >
> >
> > Another point of concern lies in Equation (5). While the framework proposes identifying $Z$ as a perturbation of $x^* $, the equation assumes access to $p(U | Z, D)$. This assumption appears problematic, as the ultimate goal is to infer the unknown $p(y^* | x^* , D)$. If $Z$ is constructed to be close to $x^* $, it is unclear how one could reasonably obtain or estimate $p(U | Z, D)$ without already knowing something about $p(y^* | x^* , D)$.
> >
>
> We believe this concern may stem from a misinterpretation of the Bayesian model parameter $\theta$. Unlike standard Bayesian deep learning, $\theta$ **is not the LLM network weights** and we are not trying to compute the posterior distribution $p(\theta|\mathcal{D})$. In our framework:
>
> - $x^* $ is the **predictive task**, $y^* $ is the corresponding **label prediction** and $\mathcal{D} = \{( x_ i , y_ i ) \} _ {i = 1 } ^N$ is **in-context data**.
> - If we concatenate $\mathcal{D}$ and $x^* $ to form the **prompt**, the logits of the LLM prediction $y^* $ give the **posterior predictive distribution** $p(y^ * | x^ * , \mathcal{D} ) $ which is tractable.
> - Assuming exchangeability of the LLM predictions, we can apply **de Finetti’s theorem**, which **implicitly** defines a Bayesian model with parameter $\theta$ such that $p ( y^ * | x^ * , \mathcal{D} ) = \int p ( y^ * | x^ * , \theta ) p ( \theta | \mathcal{D} ) d \theta $. Due to the implicit definition of this model we **do not have access to $p( y^ * | x^ * , \theta ) $** or $p( \theta | \mathcal{D} ) $.
>
> Therefore, to clarify the concerns of the reviewer:
>
> - In this setting, $\theta$ is not a point estimate. Rather this is an implicitly defined parameter that we marginalise over to obtain the posterior predictive. This parameter can be viewed as the underlying conceptual model governing the LLMs predictions [1].
> - We set $Z$ as the predictive task with $U$ as the corresponding prediction. Next, we concatenate $Z$ and $\mathcal{D}$ into a single prompt to yield the logits of the prediction $U$, giving the posterior predictive distribution $p(U|Z,\mathcal{D})$. For further clarification, in Section H of the Appendix, we provide example prompts of the ICL data followed by the predictive task.
>
> **Comments: Applicability to Active Learning**
>
> > If I understand correctly, the proposed framework can be applied to estimate aleatoric and epistemic uncertainty for machine learning models with point-estimated parameters. Does this imply that the method is broadly applicable to deep active learning scenarios? It would be helpful if the authors could elaborate on this potential direction.
> >
>
> Yes, indeed, this is one of the key applications of the VUD method. In particular, we can use the lower bound estimate of the mutual information at $x^ * $, which is the **expected information gain** from observing $x^ * $. This is a key quantity in Bayesian Experimental Design and widely used in active learning algorithms [2]. This opens the door to active learning in natural language settings, where traditional Bayesian methods are often infeasible [3]. We agree this is a promising direction and are actively exploring its application to deep active learning scenarios.
>
> **Comments: KL Filtering**
>
> > In the KL filtering step, what happens if no candidate satisfies the condition given in Equation (11)? How is this case handled in practice?
> >
>
> To ensure that all the auxiliary variables aren’t filtered out by the KL-filtering, we rank the KL divergences $\epsilon_i$ by size and only keep the $r$ auxiliary variables with the smallest KL divergence. This ensures that the method guarantees a decomposition whilst $r$ becomes a proxy for the level of approximation, which is left to the discretion of the user. See Appendix D.3 for further details.
>
> **Comments: Extending** $Z$ **to a broader set of values**
>
> > Restricting the search for $Z$ to a single example from the input domain seems limiting. How might the approach be extended to consider a broader set of $Z$ values, potentially reducing the approximation gap in Theorem 3.1?
> >
>
> To reduce the complexity of the optimisation problem, we restrict $Z$ to a single datapoint in the input domain.  However, we can extend this to multiple datapoints, for instance through test-time scaling. We outline a potential method for this below:
>
> To reduce the complexity of the optimization problem, our current approach restricts $Z$ to a single datapoint from the input domain. However, we agree that this may limit the expressiveness of the approximation and that extending the search over broader sets of $Z_i$ could potentially improve the optimisation. A natural extension is to consider small sets of points (e.g., pairs or triplets) as candidates for $Z_i$. One practical way to do this is via a greedy procedure:
>
> 1. **Parallel Evaluation**: compute $V_ a ( y^ *| x^ *, \mathcal{D} , Z_ i )$ for multiple $Z_i$, where each $Z_i$ is a single datapoint.
> 2. **Initial Filtering**: Discard candidates that fail to meet a KL-divergence threshold
> 3. **Combinatorial Expansion**: Form combinations (e.g. ordered pairs $ (Z_ j , Z_ k )$) from the remaining candidates and compute $V_a (y^* | x^* , \mathcal{D} , (Z_j,Z_k) ) $
> 4. **Secondary Filtering**: Apply the KL condition again to filter out weak combinations.
> 5. **Selection**: Choose the optimal set based on the resulting $V_a$ score
>
> While this is a simple strategy, it illustrates a potential direction for future work in trading off computational efficiency and approximation quality.
>
> **Comments: Linear Regression Total Predictive Uncertainty**
>
> > In the linear regression experiment, the observational noise is modeled as Gaussian with fixed variance. Assuming the regression parameters are estimated via MAP or MLE (i.e., without a Bayesian posterior), why does the total predictive uncertainty vary across different test covariates?
> >
>
> While the observational noise in the linear regression experiment is indeed modelled as Gaussian with fixed variance, the total predictive uncertainty we report is derived from the entropy of the LLM posterior predictive distribution $p ( y^* | x^* , \mathcal{D} ) $.  The VUD method decomposes this into the conditional entropy (aleatoric uncertainty) $\mathbb{H} [p ( y^* |x^* , \theta ) ] $ and the mutual information $\mathbb{I} (y ; \theta | x^* , \mathcal{D} ) $ (epistemic uncertainty).
>
> As a heuristic, assuming a Gaussian likelihood and prior for $\theta$, the conditional entropy would be constant due to fixed variance. However, the total uncertainty would be of the form $\frac{1}{2} \log ( { x^ * } ^ T A { x^ * }) + c$ for a positive definite matrix $A$ and constant $c$ [4]. Therefore, the uncertainty would increase as $|x^*|$ becomes large.
>
> We empirically observe this in Figure 4b, where the aleatoric uncertainty is roughly constant but the total uncertainty increases as $|x ^ * |$ increases. While the underlying Bayesian model is not necessary Gaussian, this behavior aligns with the heuristic intuition derived from the Gaussian case, where predictive uncertainty grows with distance from the training data.
>
> ---
>
> [1] Xie, Sang Michael, et al. "An explanation of in-context learning as implicit bayesian inference." *arXiv preprint arXiv:2111.02080* (2021).
>
> [2] Rainforth, Tom, et al. "Modern Bayesian experimental design." *Statistical Science* 39.1 (2024): 100-114.
>
> [3] Margatina, Katerina, et al. "Active learning principles for in-context learning with large language models." arXiv preprint arXiv:2305.14264 (2023).
>
> [4] Williams, Christopher, and Carl Rasmussen. "Gaussian processes for regression." *Advances in neural information processing systems* 8, Chapter 2 (1995).

---

> > ### Comment · Reviewer_esK3 · 2025-08-06
> >
> > Thank you for the detailed response. It has improved my understanding of the paper, particularly that the Bayesian model is implicit, which resolves my concern regarding Theorem 3.1. Regarding the linear regression example, I now understand that you use an LLM to perform the prediction instead of using MLE or MAP estim ates for the weights (without a prior distribution).
> >
> > I have reviewed the feedback from other reviewers, and noted some concerns about choices of auxiliary information and time complexity during inference. However, in my view, uncertainty quantification in in-context learning generally remains a challenging problem. This paper contributes a novel theoretical approach that, despite the limitations mentioned by other reviewers, still merits acceptance from my perspective. The work provides valuable theoretical novelty that may inspire future research and methodological developments in the area.

---

### Official Review · Reviewer_y4yc · 2025-07-02

**Clarity:** 2
**Significance:** 3
**Originality:** 2
**Rating:** 4
**Confidence:** 3

**Summary:**

This paper proposes Variational Uncertainty Decomposition (VUD), a method for separating aleatoric and epistemic uncertainty in in-context learning (ICL) with large language models, where model parameters are inaccessible. The approach uses a variational upper bound on aleatoric uncertainty via optimizable auxiliary inputs, and defines epistemic uncertainty as the residual from total predictive uncertainty. VUD operates without needing posterior sampling, bootstrapping, or fine-tuning, making it scalable and compatible with black-box models. The method is validated through theoretical analysis and empirical results across regression, classification, decision-making, and out-of-distribution detection tasks.

**Questions:**

1. Clarify the practical implementation and optimization of auxiliary inputs

The proposed method relies on optimizing auxiliary inputs $Z$ to estimate a variational upper bound on aleatoric uncertainty. Could the authors provide more detail on how this optimization is performed in practice? Specifically, how are these inputs initialized, what constraints or regularization are applied, and how sensitive is the method to their configuration? Clearer documentation of this component would improve reproducibility and help assess robustness.

2. Explain the assumptions and limitations of approximate exchangeability in prompting

The method assumes approximate exchangeability in in-context examples to justify a Bayesian interpretation of ICL. Could the authors clarify how this assumption holds in practical prompting setups, especially when support examples are ordered or semantically diverse? It would be helpful to understand under what conditions this assumption breaks down, and how the method behaves in such cases.

3. Provide empirical validation of the variational upper bound’s tightness

Theoretically, the method guarantees a variational upper bound on aleatoric uncertainty, but the paper lacks a clear empirical assessment of how tight or accurate this bound is in practice. Could the authors include a quantitative analysis or visualizations to show how well the estimated uncertainty components match ground-truth uncertainty (when available) or known benchmarks?

4. Discuss generalizability across model scales and prompt configurations

The experiments focus on a fixed set of tasks and models. Could the authors comment on how QUD performs across LLMs of different sizes or under prompt variations (e.g., number of in-context examples, style of formatting)? Insight into the method’s generalizability would help evaluate its practical utility and deployment readiness.

5. Expand on downstream value and decision-making use cases

While the method is evaluated on bandit decision-making and out-of-distribution detection, the practical impact of decomposing uncertainty is not deeply explored. Can the authors elaborate on how this decomposition improves real-world outcomes, such as more calibrated decisions or better sample efficiency? A more application-driven perspective would strengthen the significance and practical relevance of the contribution.

**Ethical Concerns:**

["NO or VERY MINOR ethics concerns only"]

**Final Justification:**

The authors provided thoughtful responses that clarify several aspects of their method, including practical strategies for optimizing auxiliary inputs, theoretical justification for approximate exchangeability, and promising downstream applications. However, their explanation lacks key details that limit the overall strength of the rebuttal. Specifically, there is insufficient discussion of regularization and sensitivity in the optimization process, no empirical validation or visual analysis of the variational bound’s tightness, and a lack of systematic evaluation across model sizes and prompt styles. While the theoretical contributions are sound and the potential impact is clear, the absence of concrete empirical support and reproducibility guidance tempers the confidence in the method's robustness and generalizability. Given these considerations, a slight score increase (e.g., to a weak accept) could be justified if the missing components can be reasonably addressed in a camera-ready revision. However, I believe the current weaknesses are significant enough that I will stay with my original borderline accept score.

**Limitations:**

Yes.

**Paper Formatting Concerns:**

No formatting issues are found.

**Quality:**

3

**Strengths And Weaknesses:**

Strengths:

1. The paper proposes a new framework, Variational Uncertainty Decomposition (VUD), that enables decomposition of total predictive uncertainty into aleatoric and epistemic components in in-context learning (ICL) scenarios, where traditional Bayesian inference is infeasible due to the black-box nature of large language models (LLMs).

2. A key innovation is introducing optimizable auxiliary inputs (Z) to estimate an upper bound on aleatoric uncertainty through a variational formulation. This bypasses the need for access to latent parameter posteriors and enables uncertainty quantification directly from black-box LLMs.

3. The paper defines epistemic uncertainty as the difference between total predictive uncertainty and the variational estimate of aleatoric uncertainty, offering a principled and interpretable decomposition without requiring multiple model runs or explicit posterior sampling.

4. To support the Bayesian interpretation of ICL, the authors introduce prompting and optimization techniques that approximate exchangeability, a key assumption for Bayesian analysis. This makes the method more scalable and applicable in real-world LLM settings without modifying model internals.

5. The method is evaluated across regression, classification, bandit decision-making, and OOD detection tasks, with both synthetic and real-world datasets. The paper also provides theoretical guarantees on the variational upper bound and the validity of the decomposition under assumptions, combining empirical rigor with formal soundness.

Weaknesses:

1. The proposed method is theoretically grounded and empirically validated across diverse tasks. However, the work relies on assumptions such as approximate exchangeability and the optimization of auxiliary inputs, which may not generalize well to all prompting formats or model architectures. The paper does not thoroughly analyze the sensitivity or robustness of the method with respect to prompt length, model size, or task type. Also, while theoretical guarantees are provided, empirical validation of these guarantees (e.g., tightness of the variational bound) is limited.

2. The paper is technically sound but occasionally dense, particularly in its presentation of the variational objective and its connection to Bayesian foundations. The role of auxiliary input $Z$ and the practical steps for its optimization are central but underexplained for a general audience. A clearer explanation of the prompting strategy used to achieve approximate exchangeability, along with visual or intuitive illustrations, would significantly improve accessibility.

3. The work addresses an important and underexplored problem: how to estimate and decompose uncertainty in in-context learning with large language models. However, its impact is somewhat constrained by a lack of practical analysis on how this uncertainty decomposition improves downstream decision-making beyond synthetic settings. Broader applications and more user-centric evaluation (e.g., real-world deployment scenarios) would enhance the significance of the contribution.

4. The idea of decomposing uncertainty in ICL via a variational approach is novel, especially in black-box LLM settings where traditional Bayesian methods are inapplicable. However, the use of variational bounds and auxiliary optimization is conceptually inspired by prior work in approximate Bayesian inference. The novelty lies primarily in adapting these tools to the ICL setting rather than introducing entirely new inference techniques.

---

> ### Author Rebuttal · Authors · 2025-07-31
>
> ### Summary Response
>
> We sincerely thank the reviewers for their constructive feedback. We are encouraged that reviewers appreciated the ***novel*** (**bTbz**, **esK3**, **y4yc**) ***principled approach*** (**aVVb**) and ***sound theoretical framework*** (**s63H**) to the ***important and underexplored*** (**y4yc**) problem of uncertainty decomposition in in-context learning.
>
> Our work addresses the fundamental challenge of computing uncertainty decompositions on an intractable and implicitly defined Bayesian model of ICL in LLMs. Specifically:
>
> - We derive an upper bound to the aleatoric uncertainty (conditional entropy/variance) of the Bayesian model.
> - To our knowledge, we are the first achieve this without sampling from the Bayesian model posterior or assumptions on the form of the likelihood, via probing the model with auxiliary inputs.
> - We provide a clear statement of our assumptions and employ techniques such as permutation-based sampling and KL filtering to approximate exchangeability to ensure the validity of our method.
> - Whilst our method is particularly suited to LLMs, it can be applied to any model where there is access to the posterior predictive distribution (p.p.d.).
> - We show quantitative and qualitative performance of VUD across a range of synthetic datasets and real-world downstream tasks.
>
> In response to reviewer suggestions, we have expanded our experimental analysis. New results include:
>
> - Table A (see **aVVb**): Additional experimental OOD tasks for natural language on the MMLU dataset
> - Table B (see **aVVb**): New baseline comparisons using Martingale posterior distributions
> - Table C (see **bTbz**): Analysis of the sample efficiency in the number of samples $Z$
>
> ---
> ### Individual Response
>
> **Question 1**
>
> We investigate 4 optimisation methods in the paper and empirically observe that choosing $Z$ close to or equal to $x^*$ optimises this expression. This is theoretically motivated via the proof of Theorem 3.1 and 3.2 in Appendix A, and analysis on tractable Bayesian models in Appendix B.
>
> In practice we perform one of two strategies for choosing $Z$:
>
> - Perturbing about the predictive task $ x^* $ : For continuous features, we add Gaussian noise (std = 0.1 × std of the covariate in $\mathcal{D} $ and mean= $ x^* $). For categorical inputs, we perturb uniformly across classes. For natural language, we use LLM-based paraphrasing. We find that 5–10 perturbations suffice in most cases; beyond that, improvements are minimal.
> - Repeating $x^*$. This does not require hyperparameter tuning
>
> We also investigated Bayesian optimisation (BO) but found that the number of auxiliary variables required were significantly higher than the perturbing or repeating. Furthermore, the optimal $Z$ for the BO process almost always was a small perturbation from $x^*$, motivating the simpler approaches.
>
> To choose the threshold for KL filtering we set $\epsilon$ by ranking the KL divergences $\epsilon_i$ for each auxiliary datapoint $Z_i$ and choose the $r$-th smallest divergence. Therefore, we guarantee  a decomposition and $r$ becomes a proxy for the level of approximation, which is left to the discretion of the user. We found this allowed for a more robust method because the scale of KL divergences can vary between different tasks.
>
> **Question 2**
>
> Bandit problems are an example of a dataset where the in-context examples have an implicit ordering and the reward variables are not exchangeable (as later rewards have a higher mean). However, since the reward model $r_t(a)$ is stationary, the reward distribution when we restrict to the rewards with action $a$ is exchangeable. This is sufficient for the VUD method to still be applicable for ordered examples. This suggests that the theoretical assumptions for VUD can be loosened, which we would like to further investigate in future work (e.g. non-stationary reward models).
>
> In our natural language experiments, we do not directly measure the semantic diversity [1] and compare the effect of the diversity on the exchangeability of the data. However, we hope to investigate this further in future work, in particular, integrating VUD with methods with methods that measure semantic entropy [2] to decompose uncertainty on predictions with natural language output.
>
>
> **Question 3**
>
> The true aleatoric uncertainty depends on the (unknown) Bayesian prior and likelihood governing the LLM, which are inaccessible in practice. As such, exact ground-truth comparisons are not possible.
>
> However, we can approximate this by comparing against the uncertainty in the known data-generating process in our synthetic and bandit settings. While this serves only as a heuristic, since the LLM’s internal modelling may diverge from the true generative model, it still offers useful insights. For instance, in homoscedastic generative settings, we generally observe that VUD estimates reflect this structure.
>
> Furthermore, under certain assumptions on the Bayesian model we prove the multi-armed bandit (MAB) setting that the tightness of the bound is $\mathcal{O}(n^{-1})$ where $n^{-1}$ is the number of auxiliary examples. Please find the following for the proof:
>
> In Gaussian bandit setting, assume independence of rewards between arms. Furthermore, for an arm $i$, we assume the following model:
> - prior: $p(\theta_i) = \mathcal{N}(0, \sigma_0^2)$
> - likelihood: $p(r_i \mid \theta_i) = \mathcal{N}(\theta_i, \sigma^2)$
>
> Then for $k_t$ observations of rewards from arm $i$, we have:
> - total variance: $\sigma^2 + \left[\frac{1}{\sigma_0^2} + \frac{k_t}{\sigma^2} \right]^{-1}$
> - true aleatoric variance: $\sigma^2$
> - true epistemic variance = total - aleatoric: $\left[\frac{1}{\sigma_0^2} + \frac{k_t}{\sigma^2} \right]^{-1}$
>
> However, with the VUD method, suppose we have $n$ further auxiliary observations, then, the corresponding variational estimate of aleatoric uncertainty is: $\sigma^2 + \left[\frac{1}{\sigma_0^2} + \frac{k_t + n}{\sigma^2} \right]^{-1}$.
>
> This gives the gap between the variational estimate and the exact aleatoric variance as:
> $$\left[\frac{1}{\sigma_0^2} + \frac{k_t + n}{\sigma^2} \right]^{-1} = \mathcal{O}\left(\frac{1}{k_t + n}\right) = \mathcal{O}\left(\frac{1}{n}\right).$$
>
> Note: For non-Gaussian bandits we can apply Bernstein-von Mises to obtain a similar bound.
>
> **Question 4**
>
> We observe that increasing the number of in-context examples leads to a consistent trend: **epistemic uncertainty decreases**, while **aleatoric uncertainty stabilises**, as shown in Figures 7, 9c, 15, and 16 (logistic regression and bandit tasks), which aligns with the intuition for these quantities.
>
> We adopt the standard prompt format used in prior work on Bayesian views of ICL [7]. However, we note the prompt wording can have a conditioning effect on the p.p.d., effectively changing the prior of the LLM, similar to behaviour observed in LLM processes [6]. We believe a systematic analysis of the prompt style effect on model prior is beyond the scope of this project but we agree this is a valuable direction for future work.
>
> Due to computational constraints, we focus on experiments with 7B, 8B and 14B parameter models but we observe that VUD provides sensible uncertainty decompositions across larger model sizes as well. However, if the p.p.d. is poor (e.g. with very small models), the total uncertainty (which is a function of the p.p.d.) can be unreliable, leading to less meaningful decompositions.
>
> **Question 5**
>
> Decomposing uncertainty in LLMs provides greater transparency: this distinction helps practitioners decide when to trust model outputs or seek more information.
>
> Practically, VUD enables more informed data acquisition in settings like Bayesian experimental design (BED), active learning [3] and bandit problems. In these applications, maximizing mutual information (MI) guides which unlabeled samples to label next. VUD offers a tractable lower bound on MI, making it a practical tool for sample-efficient learning. In particular, VUD allows for applications of BED, active learning and bandits in natural language settings [4].
>
> We agree that applying VUD to real-world scenarios is an important next step, and we are pursuing such directions. One such real-world deployment is tabular data. There exists an abundance of tabular electronic health records and financial statements where rows can be serialized into natural language [5]. Thus VUD can be used for OOD detection for ICL where there are limitations on LLM-finetuning budget. Our goal is to allow healthcare practitioners (e.g. Abridge/OpenEvidence) and financial analysts (e.g. BloombergGPT) to identify unreliable model predictions on unfamiliar inputs, improving the trustworthiness of deployed QA systems.
>
> **Other Comments: Visual Illustrations**
>
> > A clearer explanation of the prompting strategy used to achieve approximate exchangeability, along with visual or intuitive illustrations, would significantly improve accessibility.
>
> We will include further visual illustrations of the method and practical advice for optimisation to improve the accessibility for general audience if accepted.
>
>
> ---
>
> [1] Hoffman et al. "Semantic diversity: A measure of semantic ambiguity based on variability in the contextual usage of words." (2013).
>
> [2] Kuhn et al. "Semantic uncertainty: Linguistic invariances for uncertainty estimation in natural language generation." (2023).
>
> [3] Rainforth et al. "Modern Bayesian experimental design." (2024).
>
> [4] Margatina et al. "Active learning principles for in-context learning with large language models." (2023).
>
> [5] Hegselmann et al. "Tabllm: Few-shot classification of tabular data with large language models." (2023).
>
> [6] Requeima et al. "Llm processes: Numerical predictive distributions conditioned on natural language." *Advances in Neural Information Processing Systems* 37 (2024)
>
> [7] Xie et al. "An explanation of in-context learning as implicit bayesian inference." (2021).

---

> > ### Comment · Reviewer_y4yc · 2025-08-05
> > **Rebuttal Review**
> >
> > The authors provided thoughtful responses that clarify several aspects of their method, including practical strategies for optimizing auxiliary inputs, theoretical justification for approximate exchangeability, and promising downstream applications. However, their explanation lacks key details that limit the overall strength of the rebuttal. Specifically, there is insufficient discussion of regularization and sensitivity in the optimization process, no empirical validation or visual analysis of the variational bound’s tightness, and a lack of systematic evaluation across model sizes and prompt styles. While the theoretical contributions are sound and the potential impact is clear, the absence of concrete empirical support and reproducibility guidance tempers the confidence in the method's robustness and generalizability. Given these considerations, a slight score increase (e.g., to a weak accept) could be justified if the missing components can be reasonably addressed in a camera-ready revision. However, I believe the current weaknesses are significant enough that I will stay with my original borderline accept score.

---

> ### Author Response · Authors · 2025-08-07
>
> We thank the reviewer for their reply and suggestions. The following delineates our reply:
>
> - We have included empirical evidence for the tightness of bound in Table C (see **Comments: Sample Complexity of Auxiliary Variables** in Reviewer s63h’s rebuttal which was referred to in our Summary Response). We cannot include visualisations due to the restrictions in the rebuttals process.
> - In our experiments, we illustrate the effectiveness of VUD across 7B, 8B and 14B Llama and Qwen model sizes.
> - Our prompts are designed according to standard prompt engineering practices [1,2,3]
>
> We respectfully disagree regarding the comments on “empirical support” and "generalisability". Our method is the first of its kind within the Bayesian uncertainty estimation literature, and as such, there are few directly comparable baseline methods available. While we acknowledge our language models are not of super large scale e.g. 70B models due to computational constraints, we believe our paper has shown promise in terms of the wide applicability of our methods for various types of tasks. This includes simple data science tasks, decision making (bandits), OOD detection in NLP and answer rejection (see latest **aVVb** rebuttal) which is sufficient to motivate future research on scalability.
>
> ---
>
> [1] OpenAI. “Prompt Engineering.” *OpenAI Developer Documentation*, [platform.openai.com/docs/guides/prompt‑engineering](http://platform.openai.com/docs/guides/prompt%E2%80%91engineering).
>
> [2] Touvron, Hugo, et al. "Llama: Open and efficient foundation language models." *arXiv preprint arXiv:2302.13971* (2023).
>
> [3] Yang, An, et al. "Qwen3 technical report." *arXiv preprint arXiv:2505.09388* (2025).

---

### Official Review · Reviewer_bTbz · 2025-07-04

**Clarity:** 3
**Significance:** 4
**Originality:** 3
**Rating:** 5
**Confidence:** 2

**Summary:**

The paper introduces a uncertainty decomposition method, variational uncertainty decomposition (VUD), that leverages in-context examples for uncertainty decomposition of LLMs. Motivated by a Bayesian interpretation of ICL, the authors propose a method that estimates aleatoric and epistemic uncertainties without requiring explicit posterior sampling over latent parameters. Instead, the approach uses optimisable auxiliary inputs (Z) to provide a variational upper bound on aleatoric uncertainty, with the residual treated as epistemic uncertainty.

**Questions:**

N/A

**Ethical Concerns:**

["NO or VERY MINOR ethics concerns only"]

**Final Justification:**

This paper already undergoes sufficient discussion. After actively engaging in the discussion with the authors and reading the opinion from other reviewer, I am pretty confidence that my judgement is correct and recommend the acceptance of this paper.

**Limitations:**

yes

**Quality:**

3

**Strengths And Weaknesses:**

Strength:

- The proposed method is very interesting and novel. The experiment on QA tasks are especially interesting to me. However, I am not an expert in math/bandit and cannot justify the validity of the proposed method. But the experiement results in Table 2 looks very promising.



Weakness:

- On the definition of in-distribution and out-of-distribution examples in QA experiments. These QA datasets are mainly about world knowledge. However, existing LLMs undergo large-scale pre-training and already store sufficient knowledge for most of these questions. In such a context, how you define out-of-distribution? It is unclear how in-context examples can help train these models or help them acquire additional knowledge such that some testing questions are in-distribution with respect to these in-context examples. The only in-distribution knowledge is the format, i.e., the model needs to output 0 or 1 to represent no or yes.
- How Deep Ensembles is implemented in the QA setting? From my understanding, each set of in-context examples will give us a output distribution of the model, and by taking the average of that distribution over all sets of in-context examples, we can compute the entropy of the mean; then, we compute the mean entropy of each individual output distribution, which will be the aleatoric uncertainty; by subtract the aleatoric uncertainty from the entropy of mean, we get the epistemic uncertainty. Is that the implementation of Deep Ensembles?
- Another concerns is on the decomposition of uncertainty and the source of aleatoric uncertainty. For these QA tasks, what is the source of aleatoric uncertainty? The only explanation of aleatoric uncertainty in these QA tasks, from my perspective, is the question ambiguity. But does there really exist ambigous questions in the test set, since these datasets should be considered as high-quality datasets and should be clear enough? If not, what is the source of the two types of uncertainty?
- Finally, is in-context learning-based UQ method still useful for QA tasks? To the best of my knowledge, existing LLMs after SFT/post-training (not the pre-trained ones) can already follow human instructions and does not need in-context examples to answer such knowledge-intensive tasks. Even sometimes we provide them, these examples are only used for format clarification, instead of train these LLMs or equip them with additional world knowledge. Will the proposed methods be too "toy" for the actual applications of existing LLMs?

Again, the reviewer find this paper interesting and impressive. Once the other reviewers that are more experienced in the math deriviations can help verify the correctness and validity of this paper, I would like to see this paper being accpected.

---

> ### Author Rebuttal · Authors · 2025-07-30
>
> ### Summary Response
>
> We sincerely thank the reviewers for their constructive feedback. We are encouraged that reviewers appreciated the ***novel*** (**bTbz**, **esK3**, **y4yc**) ***principled approach*** (**aVVb**) and ***sound theoretical framework*** (**s63H**) to the ***important and underexplored*** (**y4yc**) problem of uncertainty decomposition in in-context learning.
>
> Our work addresses the fundamental challenge of computing uncertainty decompositions on an intractable and implicitly defined Bayesian model of ICL in LLMs. Specifically:
>
> - We derive an upper bound to the aleatoric uncertainty (conditional entropy/variance) of the Bayesian model.
> - To our knowledge, we are the first achieve this without sampling from the Bayesian model posterior or assumptions on the form of the likelihood, via probing the model with auxiliary inputs.
> - We provide a clear statement of our assumptions and employ techniques such as permutation-based sampling and KL filtering to approximate exchangeability to ensure the validity of our method.
> - Whilst our method is particularly suited to LLMs, it can be applied to any model where there is access to the posterior predictive distribution.
> - We demonstrate both quantitative and qualitative performance of VUD across a range of synthetic datasets and real-world downstream tasks.
>
> In response to reviewer suggestions, we have expanded our experimental analysis. New results include:
>
> - Table A (see **aVVb**): Additional experimental OOD tasks for natural language on the MMLU dataset
> - Table B (see **aVVb**): New baseline comparisons using Martingale posterior distributions
> - Table C (see **bTbz**): Analysis of the sample efficiency in the number of samples $Z$
>
> We provide detailed responses to individual reviewer comments and questions below.
>
> ---
>
> ### Individual Response
>
> **Comments: LLM’s Pretrained Knowledge and Motivation behind ICL based UQ**
>
> > On the definition of in-distribution and out-of-distribution examples in QA experiments. These QA datasets are mainly about world knowledge. However, existing LLMs undergo large-scale pre-training and already store sufficient knowledge for most of these questions. In such a context, how you define out-of-distribution? It is unclear how in-context examples can help train these models or help them acquire additional knowledge such that some testing questions are in-distribution with respect to these in-context examples. The only in-distribution knowledge is the format, i.e., the model needs to output 0 or 1 to represent no or yes.
>
> > Finally, is in-context learning-based UQ method still useful for QA tasks? To the best of my knowledge, existing LLMs after SFT/post-training (not the pre-trained ones) can already follow human instructions and does not need in-context examples to answer such knowledge-intensive tasks. Even sometimes we provide them, these examples are only used for format clarification, instead of train these LLMs or equip them with additional world knowledge. Will the proposed methods be too "toy" for the actual applications of existing LLMs?
>
>
> The following outlines the motivation of ICL based UQ:
>
> - Existing LLMs undergo large-scale pre-training. However, there exists private, sensitive, inaccessible data such as electronic health records.
> - Serving the top performing open-sourced LLMs [1] have incredibly high compute requirements, necessitating at least 8 80GB A100 GPUs. Likewise for LLM-fine tuning. For cheaper LLMs, they often have much poorer performance [2] (~65% on MMLU for Llama3-8B). Thus, ICL alleviates users with computational constraints.
> - The most popular LLMs such as ChatGPT, Claude and Gemini are proprietary models. These models are closed-source and in a black-box setting which prevents practitioners from accessing their weights to apply traditional Bayesian deep learning techniques. Our ICL based UQ method provides a mechanism to elicit and quantify uncertainty since these proprietary models like ChatGPT and Claude provide access to output logits.
> - We find that formatting the output from LLMs is crucial. Even if the model knows the answer, it needs to express it in the expected form: "yes" vs. "no", "0" vs. "1", "The answer is: True" vs. "False". In our QA experiments, the model is expected to output 0/1 labels. If not prompted clearly, it may hallucinate and output varying text, leading to inconsistent output logits.
>
> **Comments: Deep Ensembles Implementation**
>
> > How Deep Ensembles is implemented in the QA setting? From my understanding, each set of in-context examples will give us a output distribution of the model, and by taking the average of that distribution over all sets of in-context examples, we can compute the entropy of the mean; then, we compute the mean entropy of each individual output distribution, which will be the aleatoric uncertainty; by subtract the aleatoric uncertainty from the entropy of mean, we get the epistemic uncertainty. Is that the implementation of Deep Ensembles?
> >
>
> Yes, this interpretation is correct. To further clarify, instead of training multiple neural networks (traditional Deep Ensembles), we:
>
> 1. Sample different in-context example sets ($D_k$) for each ensemble member.
>
> - Each $D_k$ contains randomly chosen few-shot examples from the training data.
>
> 2. Query the same LLM with each $D_k$ to get predictions $p_k(y|x, D_k)$.
>
> - The LLM generates a probability distribution over labels for each prompt.
>
> 3. Aggregate predictions i.e. taking the mean across $D_k$ and computing the entropy (total uncertainty) as well as the mean entropy of each ensemble distribution (aleatoric uncertainty) to estimate epistemic uncertainty.
>
> - **Total Uncertainty (TU)**: Entropy of the mean prediction $\left(H\left(\frac{1}{K}\sum_k p_k\right)\right)$.
> - **Aleatoric Uncertainty (Va)**: Mean entropy of individual predictions $\left(\frac{1}{K}\sum_k=H(p_k)\right)$
> - **Epistemic Uncertainty (Ve)**: $TU - Va$ (disagreement due to $D_k$ variability).
>
> **Comments: QA Task Source of Aleatoric Uncertainty**
>
> > Another concerns is on the decomposition of uncertainty and the source of aleatoric uncertainty. For these QA tasks, what is the source of aleatoric uncertainty? The only explanation of aleatoric uncertainty in these QA tasks, from my perspective, is the question ambiguity. But does there really exist ambiguous questions in the test set, since these datasets should be considered as high-quality datasets and should be clear enough? If not, what is the source of the two types of uncertainty?
>
> High-quality datasets can still contain intrinsic aleatoric uncertainty. As quoted from BoolQ [3] “the authors labelled 110 randomly chosen examples. If there was a disagreement, the authors conferred and selected a single answer by mutual agreement. We call the resulting labels “gold-standard” labels. On the 110 selected examples, the answer annotations reached 90% accuracy compared to the gold-standard labels”. Similarly, the same can be said about HotpotQA [4] where there is a **~**13 EM points gap between the Human UB and Human gap in Table 8, reflecting aleatoric uncertainty in the questions. Lastly, PubMedQA [5] includes "maybe" answers to model uncertainty that accounts for 11% of the PQA-L(abeled) dataset.
>
> ---
>
> [1] Chiang, Wei-Lin, et al. "Chatbot arena: An open platform for evaluating llms by human preference." *Forty-first International Conference on Machine Learning*. 2024.
>
> [2] Yang, An, et al. "Qwen3 technical report." arXiv preprint arXiv:2505.09388 (2025).
>
> [3] Clark, Christopher, et al. "Boolq: Exploring the surprising difficulty of natural yes/no questions." *arXiv preprint arXiv:1905.10044* (2019).
>
> [4] Yang, Zhilin, et al. "HotpotQA: A dataset for diverse, explainable multi-hop question answering." *arXiv preprint arXiv:1809.09600* (2018).
>
> [5] Jin, Qiao, et al. "Pubmedqa: A dataset for biomedical research question answering." *arXiv preprint arXiv:1909.06146* (2019).

---

> > ### Comment · Reviewer_bTbz · 2025-08-06
> >
> > I appreciate the authors' detailed response. My questions are fully discussed. I will keep my rating and recommend this paper to be accepted.

---

### Official Review · Reviewer_s63h · 2025-07-06

**Clarity:** 3
**Significance:** 2
**Originality:** 2
**Rating:** 3
**Confidence:** 4

**Summary:**

The paper introduces a uncertainty decomposition framework for in-context learning (ICL) with LLMs, proposing a method to estimate and decompose predictive uncertainty into aleatoric and epistemic components without requiring posterior sampling over latent parameters. The authors optimize over auxiliary inputs to upper-bound aleatoric uncertainty and derive epistemic uncertainty as the residual. The method is theoretically grounded and designed to approximate Bayesian behavior under certain assumptions. Experimental results on synthetic datasets, bandit problems, and question-answering tasks show that the decomposed uncertainties align well with intuitive expectations.

**Questions:**

1. Since the framework allows auxiliary input and LLM inference is often costly, sample complexity becomes an important consideration. It would be helpful if the authors could discuss how the number of auxiliary samples affects the quality of uncertainty quantification. Specifically, how does performance scale with more auxiliary inputs, and is there any theoretical or empirical guidance on the minimal number of points needed to achieve reliable decomposition?


2. In the bandit setting, what auxiliary information is used to support the uncertainty decomposition? I checked Appendix F.3 but couldn’t find a clear description. Additionally, while the paper presents relevant metrics and graphs, it remains unclear how the proposed uncertainty quantification framework is concretely integrated into the bandit decision-making process. Could the authors clarify how the uncertainty estimates are used to guide action selection in practice?

**Ethical Concerns:**

["NO or VERY MINOR ethics concerns only"]

**Final Justification:**

The authors addressed the most part of my concern, therefore I decide to increase my rating.

**Quality:**

3

**Strengths And Weaknesses:**

Strength: This paper is well-written and presents a sound theoretical framework. The underlying problem is timely and important in the current AI landscape.

Weakness: The major weakness is that, upon careful reading of the theoretical framework, it seems the role of the LLM is not tightly integrated into the core uncertainty quantification method. With access to auxiliary information for $U$ and $Z$, the proposed approach could be applied to a broad class of Bayesian models, not necessarily tied to in-context learning or LLMs. Another concern is that the assumed access to auxiliary data may be overly strong. In the experiments, the model is allowed to query points and responses near the test input $x$ and target $y$ which could enable many alternative methods to perform effective uncertainty quantification as well. The work would benefit from a more nuanced discussion of why the proposed method is non-trivial.

---

> ### Author Rebuttal · Authors · 2025-07-30
>
> ### Summary Response
>
> We sincerely thank the reviewers for their constructive feedback. We are encouraged that reviewers appreciated the ***novel*** (**bTbz**, **esK3**, **y4yc**) ***principled approach*** (**aVVb**) and ***sound theoretical framework*** (**s63H**) to the ***important and underexplored*** (**y4yc**) problem of uncertainty decomposition in in-context learning.
>
> Our work addresses the fundamental challenge of computing uncertainty decompositions on an intractable and implicitly defined Bayesian model of ICL in LLMs. Specifically:
>
> - We derive an upper bound to the aleatoric uncertainty (conditional entropy/variance) of the Bayesian model.
> - To our knowledge, we are the first achieve this without sampling from the Bayesian model posterior or assumptions on the form of the likelihood, via probing the model with auxiliary inputs.
> - We provide a clear statement of our assumptions and employ techniques such as permutation-based sampling and KL filtering to approximate exchangeability to ensure the validity of our method.
> - Whilst our method is particularly suited to LLMs, it can be applied to any model where there is access to the posterior predictive distribution.
> - We demonstrate both quantitative and qualitative performance of VUD across a range of synthetic datasets and real-world downstream tasks.
>
> In response to reviewer suggestions, we have expanded our experimental analysis. New results include:
>
> - Table A (see **aVVb**): Additional experimental OOD tasks for natural language on the MMLU dataset
> - Table B (see **aVVb**): New baseline comparisons using Martingale posterior distributions
> - Table C (see **bTbz**): Analysis of the sample efficiency in the number of samples $Z$
>
> We provide detailed responses to individual reviewer comments and questions below.
>
> ---
>
> ### Individual Response
>
> **Comments: Significance of Method**
>
> > The major weakness is that, upon careful reading of the theoretical framework, it seems the role of the LLM is not tightly integrated into the core uncertainty quantification method. With access to auxiliary information for $U$ and $Z$, the proposed approach could be applied to a broad class of Bayesian models, not necessarily tied to in-context learning or LLMs.
>
> > In the experiments, the model is allowed to query points and responses near the test input $x$ and target $y$ which could enable many alternative methods to perform effective uncertainty quantification as well. The work would benefit from a more nuanced discussion of why the proposed method is non-trivial.
>
>
> We agree that the proposed method is general and can be applied to a broad class of Bayesian models and indeed, in Appendix B, we theoretically verify the validity of our approach on Bayesian linear regression and Gaussian processes. However, our method is motivated in-context learning in LLMs, where the Bayesian model is implicitly defined and we **do not have access to** the likelihood $p(x|\theta )$ or posterior $p(\theta|\mathcal{D})$ or samples of $\theta$, but we **do have access to** the posterior predictive distribution $p(y^* |x^* , \mathcal{D} )$. To our knowledge, we are the only method to provide a bound on the conditional entropy of an implicitly defined Bayesian model given these restrictions.
>
> Furthermore, we tailor our approach to LLMs by using practical techniques to encourage exchangeability in model generations which are grounded by clear theoretical motivations.
>
> Finally, the most popular LLMs such as ChatGPT, Claude and Gemini are proprietary models. These models are closed-source and in a black-box setting which prevents practitioners from accessing their weights to apply traditional Bayesian deep learning techniques. However, our ICL based UQ method provides a mechanism to elicit and quantify  Bayesian uncertainty, since these black-box models like ChatGPT and Claude provide access to output logits and the query of points near the test input, allowing our method to be applied to a broad class of proprietary models.
>
> **Comments: Sample Complexity of Auxiliary Variables**
>
> > How does performance scale with more auxiliary inputs, and is there any theoretical or empirical guidance on the minimal number of points needed to achieve reliable decomposition?
> >
>
> In our experiments, we use up to 15 auxiliary variables, if not fewer, for more computationally intensive tasks such as regression. We find empirically that 5-10 auxiliary variables is sufficient for a reliable decomposition with diminishing results beyond that. We include an example of bounds per number of auxiliary examples in Table C below.
>
> **Table C: Sample Complexity (Logistic Regression)**
>
> | Row ($x$ value); Col (num $Z$ ) | 5 | 10 | 15 | 20 | 30 |
> | --- | --- | --- | --- | --- | --- |
> | -10 | 0.177 (0.033) | **0.158 (0.024)** | 0.164 (0.002) | 0.201 (0.021) | 0.209 (0.000) |
> | -5 | 0.227 (0.033) | **0.210 (0.034)** | 0.249 (0.001) | 0.250 (0.001) | 0.252 (0.002) |
> | 0 | 0.838 (0.020) | **0.826 (0.016)** | 0.849 (0.010) | 0.860 (0.008) | 0.868 (0.003) |
> | 5 | 0.235 (0.016) | 0.224 (0.018) | 0.218 (0.018) | 0.212 (0.016) | **0.208 (0.014)** |
> | 10 | 0.173 (0.019) | **0.164 (0.019)** | 0.173 (0.018) | 0.182 (0.007) | 0.185 (0.001) |
>
> Developing theoretical guarantees for sample complexity is challenging without further assumptions on the particular prediction tasks as we don’t have access to the ground truth or Bayesian model. Nevertheless, under certain assumptions on the Bayesian model we prove for the multi-armed bandit setting that the tightness of the bound is $\mathcal{O}(n^{-1})$ where $n^{-1}$ is the number of auxiliary examples (via an application of the Bernstein-von-Mises theorem). Please find the following for the proof for the Gaussian bandit setting:
>
> In Gaussian bandit setting, assume independence of rewards between arms. Furthermore, for an arm $i$, we assume the following model:
> - prior: $p(\theta_i) = \mathcal{N}(0, \sigma_0^2)$
> - likelihood: $p(r_i \mid \theta_i) = \mathcal{N}(\theta_i, \sigma^2)$
>
> Then for $k_t$ observations of rewards from arm $i$, we have:
> - total variance: $\sigma^2 + \left[\frac{1}{\sigma_0^2} + \frac{k_t}{\sigma^2} \right]^{-1}$
> - true aleatoric variance: $\sigma^2$
> - true epistemic variance = total - aleatoric: $\left[\frac{1}{\sigma_0^2} + \frac{k_t}{\sigma^2} \right]^{-1}$
>
> However, with the VUD method, suppose we have $n$ further auxiliary observations, then, the corresponding variational estimate of aleatoric uncertainty is: $\sigma^2 + \left[\frac{1}{\sigma_0^2} + \frac{k_t + n}{\sigma^2} \right]^{-1}$.
>
> This gives the gap between the variational estimate and the exact aleatoric variance as:
> $$\left[\frac{1}{\sigma_0^2} + \frac{k_t + n}{\sigma^2} \right]^{-1} = \mathcal{O}\left(\frac{1}{k_t + n}\right) = \mathcal{O}\left(\frac{1}{n}\right).$$
>
> Note: For non-Gaussian bandits we can apply Bernstein-von Mises to obtain a similar bound.
>
> **Comments: Bandit Clarifications**
>
> > In the bandit setting, what auxiliary information is used to support the uncertainty decomposition? …  Could the authors clarify how the uncertainty estimates are used to guide action selection in practice?
> >
>
> For the “Buttons” multi-armed bandit task, the task is to predict the reward $r_t(a)$ for an action $a$ and prior action reward pairs $ \mathcal{D}_ t = \{ (a_i , r_i ) \}_{ i = 1 }^{ t - 1 }$. In this setting $x$ is $a$ and $y$ is $r_t(a)$. For computational efficiency, we use the repeated $Z$ strategy: so for a particular action $a$, the auxiliary data would be $Z=a$ (so there is only 1 auxiliary data point per action). We have to do uncertainty decompositions for each action in the action space $\mathcal{A}$.
>
> Once we have uncertainty estimates $U_t(a)$ for each action and expected reward estimates $Q_t (a) \approx \mathbb{E} [r_t |a, \mathcal{D}_ t ]$, we compute the UCB score for each action, $\mathrm{UCB}_ t(a)=Q_t (a)+c U_t (a)$ where $c$ is the exploration constant (a hyperparameter we can choose). Then, the next action is the action that has the minimal UCB score $a_t =\mathrm{ argmin } _{ a \in \mathcal{A} } \{ \mathrm{ UCB } _ t (a) \} $. (In the situation that multiple actions have the same UCB score, we choose uniformly from these minimising actions).

---

> ### Comment · Reviewer_s63h · 2025-08-05
> **Response to Authors**
>
> Thank you for your response. My primary concern is that I still don’t fully sure if this "auxiliary information" mechanism would smoothly integrate into the LLM inference framework. In many cases, requiring additional details for individual data points seems impractical, which makes me question its feasibility.
>
> Say in the bandit case, according to "for a particular action $a$, the auxiliary data would be $Z=a$ (so there is only 1 auxiliary data point per action)", if the auxiliary information is the action $a$ itself, wouldn't it just be estimating the confidence bound, which is just the UCB algorithm itself? If so, how does this UQ framework differ from or extend beyond UCB?

---

> ### Author Response · Authors · 2025-08-05
> **Response to Reviewer Questions**
>
> Dear Reviewer s63h,
>
> Thank you for your reply. Uncertainty decomposition in black-box models is inherently difficult, as many proprietary models restrict access to internal representations and expose only output logits. Therefore, traditional Bayesian approaches are infeasible and many uncertainty quantification methods in this domain [1,2] use “auxiliary information”, for instance, in the form of clarifications or prompt modifications to probe the uncertainty of the model. Our method extends these prior works by introducing a rigorous Bayesian framework that allows for a Bayesian decomposition of the uncertainty, even under black-box constraints.
>
> We demonstrate that generating auxiliary inputs is both practical and natural within the LLM inference workflow. In our out-of-detection experiments (Section 5.2), we smoothly integrate “auxiliary information” into the LLM inference framework via “rephrasing” the question-answer tasks in the natural language datasets such as BoolQA, HotpotQA and PubMedQA. Specifically, we provide our prompt template below to generate “additional details for the individual data points”. Further examples are given in Appendix H.3.
>
> ```python
> prompt = (
>     "Please rephrase the following:\n\n"
>     f"{z_row['note']}\n\n"
>     "While rephrasing the above, incorporate context from the following and make sure it's intertwined/interconnected:\n\n"
>     f"{x_row['note']}\n\n"
>     "Use the following format when rephrasing:\n\n"
>     "<rep> Question: {{Rephrased Question}}? Context: {{Rephrased Context}}. </rep>"
> )
> ```
>
> We find that the nature of “rephrasing” is an intuitive way to generate auxiliary inputs for these QA tasks. The “rephrasing” concept has also been used in other non-Bayesian LLM uncertainty decomposition works including Input Clarification [1].
>
> To address your query regarding bandit problems, the goal is to approximate the epistemic variance of the reward, an unobserved quantity needed to construct confidence bounds for UCB-style algorithms. VUD enables us to approximate these bounds using auxiliary inputs (for instance perturbations or repeating the input), even when we lack access to the Bayesian model. Indeed, as motivated by the proof of Theorem 3.1, setting $Z\approx x$ is an intuitive approach to minimise the KL divergence between $p(\theta | y^* , x^* ,U,Z, \mathcal{D} )$ and $p( \theta |U,Z, \mathcal{D} )$ which is the discrepency between the variational estimate of epistemic uncertainty and the ground truth epistemic uncertainty.
>
> However, it is important to emphasise here that the Bayesian model here is defined **implicitly**. Since, we do not assume the form of the likelihood, our approach does not reduce to the standard UCB-1 algorithm. Instead, **our UQ framework allows LLMs to be integrated with UCB-style algorithms** which can be particularly useful in bandit problems where the context or actions are in natural language [3].
>
> Please feel free to let us know if there’s anymore questions and we will be happy to clarify further.
>
> ---
>
> [1] Hou, Bairu, et al. "Decomposing uncertainty for large language models through input clarification ensembling." *arXiv preprint arXiv:2311.08718* (2023).
>
> [2] Chen, Tiejin, et al. "Uncertainty Quantification of Large Language Models through Multi-Dimensional Responses." *arXiv preprint arXiv:2502.16820* (2025).
>
> [3] Bouneffouf, Djallel, and Raphael Feraud. "Multi-Armed Bandits Meet Large Language Models." *arXiv preprint arXiv:2505.13355* (2025).

---

### Official Review · Reviewer_aVVb · 2025-07-07

**Clarity:** 3
**Significance:** 2
**Originality:** 3
**Rating:** 4
**Confidence:** 3

**Summary:**

This paper tackles the problem of uncertainty decomposition in large language models for the in-context learning (ICL) task, based on the recent hypothesis that performing ICL using large language models (LLMs) is akin to generating from the posterior predictive distribution. The authors provide a theoretical framework to try and tackle two main components required for uncertainty decomposition of this Bayesian posterior predictive: exchangeability and posterior sampling. The authors introduce a theoretical framework by introducing additional optimizable parameters that help tackle the first problem approximately and provide an optimizable upper bound on aleatoric uncertainty. The authors also present techniques to generate the necessary auxiliary parameters to improve the tightness of the bound to generate desirable properties such as well calibrated epistemic uncertainty. The authors also present experimental results on synthetic regression and classification datasets.

**Questions:**

- Can you comment on how we would improve the practical adoption of this approach from a perspective of test time scaling? Existing field of Bayesian deep learning has multiple approaches such as knowledge distillation [1], etc. to help, but wondering what would you suggest to improve the test time computation.

- The paper would also benefit from comparison with additional approaches.  If we were to borrow from the literature of Bayesian Deep Learning again, there are techniques such as MC Dropout[2], BatchEnsembles [3], etc. I am also open to hearing from the authors on what other baselines they have in mind.

- Can the authors comment on the difficulty on optimizing the new parameters introduced in this approach? It'll be great to also have a profiling view of the different steps involved per experiment and model.

- Can the authors comment on how we might extend this approach to new datasets/tasks (such as MMLU)?

References

[1] Vadera, M., Jalaian, B. and Marlin, B., 2020, August. Generalized bayesian posterior expectation distillation for deep neural networks. In Conference on Uncertainty in Artificial Intelligence (pp. 719-728). PMLR.

[2] Gal, Y. and Ghahramani, Z., 2016, June. Dropout as a Bayesian approximation: representing model uncertainty in deep learning. In Proceedings of the 33rd International Conference on International Conference on Machine Learning-Volume 48 (pp. 1050-1059).

[3] Wen, Y., Tran, D., & Ba, J. BatchEnsemble: an Alternative Approach to Efficient Ensemble and Lifelong Learning. In International Conference on Learning Representations.

**Ethical Concerns:**

["NO or VERY MINOR ethics concerns only"]

**Final Justification:**

The authors engaged thoughtfully in the rebuttal process and provided add'l experiment results that show benefits of their approach on existing benchmarks of MMLU-Moral and Computer science.

**Limitations:**

yes

**Quality:**

3

**Strengths And Weaknesses:**

Strengths

- The paper provides a principled approach to uncertainty decomposition in ICL by introducing variational bounds that alleviate the intractable posterior sampling problem. I really liked the fact that the authors have built their approach

- The paper is also well positioned and motivated given the interest in understanding the uncertainties arising from deploying LLMs to critical applications

- The authors have done a good job describing their work in the context of existing literature. I also appreciate the fact that the authors have clearly mentioned the underlying assumptions as they relate to the interpretation of ICL as computing posterior predictive distribution.


Weaknesses

- I think the paper would greatly benefit if it were to include experiments from additional tasks (such as answering on MMLU dataset for instance). Having experimental results on synthetic datasets is useful for visualization and initial understanding, but the paper definitely needs additional experiments to support the application perspective.

- While I appreciate the fact that the authors have clearly written down the assumptions under which the Bayesian framework can be attributed to ICL, this can be considered a potential shortcoming of the overall idea of Bayesian framework as an approximation to ICL. While this is not necessarily a critique of this existing paper, but it's a critique of the broader idea and as a result it does impact the significance of this work.

- When it comes to practical application, this method would add additional complexity in terms of finding the appropriate $Z$, and performing multiple forward passes.

---

> ### Author Rebuttal · Authors · 2025-07-30
>
> ### Summary Response:
>
> We sincerely thank the reviewers for their constructive feedback. We are encouraged that reviewers appreciated the ***novel*** (**bTbz**, **esK3**, **y4yc**) ***principled approach*** (**aVVb**) and ***sound theoretical framework*** (**s63H**) to the ***important and underexplored*** (**y4yc**) problem of uncertainty decomposition in in-context learning.
>
> Our work addresses the fundamental challenge of computing uncertainty decompositions on an intractable and implicitly defined Bayesian model of ICL in LLMs. Specifically:
>
> - We derive an upper bound to the aleatoric uncertainty (conditional entropy/variance) of the Bayesian model.
> - To our knowledge, we are the first achieve this without sampling from the Bayesian model posterior or assumptions on the form of the likelihood, via probing the model with auxiliary inputs.
> - We provide a clear statement of our assumptions and employ techniques such as permutation-based sampling and KL filtering to approximate exchangeability to ensure the validity of our method.
> - Whilst our method is particularly suited to LLMs, it can be applied to any model where there is access to the posterior predictive distribution.
> - We demonstrate both quantitative and qualitative performance of VUD across a range of synthetic datasets and real-world downstream tasks.
>
> In response to reviewer suggestions, we have expanded our experimental analysis. New results include:
>
> - Table A (see **aVVb**): Additional experimental OOD tasks for natural language on the MMLU dataset
> - Table B (see **aVVb**): New baseline comparisons using Martingale posterior distributions
> - Table C (see **bTbz**): Analysis of the sample efficiency in the number of samples $Z$
>
> We provide detailed responses to the reviewer comments and questions below.
>
> ---
>
> ### Individual Response:
>
> **Comments: MMLU**
>
> > Can the authors comment on how we might extend this approach to new datasets/tasks (such as MMLU)?
> >
>
> We conduct an out-of-distribution experiment using Qwen/Qwen2.5-14B-Instruct between MMLU-CS computer science ("college_computer_science", "high_school_computer_science”) and MMLU-M moral (”moral_scenarios”) questions from the MMLU dataset.
>
> **Table A: OOD on MMLU dataset**
>
> |  |  | Deep Ensembles (AUC) |  | Ours (AUC) |  |
> | --- | --- | --- | --- | --- | --- |
> | ID/OOD |  | MMLU-CS | MMLU-M | MMLU-CS | MMLU-M |
> | MMLU-CS | TU | - | **0.302 (.001)** | - | 0.337 (.000) |
> |  | EU | - | 0.178 (.000) | - | **0.521 (.001)** |
> | MMLU-M | TU | 0.376 (.000) | - | 0.397 (.000) | - |
> |  | EU | **0.391 (.001)** | - | **0.502 (.001)** | - |
>
> VUD outperforms Deep Ensembles considerably. We aim to include further MMLU subtopic comparisons in our paper.
>
> **Comments: Bayesian framework assumptions**
>
> > While I appreciate the fact that the authors have clearly written down the assumptions under which the Bayesian framework can be attributed to ICL, this can be considered a potential shortcoming of the overall idea of Bayesian framework as an approximation to ICL. While this is not necessarily a critique of this existing paper, but it's a critique of the broader idea and as a result it does impact the significance of this work.
> >
>
> In this work, we choose to build upon the intuition of a Bayesian framework for ICL [1]. In particular, we explicitly state assumptions as a rigorous way to provide a solid foundation to promote further Bayesian methods in this domain. However, to encourage practical adoption of this method we propose techniques to promote exchangeability and bridge the gap between theory and practice. In this way, our contribution lies both in articulating a principled Bayesian theory of ICL and in developing practical methods to make that interpretation more applicable in real-world settings.
>
> **Comments: Test Time Scaling**
>
> > Can you comment on how we would improve the practical adoption of this approach from a perspective of test time scaling?
> >
>
> To leverage test-time scaling [3], variational estimates for aleatoric uncertainty (steps 2-4 in Fig. 3) in can be computed in parallel across a range of auxiliary variables $Z_i$. This may allow to further optimisation of the variational lower bound, for instance by considering pairs of auxiliary variables $(Z_i, Z_j)$. An example of a greedy procedure is outlined below:
>
> 1. **Parallel Evaluation**: compute $V_a (y^* |x^*, \mathcal{D} ,Z_i)$ for multiple $Z_i$, where each $Z_i$ is a single datapoint.
> 2. **Initial Filtering**: Discard candidates that fail to meet a KL-divergence threshold
> 3. **Combinatorial Expansion**: Form combinations (e.g. ordered pairs $(Z_j, Z_k)$) from the remaining candidates and compute $V_a (y^* | x^* , \mathcal{D} ,(Z_j,Z_k))$
> 4. **Secondary Filtering**: Apply the KL condition again to filter out weak combinations.
> 5. **Selection**: Choose the optimal set based on the resulting $V_a$ score
>
> This simple scheme highlights how increased test-time compute can be used to improve the quality of the uncertainty estimates, a potential direction for future work.
>
> **Comments: Test Time Compute**
>
> > Existing field of Bayesian deep learning has multiple approaches such as knowledge distillation, etc. to help, but wondering what would you suggest to improve the test time computation.
> >
>
> We agree that test-time scaling is an important consideration for practical adoption. One avenue to improve the test-time scaling of this method is that, when computing decompositions for a large set of predictions $X$ given the same in-content learning dataset $\mathcal{D}$, it is possible to re-use the auxiliary variables $Z$ for different predictions. This can reduce the number of forward passes and in particular, for the binary classification scenario, reduce the number of forward passes by 33%.
>
> **Comments: Additional Baseline Comparisons**
>
> > The paper would also benefit from comparison with additional approaches. If we were to borrow from the literature of Bayesian Deep Learning again, there are techniques such as MC Dropout, BatchEnsembles, etc. I am also open to hearing from the authors on what other baselines they have in mind.
> >
>
> In our method, the Bayesian model is obtained via de Finetti’s theorem and thus defined implicitly. Therefore, we do not know the form of $\theta$, including the likelihood or prior so the ground truth aleatoric and epistemic uncertainties are unobservable. However, standard Bayesian DL methods assume $\theta$  to be the LLM parameters so are not comparable baselines in this setting.
>
> Nevertheless, if we assume a particular likelihood model, the Martingale posterior method [2] can be used to decompose total uncertainty (as it similarly uses de Finetti’s theorem). However, this approach is highly sensitive to the choice of likelihood model. Indeed, as we show in Table B, across various tasks and likelihood choices, the L2 divergence between the Martingale total uncertainty and empirically observed uncertainty remains significant, highlighting the difficulty of selecting a suitable likelihood model in practice.
>
> **Table B: L2 Divergence between uncertainty from Martingale Posterior Total Uncertainty and Empirically Observed Uncertainty**
>
> | Likelihood (Row); Model (Column) | Qwen7B | Qwen14B | Llama8B |
> | --- | --- | --- | --- |
> | Logistic Regression (Linear Features) | 1.023 | 0.804 | 2.158 |
> | Probit Regression (Linear Features) | 1.518 | 0.941 | 2.680 |
> | Logistic Regression (Quadratic Features) | 3.398 | 2.556 | 3.369 |
> | Probit Regression (Quadratic Features) | 3.319 | 2.441 | 2.429 |
>
> **Comments: Optimizing Parameters**
>
> > Can the authors comment on the difficulty on optimizing the new parameters introduced in this approach? It'll be great to also have a profiling view of the different steps involved per experiment and model.
> >
>
> The key hyperparameters in the method are the KL-threshold $\epsilon$ and the choice of auxiliary data $Z_i$.
>
> To ensure that all the auxiliary variables aren’t filtered out by the KL-filtering, we rank the KL divergences $\epsilon_i$ by size and only keep the $r$ auxiliary variables with the smallest KL divergence. This ensures that the method guarantees a decomposition whilst $r$ becomes a proxy for the level of approximation, which is left to the discretion of the user. See Appendix D.3 for further details.
>
> The hyperparameter tuning for auxiliary variable choice is dependent on sampling method. We discuss 4 strategies for sampling $Z$ in the main text and the best performing methods are:
>
> - Perturbing about the predictive task $x^*$: For continuous features, we add Gaussian noise (std = 0.1 $\times$ std of the covariate in $\mathcal{D}$). For categorical inputs, we perturb uniformly across classes. For natural language, we use LLM-based paraphrasing. We find that 5-10 perturbations suffice in most cases; beyond that, improvements in the lower bound are minimal.
> - Repeating $x^*$. This does not require hyperparameter tuning
>
> Overall, these two methods do not require significant hyperparameter tuning and therefore, straightforward to implement.
>
> Regarding profiling, we agree this would provide a useful insight. We provide profiling for the logistic regression problem in the table below as an example.
>
> **Table: Profiling View**
>
> | **Function** | **LLM Calls Per Function** | **Num Function Calls** |
> | --- | --- | --- |
> | $p(y\|x,D)$ | 5 | 1 |
> | $p(u\|Z_i,D)$ | 5 | 5 |
> | $p(x\|u,Z_i,D)$ | 10 | 5 |
>
> ---
>
> [1] Xie, Sang Michael, et al. "An explanation of in-context learning as implicit bayesian inference." *arXiv preprint arXiv:2111.02080* (2021).
>
> [2] Fong, Edwin, Chris Holmes, and Stephen G. Walker. "Martingale posterior distributions." *Journal of the Royal Statistical Society Series B: Statistical Methodology* 85.5 (2023): 1357-1391.
>
> [3] Muennighoff, Niklas, et al. "s1: Simple test-time scaling." *arXiv preprint arXiv:2501.19393* (2025).

---

> > ### Comment · Reviewer_aVVb · 2025-08-06
> > **Response to the rebuttal**
> >
> > I thank the authors for engaging in the rebuttal. I have read the rebuttal posted by the authors as well as the other reviews, and it seems that other reviewers are also concerned about the practical feasibility of this approach. I understand that we can parallelize using multiple values of $Z_i$, but that still incurs quite a bit of cost.
> >
> > Additionally, on my MMLU point, I meant how would we extend to a newer task of question answering and how would we leverage the uncertainty estimates generated from this approach towards that - for instance, can we leverage these uncertainty estimates to reject answering certain questions? I understand that we're a day away from the end of discussion period, so I won't hold it against the authors in my final judgement if they cannot answer this question.
> >
> > However, based on the reviews and discussion so far, I still feel that this paper is a bit borderline, and would greatly benefit from the feedback collected so far and iterating over it to improve the manuscript.

---

> ### Author Response · Authors · 2025-08-06
>
> We thank the reviewer for their reply. We appreciate the concern regarding computational cost, but it is important to note that uncertainty quantification generally demands higher computation costs than standard methods, particularly in post-hoc methods where we are not training a new model architecture from scratch. These approaches fall into two categories: post-hoc calibration, where we tune parameters or the models themselves [1,2,3], or test-time uncertainty quantification, where we compute uncertainty estimates from a model without prior tuning [4]. Whilst post-hoc calibration can offer lower test-time inference costs, it requires costly prior training of the model on a validation dataset, which is computationally infeasible for standard practitioners using LLMs and theoretically impractical if the validation dataset is too small. In contrast, our method does not require prior tuning and can be adapted to situations where the existing set of examples is small.
>
> Thank you for clarifying your MMLU inquiry. Given the time constraint, we conducted a preliminary experiment to determine if we can “leverage these uncertainty estimates to reject answering”. We ran VUD on 60 MMLU-Moral questions and MMLU-Computer Science questions (similar to the results provided in Table A), extracted the total uncertainty (TU), and the decomposed aleatoric uncertainty (AU). Preliminary results in the following table demonstrate that rejecting the top 10% highest aleatoric uncertainty samples yields larger predictive accuracy performance gains than rejecting the top 10% total uncertainty samples.
>
> | Dataset |  | Accuracy (Before Filtering) | Accuracy (After Filtering) |
> | --- | --- | --- | --- |
> | MMLU-CS | TU | 0.7667 | 0.8333 |
> |  | AU | 0.7667 | **0.8519** |
> | MMLU-M | TU | 0.6833 | 0.7037 |
> |  | AU | 0.6833 | **0.7222** |
>
> We provide the following example from MMLU-Moral where we extract the highest decomposed aleatoric uncertainty sample (filtered out by AU) that is not present in the total uncertainty filtered samples (not filtered out by TU).
>
> ```jsx
> Question: For which of these two scenarios does the main character (who uses I/me/my) do something clearly morally wrong, according to ordinary moral standards in the US as of 2020?
> Scenario 1 | I protected the catered food from my cold
> Scenario 2 | I was pole dancing during the erotic show.
> Choices:
> 0: Wrong, Wrong
> 1: Wrong, Not wrong
> 2: Not wrong, Wrong
> 3: Not wrong, Not wrong
> ```
>
> We find that since aleatoric uncertainty measures inherent randomness/stochasticity in $p(y|x)$, AU, when compared with TU, focuses more on evaluating whether the answer to the question is inherently ambiguous. To interpret the question, we find that the example of “Scenario 1 | I protected the catered food from my cold” could be morally “Wrong” or “Not Wrong”. If the character did not attend the event to protect the food, it is “Not Wrong”. If the character did attend but protected the food by staying further away from it, it could be morally “Wrong” since the individual could still pass the germs onto the food. This task is an example of the broader class of selective classification problems [5], which has previously shown that AU can be more effective than total uncertainty as a thresholding metric for selective classification [6]. In general, we believe quantified estimates of AU and EU combined with the rigorous and principled Bayesian justification that VUD provides can have many practical applications in a wide range of domains, such as OOD and hallucination detection, selective classification, active learning, and bandit problems.
>
> We are confident that addressing the reviewer's valuable suggestions has greatly strengthened our paper, making the contributions clearer and further establishing the advantages of VUD. We would particularly like to thank the reviewer for requesting additional MMLU experiments, as we believe these have helped us establish the applicability of our method even further.
>
> ---
>
> [1] Shafer, Glenn, and Vladimir Vovk. "A tutorial on conformal prediction." *Journal of Machine Learning Research* 9.3 (2008).
>
> [2] Gal, Yarin, Jiri Hron, and Alex Kendall. "Concrete dropout." *Advances in neural information processing systems* 30 (2017).
>
> [3] Yang, Adam X., et al. "Bayesian low-rank adaptation for large language models." *arXiv preprint arXiv:2308.13111* (2023).
>
> [4] Hou, Bairu, et al. "Decomposing uncertainty for large language models through input clarification ensembling." *arXiv preprint arXiv:2311.08718* (2023).
>
> [5] Geifman, Yonatan, and Ran El-Yaniv. "Selective classification for deep neural networks." *Advances in neural information processing systems* 30 (2017).
>
> [6] Vazhentsev, Artem, et al. "Hybrid uncertainty quantification for selective text classification in ambiguous tasks." *Proceedings of the 61st Annual Meeting of the Association for Computational Linguistics (Volume 1: Long Papers)*. 2023.

---

### Decision · Program_Chairs · 2025-09-17

**Decision:**

Accept (poster)

**Comment:**

Following the discussion period, the majority of reviewers are positive about this Bayesian approach to uncertainty quantification in generative models such as LLMs, finding the conceptual approach to be interesting and well-motivated.  Concerns about experiments were mostly be addressed by some additional results included in the author rebuttals; please take care that these are integrated in the final manuscript.  One reviewer critiques the method's need to define and evaluate perturbations of the input to assess uncertainty, but the consensus is that this is not a huge assumption, and in fact may be the only option for LLMs that are only accessible via "black box" function evaluations.

Reviewers did have a number of confusions about the submitted manuscript, which were addressed in rebuttals.  Please be sure to carefully revise for readability and clarity.